Resource

# Multi-proteomic profiling of the varicella-zoster virus–host interface reveals host susceptibilities to severe infection

Virginie Girault [1], Alexey Stukalov [1], Madalina Elena Carter-Timofte[2], Jonny Hertzog[3], Melissa Verin[1,4], Katharina Austen[1], Darya A. Haas [1], Lila Oubraham [1], Antonio Piras[1], Susanne Maidl [1], Rupert Öllinger [5], Roland Rad [5], Ulrike Protzer [1,6], Benedikt B. Kaufer [7], Robert J. Lebbink [8], Jan Rehwinkel[3], Trine H. Mogensen [2,9,10] & Andreas Pichlmair [1,4,6,10]✉

Varicella-zoster virus (VZV) infects most humans and causes chickenpox, shingles and central nervous system pathologies. The molecular basis for these phenotypes remains elusive. Here we conducted a multi-proteomic survey on 64 individual VZV proteins and infection-induced perturbations in a neuronal cell line, identifying 900 interactors and 3,618 regulated host proteins. Data integration suggested molecular functions of viral proteins, such as a mechanism for the ORF61-mediated IFI16 degradation via the recruitment of E3 ligase co-factors. Moreover, we identified proviral host factors (MPP8 and ZNF280D) as potential targets to limit infection. Integration of exome sequencing analysis from patients with VZV-associated central nervous system pathologies identified nephrocystin 4 as a viral restriction factor, and its S862N variant, which showed reduced activity and decreased binding to the regulatory proteins 14-3-3. Collectively, our study provides A comprehensive herpesvirus–host interface resource, which aids our understanding of disease-associated molecular perturbations and data-driven identification of antiviral treatment options.

The majority of the human population is infected with the alphaherpesvirus varicella-zoster virus (VZV)[1]. Primary infection triggers a disseminated skin rash known as chickenpox[2]. Concurrently, the virus reaches peripheral neuronal ganglia, where its genome remains latent for life. Latent VZV can reactivate and cause painful, localized skin rashes (shingles or herpes zoster) in up to 50% of infected individuals by age 85 (refs. 3,4). In severe cases, VZV infection results in chronic pain or central nervous system (CNS) infection[5]. More recently, clinical data analyses have revealed unexpected associations between reactivation and neurodegenerative diseases[6]. Despite the availability of preventive anti-VZV vaccination[7], antiviral drugs (for example, acyclovir and amenamevir) remain the most accessible treatment strategy[8,9]. However, the emergence of drug-resistant VZV strains[10]

necessitates the search for new viral and host drug targets. VZV encodes 70 canonical proteins[11], which serve specialized functions and collectively determine viral fitness. Molecular and clinical knowledge indicates that balanced interactions between viral and human proteins are essential in the development of VZV pathogenicity[12–16]; however, these interactions remain largely unknown. Remarkably, most of the interactions known for VZV are inferred from herpes simplex virus (HSV), its closest human virus homologue[17], despite limited amino acid sequence identity and the existence of six VZV-unique proteins[18]. Omics technologies have enabled the mapping of virus–host interaction networks, including for herperviruses[19–22]. Transcriptome and proteome analyses of VZV-infected cells have been reported[23,24], but there is no systematic virus–host protein interaction study on

alphaherpesviruses in human cells. Moreover, orthogonal approaches can help translate multilevel regulatory data into molecular mechanisms of viral activities and disease progression[25,26]. Here we systematically profiled VZV–host interactions in neuronal SK-N-BE2 cells, combining mass spectrometry (MS)-based proteomics and interactomics and unbiased data integration (interactive website: https://varizonet.innatelab.org). Loss-of-function screening of identified host proteins revealed both dependency and restriction factors of VZV replication. Integrating the generated VZV–host interface with whole-exome sequencing (WES) of patients with VZV-driven CNS diseases identified the variant S862N in nephrocystin 4 (NPHP4) as a potential contributor to VZV neuronal pathogenicity.

## Results

### VZV perturbs neuronal and antiviral cellular factors

VZV infection of neuronal tissues can trigger severe pathogenicity[27]. To understand the underlying molecular features, we applied an orthogonal proteomic approach that assayed the interactions between VZV and its host in the neuroblastoma SK-N-BE2 cell line, a neuron-like cell model previously used to study neurotropic infections[28,29] (Fig. 1a). First, SK-N-BE2 cells were co-cultured for 48 h with non-infected or VZV recombinant Oka (rOka)-infected[30] MeWo cells using a transwell system[31] (Extended Data Fig. 1a). Efficient infection of SK-N-BE2 cells was confirmed by flow cytometry analysis (Extended Data Fig. 1b). We used liquid chromatography coupled with tandem mass spectrometry (LC–MS/MS) (Fig. 1a, 'Infection') to analyse the proteome of infected and non-infected control cells. In total, 6,018 proteins were identified, including 66 of the 70 annotated VZV proteins[11]. We identified 158 and 212 cellular proteins with significantly increased or decreased abundance, respectively (Fig. 1b and Supplementary Table 1-1). Functional annotation of the dysregulated proteins (Fig. 1b) combined with Gene Ontology (GO) statistical enrichment analysis (Fig. 1c and Supplementary Table 1-2) highlighted cellular functions that were affected by VZV. Among the most upregulated proteins were cell cycle factors (CKS2, KIFC1 and DLGAP5), the proliferation-associated protein MYCN and components of the spindle apparatus (AURKA, ASPM, CCNB1, CDC20, CDC6, CKAP2, CKAP2L, FAM83D, KIF11, PLK1, SBDS and SPDL1), which is indicative of a promotion of late cell cycle progression. The downregulation of the pro-apoptotic factor TP53 and the upregulation of the anti-apoptotic protein BAG3 indicated an inhibition of apoptosis in VZV-infected cells. We also observed the downregulation of the neuronal growth factor GAP43 as well as neuronal development and differentiation factors (GATA3, HAND2, PHOX2A, TFAP2B, NRCAM, PROX1, TCF12, TCF3, TCF4 and ZEB1). A subset of these proteins contributed to the downregulation of E-box-motif-binding transcriptional regulators, indicating that cellular perturbations after VZV infection are partially due to rearrangement of the transcriptional machinery. Further evidence on perturbed transcription comes from the decreased abundance of transcriptional regulators, including the chromatin silencers SMCHD1 and the SMC5/6 complex. Notably, classical antiviral and inflammatory signatures were not induced, pointing towards tight control of cellular defence mechanisms by VZV in our cellular model. Several dysregulated proteins, which have previously been shown to restrict herpesviruses, could account for this lack of innate immune response: IFI16 (ref. 32), hnRNPA2B1 (ref. 33), the subunits of the RNA polymerase III complex (for example, GTF3C1, GTF3C2 and GTF3C3)[34], XRN2 and CDKN2AIPNL[35] (Supplementary Discussion). Only a subset of proteins of the innate immune system were upregulated, including the interferon-γ receptor IFNGR1, the DNA sensing co-factor TRIM26 and the endoribonuclease SLFN14, which were shown to be active against VZV or influenza virus[36–38]. Altogether, the analysis of host proteome changes during VZV infection revealed pivotal cellular functions activated or repressed in VZV-infected cells.

### A VZV–host interaction network deciphers cell perturbations

The changes in proteome expression during infection arise in part from physical interactions between viral and host proteins. To systematically assess the VZV–human interactome, we used affinity purification (AP) followed by LC–MS/MS (AP–MS) (Fig. 1a, 'Interactome'). Briefly, 64 of the 70 canonical VZV open reading frames (ORFs) (91%) were transduced as carboxy-terminal (C-terminal) V5-tagged proteins in SK-N-BE2 cells (Supplementary Table 2). The expression of 61 of 64 transgenes was verified by western blotting, MS or both (Extended Data Fig. 2a and Supplementary Table 3-1). Three ORFs were not expressed at a detectable level (ORF6, ORF11 and ORF29). Immunofluorescence (IF) analysis confirmed that four representative ORFs (ORF9, ORF49, ORF63 and ORF67) localized to subcellular compartments expected from their known localization during VZV infection[39–42] (Extended Data Fig. 2b), indicating minor effects of the C-terminal affinity tag on intracellular protein distribution. To minimize sampling bias, we used a robotic platform in semi-automated and sample-randomized mode to conduct 392 individual APs. Precipitates were analysed by LC–MS/MS, and cellular interaction partners of the individual VZV proteins were identified using Bayesian linear modelling[43]. This enabled us to construct a global VZV–human interaction network consisting of 1,181 specific interactions between 56 viral baits and 900 cellular 'prey' proteins (Fig. 2a and Supplementary Table 3-2). The topology of the network featured 5 densely connected VZV proteins with more than 100 cellular interaction partners: ORF4 (mRNA export factor ICP27), ORF9 (tegument protein VP22), ORF12 (tegument protein VP11/12), ORF49 (cytoplasmic envelopment protein 3) and ORF63 (transcriptional regulator ICP22). Of the interactions, 76% were unique to individual VZV ORFs (Extended Data Fig. 2c), and no correlation was observed between the number of precipitated proteins and the expression level of each bait (Extended Data Fig. 2d). Moreover, our data recapitulated several reported VZV–human interactions (Supplementary Discussion). We used the annotated subcellular localization of protein interaction partners (preys) to deduce the subcellular localization of the analysed bait protein[44] (Extended Data Fig. 3 and Supplementary Table 3-2). For instance, ORF49 showed the largest proportion of interactors located in the Golgi apparatus, which corroborates its expected localization[40] (Extended Data Fig. 2b). Similarly, the subcellular distribution of ORF63's and ORF67's preys were consistent with the reported localization of the viral proteins in the nucleus and at the cell membrane, respectively[41,42] (Extended Data Fig. 2b). Overall assessment of the interactome data showed a dominant association with nuclear and cytosolic proteins, which may reflect the nuclear replication of VZV and its requirement to engage cytoplasmic factors during the viral life cycle. However, interactomes of several viral ORFs highlighted distinct subcellular localization: ORF10 associated predominantly with nuclear factors and ORF38 associated with cytosolic proteins. Mitochondrial proteins were highly represented among the binders of ORF4, and cytoskeletal proteins were highly represented among the binders of ORF9 and ORF12.

We systematically compared our VZV–host protein–protein interactions with public interaction databases, including the interactions of HSV-1 proteins (Extended Data Fig. 2e and Supplementary Table 2). We confirmed 11 interactions that were reported for herpesvirus homologues, including 9 that have not yet been described for VZV[15,45–52]. Among these was the association of the transcription mediator complex with VZV ORF10, the homologue of the HSV-1 transcriptional activator VP16, with the mediator subunit MED25, which participates in hijacking the host transcriptional machinery[52]. Also, the viral thymidylate synthetase ORF13 interacted strongly with its human homologue TYMS, reproducing the homologous interaction of the Kaposi's sarcoma-associated herpesvirus[20]. However, most of the interactions were not previously described for VZV or HSV-1 proteins. Among such interactions are those of 13 viral proteins for which no cellular host-binding partners have been reported to date (ORF1, ORF13, ORF26/UL32, ORF32, ORF33.5/UL26.5, ORF36/UL23, ORF44/

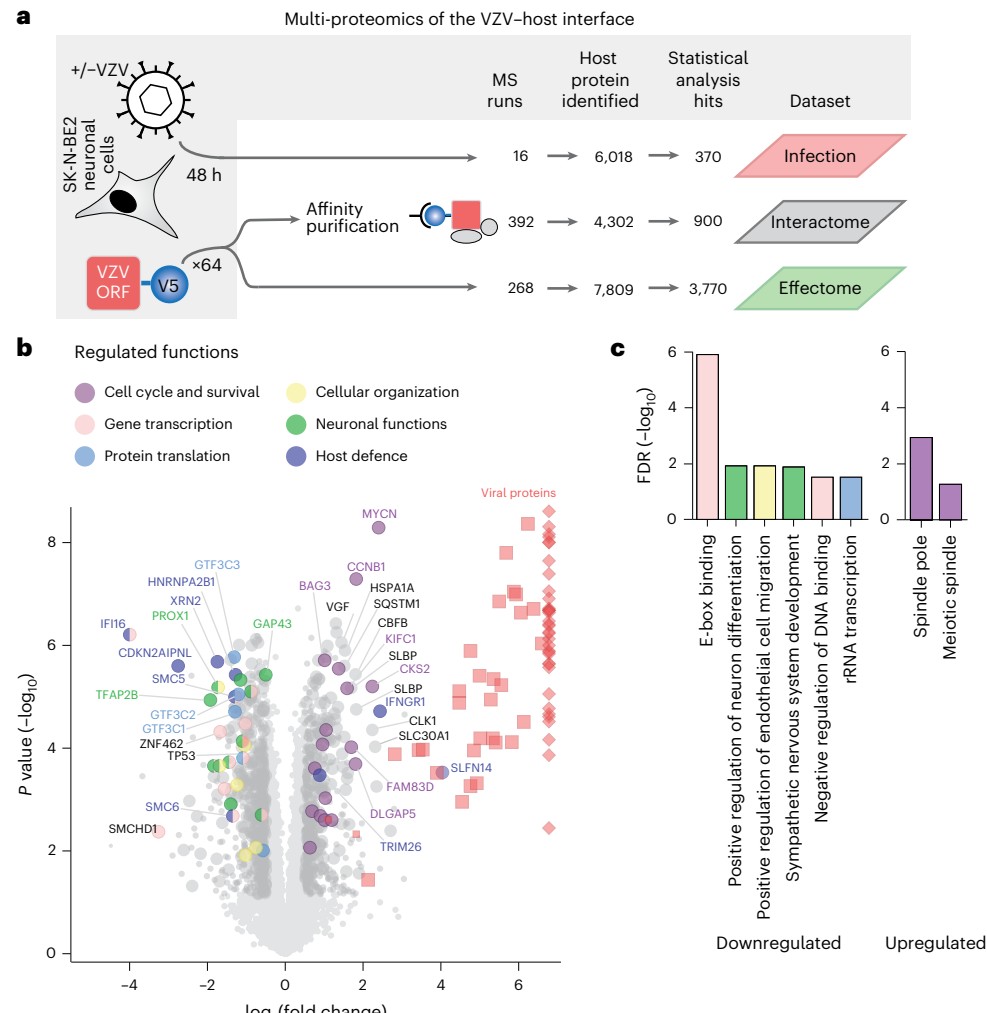

**Fig. 1 | VZV modulates the proteome signature of infected neuronal cells.**
**a**, Experimental design of the VZV–host proteomic survey. Neuroblastoma SK-N-BE2 cells were infected with VZV, and the effects of infection on their proteome were analysed by bottom-up MS to generate the infection dataset. SK-N-BE2 cells were transduced with individual V5-tagged VZV ORFs, and analysed by MS after AP of the tagged viral bait (interactome dataset) and on the proteome level (effectome dataset). **b**, Volcano plot of VZV-induced protein abundance changes in SK-N-BE2 cells infected with VZV for 48 h. Significant host protein changes (two-sided Student's $t$-test, permutation-based FDR $\leq 5 \times 10^{-2}$, |median $\log_2$-transformed fold change| $\geq 0.5$, $n = 4$ independent experiments) are marked in dark grey or coloured according to their GO annotation as presented in panel **c**. Viral proteins are coloured in red. The plot displays one representative assay of two repeats, each including four independent experiments (Supplementary Table 1). The bigger circles highlight changes observed in the two repeats. Diamonds indicate truncated $\log_2$-transformed fold change. **c**, GO terms enriched among the cellular proteins that are downregulated or upregulated in VZV-infected SK-N-BE2 cells as represented in panel **b** (one-sided Fisher's exact test, Benjamini–Hochberg FDR $\leq 5 \times 10^{-2}$, enrichment factor $\geq 4.5$). Regulated GO terms were grouped and coloured according to parental cellular functions, as defined in the legend in **b**.

UL16, ORF49/UL11, ORF52/UL8, ORF56/UL4, ORF57, ORF58/UL3 and ORF67/US7. These interactions revealed associations of these proteins with host factors involved in diverse cellular functions. Several ORF targets indicated regulations of the gene expression machinery. ORF32, which does not have homologues in other herpesviruses and whose function is so far elusive, strongly interacted with the transcriptional factor GTF2B (Fig. 2a), which we confirmed by co-immunoprecipitation and western blotting (Extended Data Fig. 2f,g). IF analysis confirmed nuclear localization of V5-tagged ORF32 (ref. 53), where it co-localized with GTF2B (Extended Data Fig. 2h). We also identified binding of viral uracil-DNA glycosylase ORF59 to the chromatin remodellers CHD8 and CHD9 (Fig. 2a), which regulate genome replication. Several interactions could regulate the cell-intrinsic antiviral immune response, such as binding of the viral E3 ubiquitin ligase ORF61 and the kinase ORF66 to the NF-κB regulators USP11 and NKIRAS2, respectively. Also of interest were the interactions between viral proteins that are involved in virion assembly and release with cytoskeleton- and internal membrane-associated proteins, which point towards mechanisms to exploit the intracellular transport machinery for virion egress. For instance, the cytoplasmic envelopment protein ORF44 interacted with the cholesterol transporter GRAMD1A, and ORF53 interacted with DENND4A and DENND4C. Notably, DENND4 proteins were recently reported as binders of RAB10 (ref. 54), which is a suggested factor for herpesvirus egress[55]. Similarly, we identified contributors to neuronal cytoskeleton organization and spine formation in the interactomes of ORF67 (SIPA1L1, SPICE1 and SYNPO) and ORF68 (IQSEC1), the viral glycoproteins I and E required for intracellular transport of the virion and cell-to-cell spread[56,57].

Many human proteins assemble into complexes to execute their functions. Thus, we specifically searched the VZV interactome for functional host complexes (Fig. 2b and Supplementary Table 3-3). This analysis corroborated reported interactions for ORF12, ORF4 and ORF38 (Supplementary Discussion) and previously unreported findings. The regulator of host and viral transcription, ORF63, bound to the

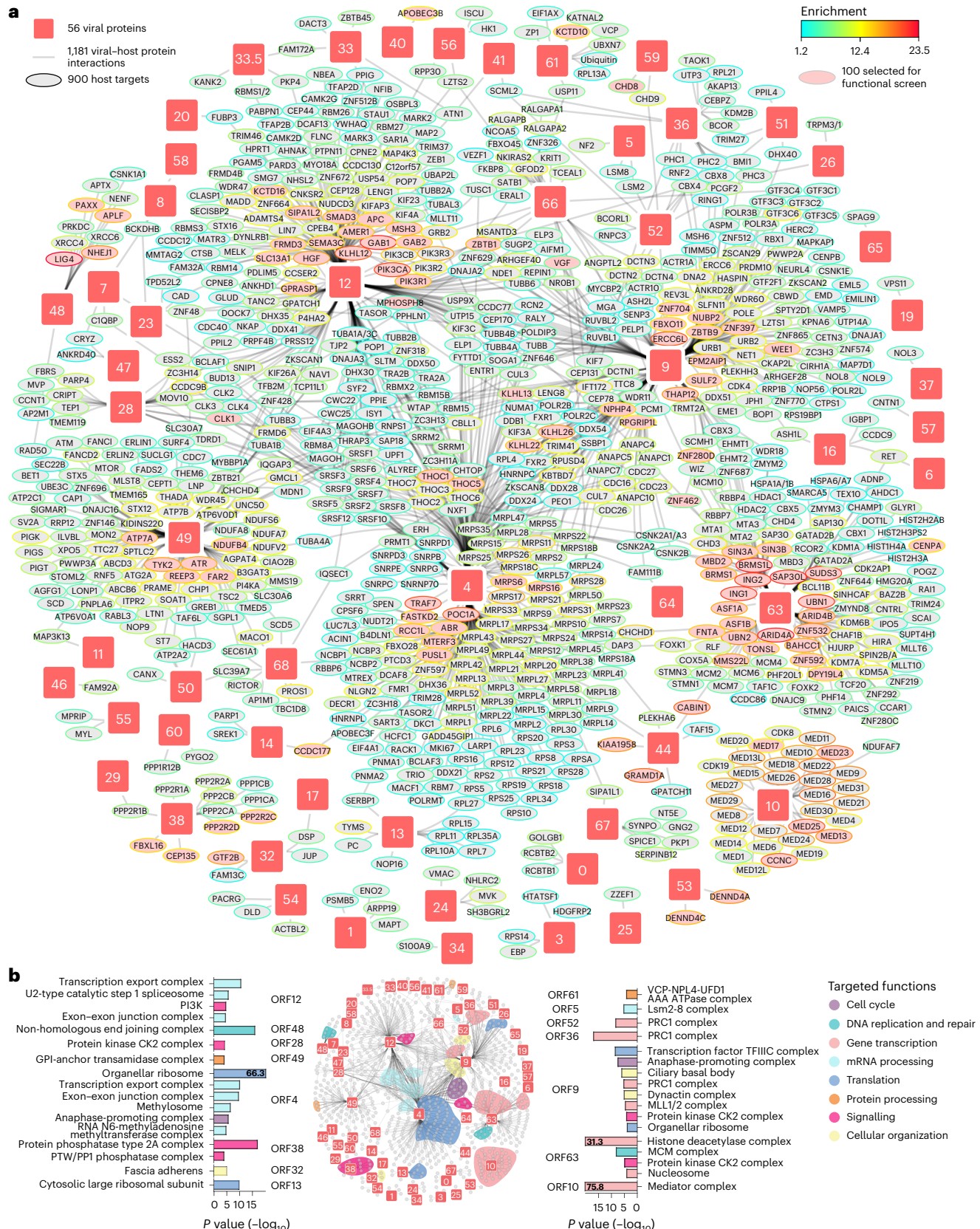

**Fig. 2 | The VZV–host protein–protein interaction network in neuronal cells. a**, Assembled network of the individual V5-tagged VZV protein–host interactomes generated by AP–MS in neuroblastoma SK-N-BE2 cells. VZV baits and host preys are shown as squares and ellipses, respectively (*n* = 4 independent experiments) (Supplementary Table 3). Viral proteins are numbered according to their gene name. The prey border colour specifies the enrichment factor; for preys targeted by several baits, the strongest enrichment is displayed. Host proteins selected for CRISPR–Cas9 knockout screen are highlighted in pink. **b**, Bar plots of GO Cellular Components enriched among the host proteins interacting with individual VZV proteins (one-sided Fisher's exact test, unadjusted *P* ≤ 10⁻⁴). Actual −log₁₀*P* values are indicated when truncated. Targeted GO Cellular Components were coloured according to parental cellular functions as defined in the legend.

histone deacetylase complex and nucleosomal proteins, which are factors of epigenetic regulation of gene expression. The viral E3 ubiquitin ligase ORF61 interacted with the VCP-NPL4-UFD1 AAA ATPase complex (VCP-UBXN7), a known regulator of host E3 ubiquitin ligases[58]. Finally, the ORF9 interactome matched the functions inferred from its HSV-1 homologue VP22 in both cytoplasmic and nuclear compartments. In particular, the enrichment of the ciliary basal body and dynactin complex supported a role in cytoskeleton reorganization[59], and the binding to the PCR1 and MLL1/2 complexes could explain ORF9 localization to chromatin in mitotic cells[60]. This complex-centric analysis highlighted parallel targeting of cellular pathways by VZV proteins, which appear to be of particular importance for the viral replication cycle. ORF4 and ORF12 shared affinity for mRNA processing factors, yet with subcomplex specificities. Although both proteins interacted with the splicing exon–exon junction complex and the transcription export complex, ORF12 co-precipitated with components of the activated spliceosome (U2-type catalytic step 1 spliceosome) and ORF4 enriched for factors of the RNA N6-methyladenosine methyltransferase (m6A) complex. Importantly, engagement of the cellular m6A pathway is essential for efficient viral replication[61], and during HSV-1 infection m6A methylation of host and viral RNAs is modulated by the mRNA export factor ICP27, the homologue of VZV ORF4 (ref. [62]). Complexes linked to DNA replication, DNA maintenance and transcription were also targeted by several viral proteins. ORF36 (thymidine kinase), ORF52 (primase-associated protein) and ORF9 (tegument protein VP22) interacted with components of the polycomb complex (PRC1), a chromatin remodeller known to regulate gene expression. Interestingly, it has been suggested that PRC1 uses the viral DNA replication machinery to remain in proximity to the viral genome during its replication phase[63]. Finally, several enriched cellular complexes indicate previously unreported functions of viral proteins. For instance, ORF48, a conserved herpesvirus alkaline nuclease required for viral DNA processing, associated with the non-homologous-end-joining DNA repair complex, while UL12, the HSV-1 homologue of ORF48, recruited the homology-directed repair machinery and not the non-homologous-end-joining complex at the replication compartment[64]. ORF9 associated with the transcription factor TFIIIC (GTF3C) complex, which may indicate an involvement of ORF9 in the downregulation of GTF3C, as observed in VZV-infected SK-N-BE2 cells (Fig. 1b). Our data also revealed a specific association of the RNA-processing LSM2-8 complex with ORF5 (glycoprotein K), suggesting a previously unrecognized function for the viral protein in splicing. Altogether, our VZV–host interaction network provides an extensive resource describing molecular features used by the virus to engage with its host.

### Proteome changes define specific roles of VZV proteins

Virus–host protein–protein interactions perturb host cell mechanisms, leading to differential gene expression and protein abundance changes during infection. To systematically assess the ability of each VZV protein to affect the abundance of cellular proteins, we used LC–MS/MS to measure the proteomes of SK-N-BE2 cells expressing individual VZV ORFs (Fig. 1a, 'Effectome'). We quantified 7,809 host proteins, of which 3,770 were affected by the expression of at least 1 VZV ORF. In total, we detected 4,923 protein abundance changes across 39 viral ORFs (Fig. 3a and Supplementary Table 4-1). Of these effects, 64% could be attributed to the expression of individual viral ORFs (Extended Data Fig. 4a). The number of affected proteins did not correlate with the expression level of individual baits (Extended Data Fig. 4b), demonstrating the specificity of this analysis. Although little is known about the effects triggered by individual VZV proteins, we could recapitulate effects that could be inferred from HSV-1 homologues, such as the upregulation of β-catenin (CTNNB1) by ORF10, which was reported for VP16 (ref. [65]) (Extended Data Fig. 4c), or the decrease in abundance of IFI16 in cells expressing ORF61, the homologue of ICP0, known to mediate IFI16 degradation during infection[66] (Extended Data Fig. 4d).

To gain a comprehensive overview of the cellular functions affected by VZV protein expression, we evaluated the pathways, transcriptional regulators and protein complexes enriched in the effectome dataset (Fig. 3b and Supplementary Table 4-2). We identified 168 enrichments, which explain individual viral proteins' functions, including effects of viral proteins that are so far uncharacterized. ORF1 regulated ion transport, ORF15 regulated amino acid and lipid metabolism and protein post-translational modification, ORF46 regulated mitochondrial respiration and nuclear organization, and ORF58 regulated energy metabolism, DNA repair and protein secretion. Notably, ORF61 has the ability to induce the peptidyl-diphthamide metabolic pathway (DPH1, DPH2 and DPH5) (Extended Data Fig. 4e), which modulates NF-κB and apoptosis susceptibility[67,68]. Moreover, the glycoprotein N, ORF9A, strongly reduced the abundance of the cytoplasmic tight junction proteins (regulation of blood-brain-barrier permeability; TJP1 and TJP2) (Extended Data Fig. 4f), which are important for viral dissemination, including VZV[69]. Furthermore, this joint analysis of the VZV ORF effectomes allowed us to identify overlapping functionalities of individual proteins. For instance, we observed that ORF38 stabilized adherens cell–cell junction complexes similarly to ORF10 (Extended Data Fig. 4g). ORF8 and ORF12 downregulated protein complexes, which pointed towards an arrest of the cell cycle, as exemplified by CCNB1-CDK1 and CCND1-CDK4 complexes, respectively (Extended Data Fig. 4h). Surprisingly, we could also identify antagonizing activities, which may point towards fine-tuned adjustable regulation of cellular functions. A notable example was the proteins regulated by the neural gene transcriptional silencer REST, which were induced in the presence of ORF66 and ORF9A, while reduced in cells expressing ORF38 (Fig. 3b) (Supplementary Discussion).

This comprehensive catalogue may be valuable for studying other herpesviruses, as it highlights many functions not previously linked to specific herpesviral proteins.

### Systematic data integration uncovers molecular strategies

The activity of individual viral proteins partially reflected the protein expression patterns observed in VZV-infected cells (Fig. 4a, number 1). For instance, VZV infection downregulated the neuronal regeneration factor GAP43 and the protocadherin DCHS1 (Fig. 1b), which we observed after the expression of ORF38 and ORF12, respectively (Extended Data Fig. 5a,b). Similarly, the reduction of IFI16 observed in VZV-infected conditions (Fig. 1b) was apparent in cells expressing ORF61 (Extended Data Fig. 4d). ORF61 is a homologue to HSV-1 ICP0, which was reported to mediate IFI16 proteasome-dependent degradation[66]. Indeed, the expression of ORF61 reduced co-expressed IFI16 to a similar degree as ICP0 (Fig. 4b and Extended Data Fig. 5c). VZV infection downregulated IFI16 in primary human foreskin fibroblast (HFF) cells and the proteasome inhibitor MG132 partially rescued IFI16 expression (Fig. 4c and Extended Data Fig. 5d). Further integration of the interactome data (Fig. 4a, number 2) highlighted an association of ORF61 with the VCP-UBXN7 complex (Fig. 2), a co-factor of cellular E3 ubiquitin ligase-dependent proteasomal degradation[58]. HA-tagged ORF61 expressed from recombinant VZV co-localized with UBXN7 in HFF cells (Fig. 4d) and short hairpin RNA (shRNA)-mediated depletion of UBXN7 (Extended Data Fig. 5e) prevented ORF61-driven reduction of IFI16 (Fig. 4e). Collectively, this shows that integrating orthogonal datasets provides mechanistic insights into molecular mechanisms of VZV ORF proteins that are also apparent in VZV-infected conditions.

Protein expression changes that are identified in the effectome analysis follow a cascade of events (for example, pathways) initiated by the virus–host protein–protein interactions. To systematically assess such functional connections, we mapped the interactors and effects of each individual VZV ORF on the cellular functional network (ReactomeFI[70]) and used a network diffusion approach[25,71], which revealed ORF-specific dysregulated subnetworks (Fig. 4f, Extended Data Fig. 5f and Methods). Particularly valuable subnetworks were identified for

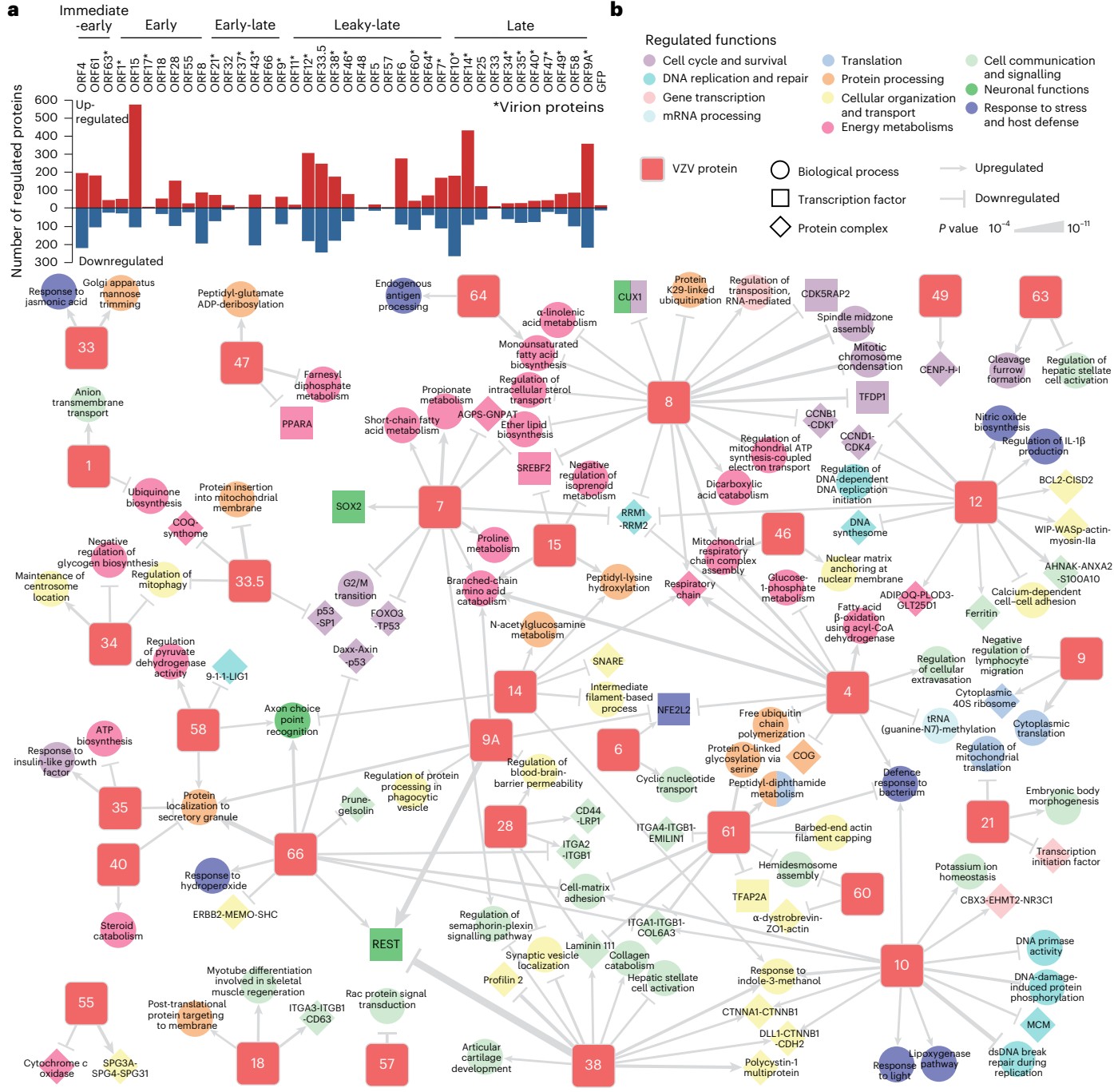

**Fig. 3 | The effectome of viral proteins identifies individual functions. a**, The number of cellular proteins upregulated or downregulated in neuroblastoma SK-N-BE2 cells expressing individual VZV ORFs, as detected by full proteome MS analysis. ORFs are ranked according to their expression kinetics during viral replication, and viral proteins that are part of the virion are annotated with an asterisk (*) (*n* = 4 independent experiments) (Supplementary Table 4, 'Effectome significant' and 'is effect'). **b**, Network of enriched pathways (GO Biological Processes), transcriptome factor target gene sets (OmniPath) and human protein–protein complexes (CORUM, IntAct) among the cellular proteins that are regulated in SK-N-BE2 cells expressing the indicated individual VZV proteins (one-sided Fisher's exact test, unadjusted $P \leq 10^{-4}$). Viral proteins are numbered according to their gene name. Edge thickness indicates the *P* value. Regulated terms were coloured according to parental cellular functions as defined in the legend. dsDNA, double-stranded DNA.

ORF4, ORF7, ORF9, ORF10, ORF12, ORF49, ORF61 and ORF66 (Supplementary Data 1). The ORF12 subnetwork, for instance, prioritized 45 of the 220 targets detected by AP–MS (Extended Data Fig. 5g). Several of these associations were enriched for proteins that are consistent with known ORF12 functions (featured in Fig. 4g). Notably, it linked ORF12's targeting of the phosphoinositide 3-kinase (PI3K) complexes (PIK3R1, PIK3R2, PIK3R3, PIK3CA and PIK3CB) and receptor tyrosine kinase

(RTK) complexes (GRB2, GAB1, GAB2 and PTPN11) to an anti-apoptotic function, and suggested the protein kinase C-α (PRKCA) and the cyclic AMP-responsive element-binding protein (CREB1) as mediators of this activity (Supplementary Discussion). ORF9 is a tegument protein that binds to the cytoskeletal network during virion assembly and egress[72,73] and is essential for viral replication[74]. We identified a functional subnetwork connecting 49 of the 209 targets and 14 of the

41 effects of ORF9 (Extended Data Fig. 5h). The network was enriched for proteins involved in cytoskeleton organization (Fig. 4h) and further highlighted two ORF9-bound complexes (UV-DDB-ubiquitin ligase and CBX4-PCGF2-PHC2) that are known to recruit and SUMOylate the upregulated centrin 2 (CETN2) (Supplementary Discussion). In sum, integrating effectome data with the VZV–host protein–protein interaction network constitutes an extensive resource that consolidates expected and unreported activities of viral proteins and allows for the generation of hypotheses on the molecular mechanisms engaged during VZV infection.

## Functional screen identifies host factors of VZV infection

To characterize the functional role of the cellular proteins identified through the profiling of VZV–host interactions, we selected the 116 most prominent proteins (Methods) for a CRISPR–Cas9-based loss-of-function screen evaluating their effects on VZV growth (Fig. 5a). Briefly, differentially fluorescently labelled control and Cas9-expressing (knockout) SK-N-BE2 cells were co-transduced with individual single guide RNAs (sgRNAs) and infected by co-culture with a recombinant red fluorescence protein (RFP) reporter VZV for 48 h. Cells were analysed by flow cytometry (Fig. 5b), allowing for the calculation of the normalized median RFP intensity (MRI). The experimental design adjusted for variability of cell density, which could affect VZV spread (Extended Data Fig. 6a), and allowed the exclusion of toxic knockouts. Overall, we identified five anti-VZV and seven pro-VZV host genes (Fig. 5c). The interferon-γ receptor 1 (IFNGR1) was recapitulated as an essential restriction factor of VZV[75]. Host proteins with unexpected antiviral functions were MTERF3, ABR, KLHL12 and ASF1A. Among the proteins with proviral functions were ING1, GRAMD1A, PUSL1, MPHOSPH8 (MPP8), ZNF280D and HGF. The proviral function of PIK3CA is corroborated by the network diffusion analysis of ORF12 (Fig. 4g) and is in line with the previously reported dependency of VZV replication on activated PI3K signalling[76]. The screen revealed unexpected proviral activity of MPP8, which is reported to silence viral DNA and provirus expression as part of the HUSH complex[77]. Similarly interesting was ZNF280D, an uncharacterized zinc-finger-containing protein with a putative function in the regulation of transcription. To confirm these effects, we generated stable knockout SK-N-BE2 cells, which we validated for normal cell growth and gene depletion at both gene and protein levels (Extended Data Fig. 6b–h), and analysed the propagation of VZV in gene-depleted or non-targeting control (NTC) cells. We confirmed the proviral activity of both MPP8 and ZNF280D by flow cytometry (Fig. 5d,e) and VZV growth kinetics using live-cell imaging (Extended Data Fig. 6i,j). Proteome analysis revealed that the depletion of MPP8 destabilized the two other components of the HUSH complex, TASOR/FAM208A and PPHLN1, supporting the notion that dysregulating components of the HUSH complex can limit VZV spread (Extended Data Fig. 6k and Supplementary Table 5-1). Together, this functional screen characterized viral restriction and dependency factors and confirmed the efficiency of our approach to identify functional VZV–host interactions by multi-proteomic profiling.

## Patient gene variants affect VZV restriction factors

VZV can spread to the CNS, where it triggers serious disorders, such as encephalitis, meningitis or cerebral vasculitis[27], which are often associated with inborn errors in immunity[78]. Importantly, clinical reports of severe diseases associated with VZV infection in individuals who are otherwise immunocompetent imply that so-far-uncharacterized restriction genes might be affected by deleterious variants in these patients[79–81]. We thus screened patients with VZV CNS infection for rare genetic variants and linked them to our VZV–host proteomic data (Fig. 6a). Thirteen patients diagnosed with VZV-associated encephalitis and/or meningitis or cerebral vasculitis were subjected to WES and variant calling (Supplementary Table 6-1). Integrating the rare

predicted deleterious variants with the proteins of the VZV–host interface identified by our multi-proteomic profiling highlighted 66 genes with rare, potentially disease-causing gene variants, including in-frame deletions, missense mutations, frameshifts and premature stop codons (Supplementary Table 6-2). Notably, 11 of the identified genes encode for proteins that had been identified in this study as potent contributors of VZV–host interactions and were included in the VZV replication loss-of-function screen (Fig. 5c). Three candidates showed strong antiviral activity: the transcription factor ZNF592, interacting with the viral transcriptional regulator ORF63; the cytoskeleton modulator NPHP4, binding to ORF4 and ORF9; and MED13, a kinase subunit of the mediator complex, co-precipitated by the VZV transcriptional regulator ORF10. Interestingly, none of these candidates had previously been reported to have direct antiviral function. Among these three, the genetic variant encoding for NPHP4(S862N), which was identified as monoallelic in a 39-year-old woman diagnosed with VZV meningoencephalitis, was predicted to be the most deleterious (Combined Annotation Dependent Depletion score = 25.1; Supplementary Table 6-2). We generated a stable NPHP4 knockout in the SK-N-BE2 cell line and validated it by DNA sequencing and western blotting (Extended Data Fig. 7a,b). We confirmed that the decrease of NPHP4 expression did not impair cell growth (Extended Data Fig. 7c), and increased VZV spread in NPHP4-depleted cells compared with NTC cells (Fig. 6b,c and Extended Data Fig. 6i,j). Reconstitution of NPHP4 knockout cells by HA-tagged NPHP4 significantly reduced VZV replication compared with the knockout cells transduced with an empty vector, which confirmed the specificity of the knockout phenotype (Fig. 6b,c). Comparison with NTC cells revealed a partial rescue effect. Notably, reconstitution of knockout cells with the HA-tagged NPHP4(S862N) mutant (Extended Data Fig. 7d) did not impair VZV growth compared with the empty vector control, confirming that the identified patient mutation is of functional importance to inhibit VZV growth (Fig. 6b,c).

To further decipher the function of NPHP4, we used transcriptome analysis of NTC and NPHP4-deficient SK-N-BE2 cells that were mock infected or VZV infected using a transwell co-culture infection system. VZV infection induced overall similar transcriptional changes in wild-type and NPHP4-deficient cells (Extended Data Fig. 7e,f and Supplementary Table 5-2). A total of 21 genes were detected as differentially expressed in infected NPHP4-deficient cells compared with control cells. Notably, VZV infection triggered upregulation of WNT target genes (*FOSL1* and *ZEB2*), which was more prominent in NPHP4-depleted cells compared with NTC cells (Extended Data Fig. 7g,h). None of the differentially regulated genes were linked to antiviral immunity associated with herpesvirus infections (for example, *MX2*, *IFI16*, *CGAS*, *IFIT*s and *OAS*s[32,82–84]), indicating that the anti-VZV function of NPHP4 is probably independent of the innate immune defence.

To gain mechanistic insights into the role of the S862 residue of NPHP4, we analysed the interactomes of the NPHP4 wild-type and the S862N mutant proteins in SK-N-BE2 cells. Most interactors of both NPHP4 variants were similar (Fig. 6d, Extended Data Fig. 7i and Supplementary Table 5-3). The wild-type and mutant proteins bound similarly to the basal body proteins RPGRIP1L and NPHP1, recapitulating known findings[85]. However, direct comparison of the NPHP4 wild-type and S862N mutant revealed differential binding of several 14-3-3 proteins, most prominently 14-3-3ε (YWHAE) and 14-3-3η (YWHAH) (Fig. 6d). Strikingly, closer inspection of the S862 region of NPHP4 revealed a canonical 14-3-3 binding motif ($RX_{1-2}SX_{2-3}S$)[86] in which the distant serine residue was changed into an asparagine in the S862N mutant, thereby destroying the motif and probably hampering the interaction with the 14-3-3 partners.

Altogether, by combining our VZV–host interaction multi-proteomic data with host genetic information, we identified restriction factors relevant in a clinical context. Most importantly, we discovered a host genetic variant encoding for NPHP4(S862N), which loses its

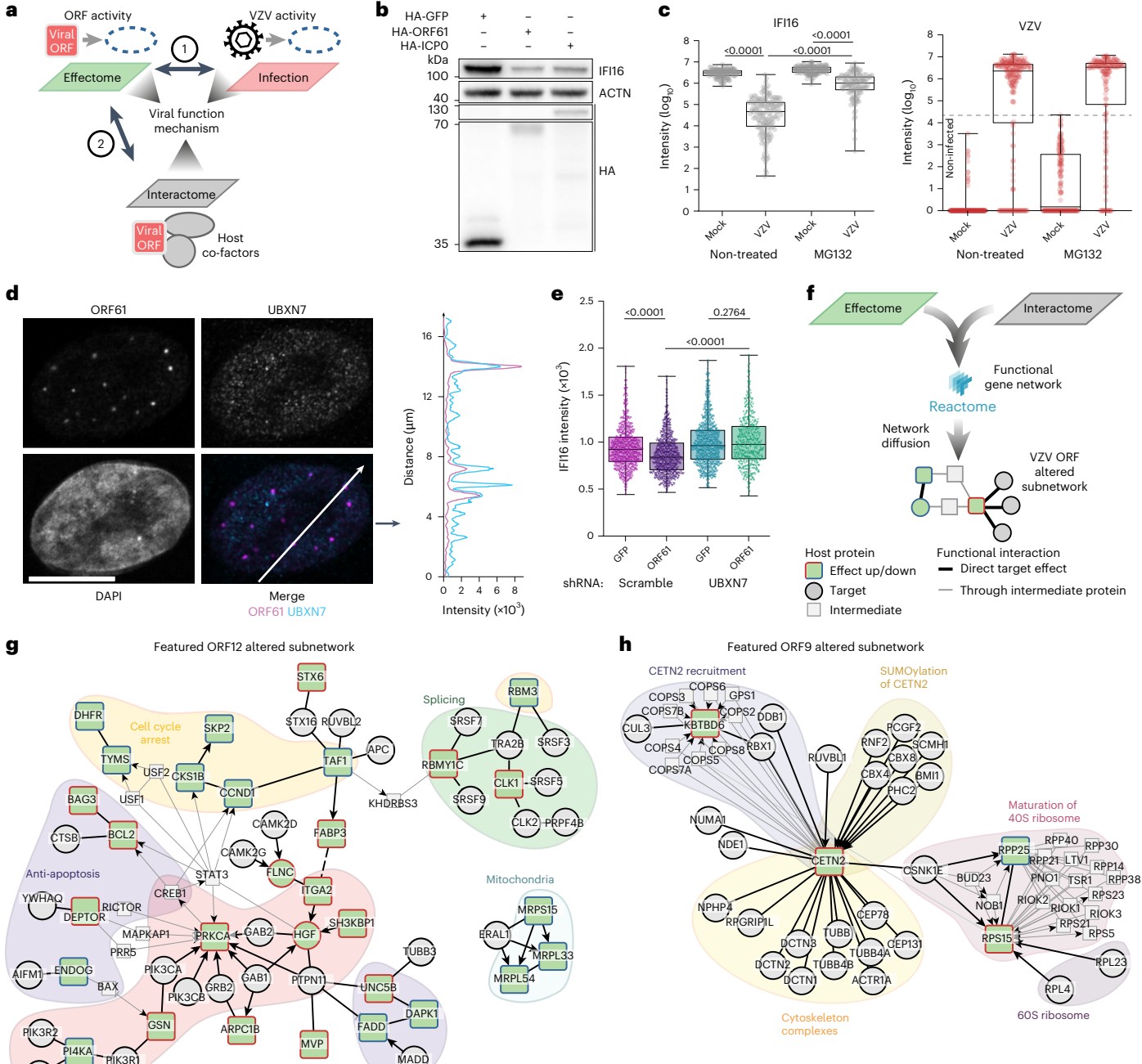

**Fig. 4 | Multi-proteomic data integration. a**, The orthogonal analysis of infection and effectome datasets (1) or effectome and interactome datasets (2) provides hypotheses on the molecular mechanism involved in VZV functions. **b**, WB analysis of IFI16 in HEK293T cells after co-transfection with HA-GFP, HA-ORF61 or HA-ICP0. Representative of $n = 3$ independent experiments. Data are summarized in Extended Data Fig. 5c. **c**, IF analysis of HFF cells mock infected or infected with recombinant HA-ORF61 VZV for 8 h, treated or not with the proteasome inhibitor MG-132. Cells were stained for IFI16 and VZV (ORF61) and with DAPI. More than 120 nuclei per condition were analysed at ×10 magnification. Minimum and maximum, first and last quantiles, and the median $log_{10}$ intensity of IFI16 or VZV are indicated (two-sided Mann–Whitney test, unadjusted). Non-infected cells, defined by the maximal observed VZV signal in mock infected cells (grey dashed line), were excluded from the infected conditions in the IFI16 plot. Representative of $n = 3$ independent experiments. Images are presented in Extended Data Fig. 5d. **d**, IF analysis of the subcellular localization of UBXN7 and VZV ORF61 in HFF cells infected with recombinant HA-ORF61 VZV for 8 h. Cells were stained for UBXN7 and ORF61 and with DAPI, and analysed at ×63 magnification. Each

channel and the merge of ORF61 and UBXN7 are displayed for one representative cell. Scale bar, 10 μm. The line profiles represent UBXN7 and ORF61 intensities extracted as indicated by the white arrow in the merge image. Representative of three independent experiments. **e**, IF analysis of HFF cells, either control or UBXN7-depleted by shRNA expression, and transduced with V5-ORF61 or V5-GFP. Cells were stained for IFI16 and V5 tag and with DAPI. More than 500 nuclei per condition were analysed at ×20 magnification. Minimum and maximum, first and last quantiles, and the median intensity of IFI16 are indicated (two-way ANOVA, adjusted with Tukey's method). Representative of $n = 2$ independent experiments. **f**, Interactome and effectome data were mapped onto the cellular gene network ReactomeFI and submitted to network diffusion analysis (Methods and Extended Data Fig. 5f) to generate individual viral ORF-altered gene subnetworks. **g,h**, Featured subnetworks resulted from network diffusion predictions of VZV ORF12 (**g**) and ORF9 (**h**). Edges indicate ReactomeFI connections that passed the random walk transition probability threshold (0.05). Complete HotNet output subnetworks are shown in Extended Data Fig. 5g,h, and interactive versions are given in Supplementary Data 1.

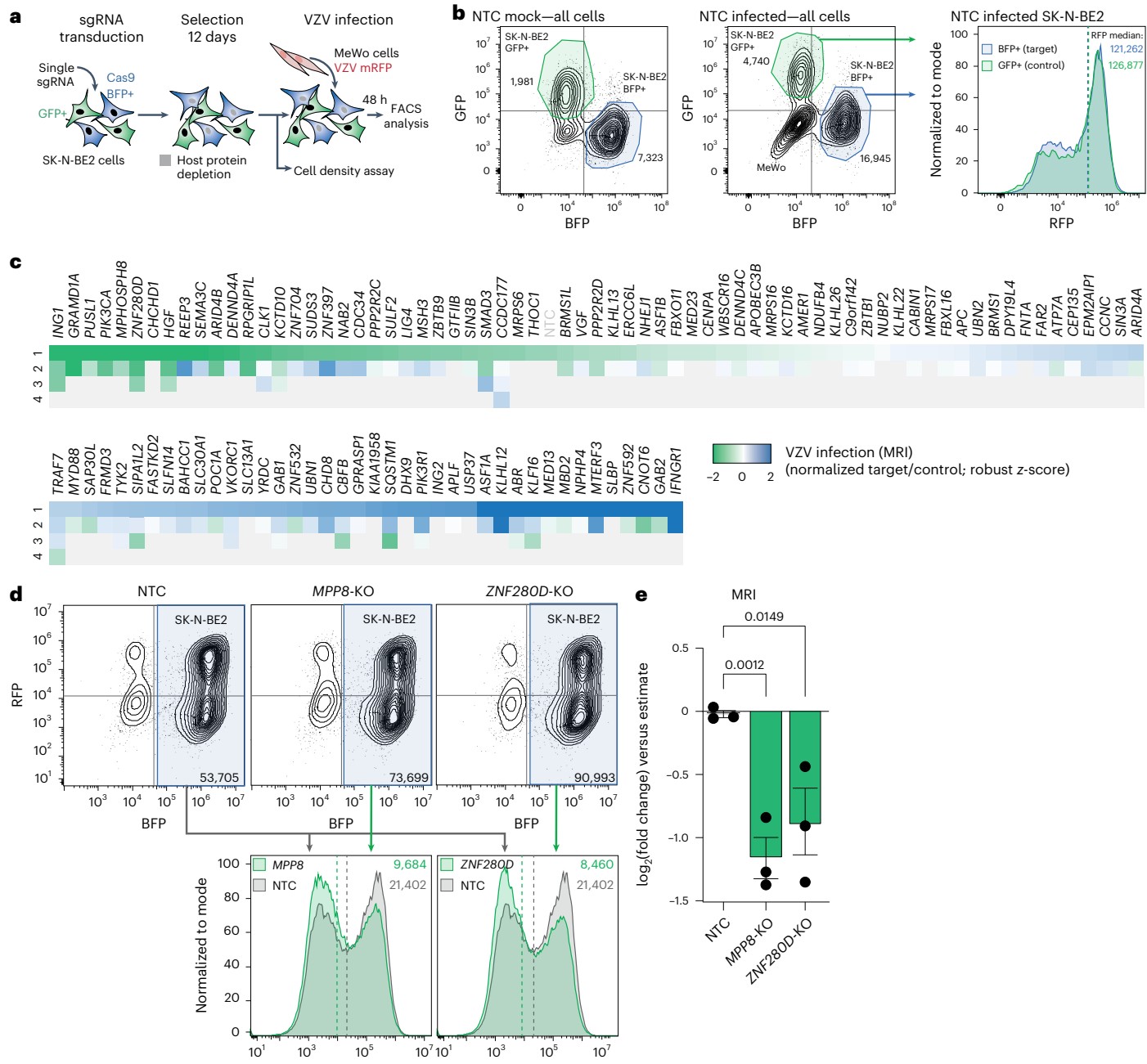

**Fig. 5 | Loss-of-function screen identifies VZV restriction and dependency factors. a**, Host gene knockout screen for VZV replication. Cas9-free GFP-expressing and Cas9 BFP-expressing SK-N-BE2 cells were co-cultured and co-transduced with sgRNA targeting host genes. Following selection, cells were infected by co-culture with mRFP-VZV(pOka)-infected MeWo cells and analysed by flow cytometry. **b**, Flow cytometry gating strategy of the knockout screen presented in **a** allows for the exclusion of the MeWo inoculum cells and individual gating of the target (BFP+) and control (GFP+) cell populations, as shown for representative NTC mock and infected wells. The number of gated cells is indicated. The histogram below shows the overlaid distribution of the RFP intensities within target and control cells, indicating their respective infection level. MRIs are indicated as values and dashed lines. **c**, Array knockout screen was performed on 116 host genes selected from the VZV proteomic survey. The heat map shows the z-scored target-to-control normalized MRI for each sgRNA (number 1 to 4) per gene, averaged across duplicates. Targeted host genes are sorted according to their most potent sgRNA. **d**, Validation by flow cytometry analysis of the function of MPP8 and ZNF280D. Knockout (KO) or NTC BFP-expressing SK-N-BE2 cells were infected via co-culture with mRFP-VZV(pOka)-infected MeWo cells. Gating the BFP+ SK-N-BE2 population allows for exclusion of the inoculum MeWo cells. The number of gated cells is indicated. The histograms below show the overlaid distribution of the RFP intensities within SK-N-BE2 cells, NTC (grey) or knockout for the indicated gene (green) (representative well). MRIs are indicated as values and dashed lines. **e**, Fold change of the MRI within SK-N-BE2 cells, NTC or knockout for the indicated gene, and infected with mRFP-VZV(pOka) as presented in **d**, compared with the estimate (Methods). Mean ± s.e.m. is indicated (n = 3 independent experiments) (one-sided Student's t-test, unadjusted).

ability to associate with regulatory 14-3-3 proteins and could thereby contribute to the development of severe VZV CNS pathology (Fig. 6e). The identified restriction factor variants may represent inborn errors in immunity in patients with severe VZV CNS infection.

## Discussion
The comprehensive proteomic survey presented here enabled a profound analysis of the interactions between VZV and the human host; however, such a design entails certain limitations that are linked to the

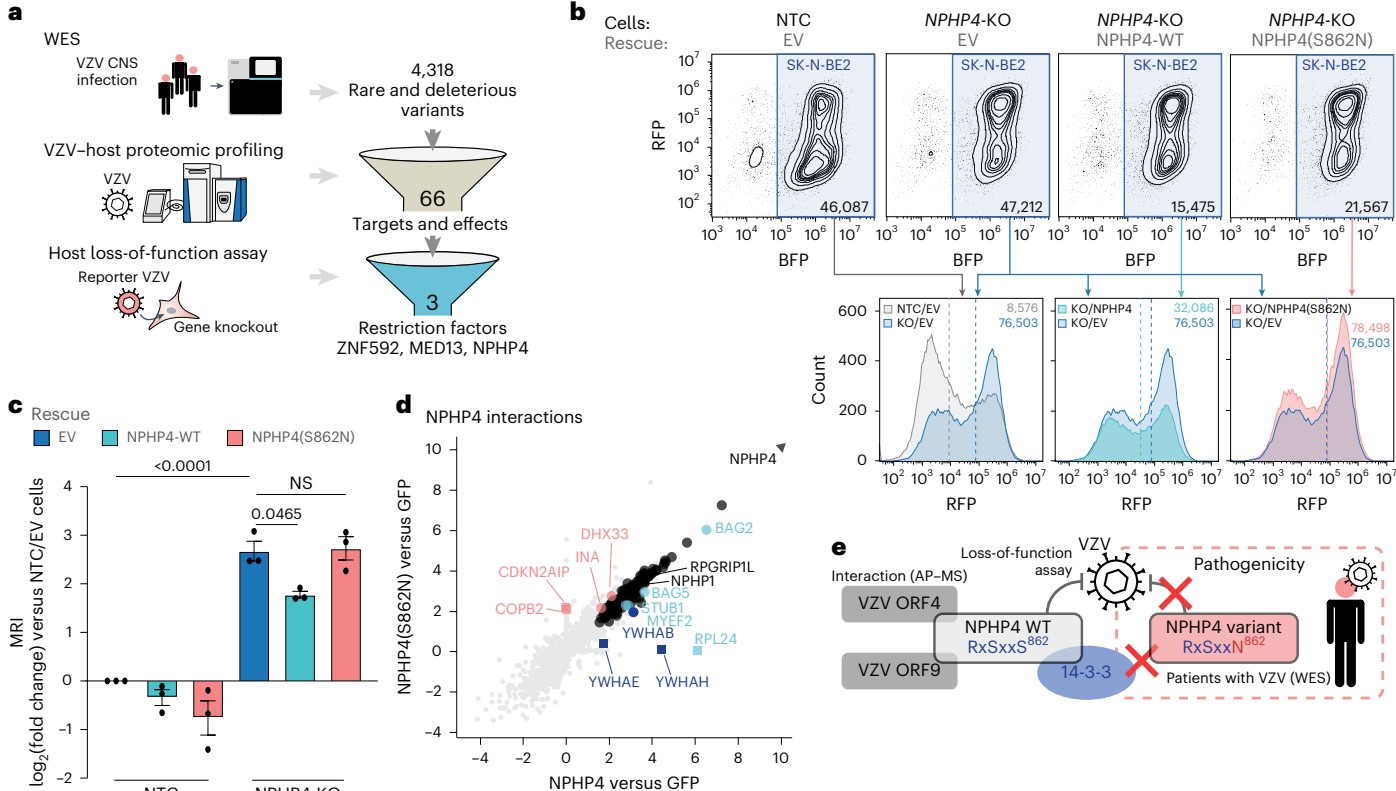

**Fig. 6 | Rare variant analysis in patients with VZV CNS infection identifies a mutation in the restriction factor NPHP4. a**, Prioritization of genes identified by WES variant analysis from 13 patients affected by VZV CNS infection who were otherwise immunocompetent. WES identified rare and predicted deleterious variants in patients, to which two biological filters were applied: the proteomic profiling of VZV–host interactions and the results of the VZV replication functional assay. **b**, Validation by flow cytometry analysis of the function of NPHP4 and characterization of the NPHP4(S862N) variant. KO or NTC BFP-expressing SK-N-BE2 cells, transduced with an empty vector (EV), wild type (WT) NPHP4 or NPHP4(S862N) as indicated, were infected via co-culture with mRFP–VZV(pOka)-infected MeWo cells. Gating the BFP+ SK-N-BE2 population allows exclusion of the inoculum MeWo cells. The number of gated cells is indicated. The histograms below show the overlaid distribution of the RFP intensity within SK-N-BE2 cells across rescue conditions compared with the knockout non-rescue cells (KO or EV) (representative well). MRIs are indicated as values and dashed lines. **c**, Fold change of the MRI within SK-N-BE2 cells that were NTC or knocked

out for NPHP4 and infected with mRFP–VZV(pOka), as presented in **b**, following the transduction of an EV, NPHP4-WT or NPHP4(S862N), compared with the NTC or EV control. Mean ± s.e.m. is indicated (*n* = 3 independent experiments) (one-way ANOVA test). **d**, Interactomes of HA–NPHP4 wild-type and S862N generated by AP–MS in neuroblastoma SK-N-BE2 cells. log$_2$ enrichment factors are shown compared with the HA–GFP control. Differential binding partners that were identified from statistical analysis of the comparison between the WT and the variant constructs are indicated: increased enrichment by the WT is shown in light blue circles; increased enrichment by the variant is shown in pink circles; grey circles indicate shared enrichment by both WT and the variant; binders that pass the statistical threshold for the comparison of WT versus mutant are indicated by squares; and 14-3-3 proteins are indicated in dark blue (*n* = 4 independent experiments) (Supplementary Table 5). **e**, Review of the virus–host interactions between VZV and NPHP4 as identified in our multi-proteomic and functional analysis, and the functional characterization of the S862N variant. NS, not significant.

choice of virus strain, cell type and experimental set-up. Thus, interactions and effects specific to other VZV strains or clinical isolates, or interactions involving cellular proteins exclusively expressed in other cell lineages (for example, skin epidermis or T cells), would not be identified in this study (Supplementary Discussion).

Our screening showed highly specific functions (Extended Data Fig. 4a), supporting the notion that individual viral gene products evolved to fulfil distinct tasks to promote viral propagation in a concerted action. Interestingly, the data of individual VZV proteins contained patterns that are not fully reflected in the proteomic data of VZV infection. For instance, the perturbation of the diphthamide metabolizers by ORF61 (Extended Data Fig. 4e) was not observed in VZV-infected cells. This discrepancy may partly be due to the improved sensitivity of the effectome analysis compared with the infected proteome (7,809 versus 6,018 proteins detected), which is favoured by the parallel proteomic analysis of the large number of samples[87] and the resulting higher robustness of the statistical models. Our data also suggest multifunctionality of viral proteins, leading to complex, sometimes opposing, regulation of cellular pathways. A notable example is the

cell cycle progression induced in VZV-infected SK-N-BE2 cells (Fig. 1b), which contrasts with the downregulation of cell cycle factors by ORF12, ORF8 and ORF7 (Fig. 3b). Similarly, the REST response was differentially regulated by ORF66, ORF9A and ORF38 (Fig. 3b). Such opposing effects could be explained by time-dependent expression of individual viral proteins[88] and replication phase-specific viral activities, both of which may be obscured in a population of non-synchronously infected cells (Supplementary Discussion).

Due to the conservation of virus–host protein interactions and ORF functionalities between herpesviruses[89] (Extended Data Fig. 2e), the data presented here provide valuable insights into general herpesvirus biology. The regulation of IFI16 abundance by ORF61 (Extended Data Fig. 4d) or the binding of TYMS by ORF13 (Fig. 2a) are examples of activities conserved between herpesviruses[20,66]. Our analysis allows us to explain the functions of viral proteins by associating their cellular binding partners with their ability to modulate cellular pathways. Some of the identified regulated pathways are essential for herpesvirus biology and associated diseases, including the DNA damage response regulated by ORF48 (UL12); chromatin remodelling

by ORF63 (ICP22), ORF9 (VP22), ORF36 (UL23) and ORF52 (UL8); host gene expression by ORF10 (VP16) and ORF32; virion release by ORF53 (UL7); evasion of the innate immunity by ORF61 (ICP0), ORF9 (VP22) and ORF66 (US3); and neuronal differentiation by ORF38 (UL21) (Supplementary Discussion).

The increased resistance of herpesviruses to nucleoside analogues[90,91] underlines the need for alternative therapeutic approaches. Here, we identified several cellular proteins that are required for the efficient growth of VZV (Fig. 5c and Supplementary Discussion). These include proteins of the trans-Golgi network (DENND4A) and cholesterol transport (GRAMD1A), which are associated with ORF53 and ORF44, respectively (Fig. 2a), and probably participate in viral assembly and egress. Moreover, we extensively validated that the targeting of the HUSH complex, interacting with ORF12 and ORF4 (Fig. 2a), and of the transcriptional factor ZNF280D, interacting with ORF63 (Fig. 2a), would impair VZV growth (Fig. 5e and Extended Data Fig. 6g). Disrupting these virus–host interactions using small molecules might complement current treatment strategies against VZV and herpesviruses in general. Collectively, our study provides entry points for therapeutic targeting of essential virus–host interactions and the framework to use computational methods to identify druggable targets aimed at mitigating VZV[25,92].

Finally, this resource can facilitate identifying host genetic susceptibilities to infection. Rare genetic variants are strongly associated with severe disease phenotypes[93] but are difficult to identify, particularly in sporadic diseases such as VZV encephalitis. By integrating our proteomic data with WES data of patients with VZV, we pinpointed rare variants in genes that we characterized as targets of VZV (Supplementary Table 6). The identification of an NPHP4(S862N) loss-of-function variant in a patient presenting with VZV-associated meningoencephalitis (Fig. 6c,d) shows the power of such a multilayered analysis to consolidate true functional hits (Supplementary Discussion). We expect that the other host proteins identified by this integrative work are valuable candidates and, conversely, that the whole virus–host interface presented may help to prioritize genetic mutations from other WES data, which together should inspire follow-up studies of high clinical impact.

Collectively, our study constitutes an extensive resource (available at https://varizonet.innatelab.org), which aids comprehensive exploration of herpesvirus biology and pathogenesis and promotes the discovery of intervention strategies to limit VZV infection.

## Methods

### Ethics
The patient study included in this work complies with the Declaration of Helsinki and national ethics guidelines and was approved by the Danish National Committee on Health Research Ethics (number 1-10-72-275-15), the Data Protection Agency and the institutional review board. Patients provided oral and written informed consent for participation in the study.

### Cell line and reagents
**Cell lines.** SK-N-BE2 cells were kindly provided by R. Klein (MPI of Neurobiology). MeWo cells were kindly provided by A.-V. Borbolla (MHH). HEK293T (CRL-11268) cells were purchased from ATCC. HeLa Kyoto-expressing GFP-tagged GTF2B from the bacterial artificial chromosome (BAC) transgene was from I. Poser[94]. HFF-1 (SCRC-1041) cells were a kind gift from M. Brinkmann (HZI). All cell lines were tested to be mycoplasma free. All cells were maintained in culture medium: DMEM medium (high glucose, pyruvate; Gibco Fisher Scientific), supplemented with 10% fetal calf serum (FCS), 100 µg ml$^{-1}$ streptomycin and 100 IU ml$^{-1}$ penicillin. Low passage cell lines were kept as frozen stock in liquid nitrogen after resuspension in culture medium supplemented with 45% FCS and 10% DMSO. Blasticidin (15205; Sigma-Aldrich), puromycin (P8833; Sigma-Aldrich) and zeocin (Invitrogen) were used for transduced cell selection in this study. For proteasome inhibition, we used MG-132 (474787; Sigma-Aldrich).

**Plasmids.** pDONR221 was purchased from Invitrogen. ICP0-pDONR201 was kindly provided by G. Superti-Furga (CEMM). pSicoR-SpCas9-ZeoR[95], pLenti6.3-GFP-blastR (number 40125; Addgene) and pWPI-tagBFP-blastR (generated in this study as described later) were used to generate lentiviruses, allowing the expression of Cas9 or reporter fluorescence proteins in SK-N-BE2 cells. The tagBFP expression cassette was amplified from pSCRPSY-PAC2A-tagBFP (kindly provided by C. Rice; Rockefeller University) and inserted into pWPI-blastR. The library of VZV ORFs were ligated into pCR8/TOPO entry vector (Invitrogen) and shuttled into pLenti6.3-TO-V5-DEST-BlastR (Invitrogen) via Gateway recombination, allowing expression of C-terminal V5-tagged proteins and expression of blasticidin resistance. The library of host gene gRNAs were ligated into pSicoR-U6-gRNA-EFS-PuroR[95], allowing expression of the individual gRNA and puromycin resistance. pENTR encoding for a codon-optimized *NPHP4* gene was purchased from Twist Bioscience. pWPI-nHA-PuroR-GW, allowing the expression of amino-terminal (N-terminal) HA-tagged proteins and puromycin resistance was kindly provided by A. Plaszczyca (University of Heidelberg). The pWPI-nHA-PuroR encoding wild type NPHP4 and NPHP4(S862N), VZV ORF61, HSV-1 ICP0 and GFP expression were generated in this study as described later. Packaging vectors pMD2-VSV-G and pCMV-Gag-Pol or pCMVR8.91 (Didier Trono's laboratory) were used to produce lentiviruses. pLVX, allowing lentiviral transduction of shRNA and expression of puromycin resistance, was kindly gifted by M. Friedrich (Technical University of Munich).

**Antibodies.** TFIIB (western blot (WB) 1:1,000; 4169; Cell Signaling Technology), MPP8 (WB 1:1,500; 16796; Proteintech), ZNF280D (WB 1:1,000; PA5-56410; Invitrogen), NPHP4 (WB 1:3,000; A8934; Abclonal), UBXN7 (IF 1:500; HPA049442; Sigma-Aldrich), IFI16 (WB 1:1,000, IF 1:500; 14970; Cell Signaling Technology), V5-tag rabbit (WB 1:1,000, IF 1:1,000; 13202; Cell Signaling Technology), V5-tag mouse (WB 1:1,000, IF 1:400; R960-25; Invitrogen), HA-tag (WB 1:2,500, IF 1:100; 2367; Cell Signaling Technology), HA-tag-HRP (WB 1:1,000; H6533; Sigma-Aldrich), β-actin-HRP (WB 1:2,500; sc-47778; Santa Cruz), β-tubulin (WB 1:500; 2128; Cell Signaling Technology). For WB, secondary antibodies conjugated to HRP detecting rabbit IgG (1:2,500) and mouse IgG (1:5,000) were purchased from Dako and Sigma-Aldrich, respectively. For IF, DAPI (1:1,000) and secondary antibody detecting rabbit or mouse IgG conjugated to Alexa Fluor 488, Alexa Fluor 594 or Alexa Fluor 647 (1:200 to 1:500) were purchased from Invitrogen. GFP-DyLight-488 was purchased from Rockland (600-141-215; 1:1,000).

### Virus strains and virus stocks preparation
VZV rOka was a gift from J. Cohen[30] (National Institutes of Health).

The recombinant VZV used in this study was generated based on pP-Oka, an infectious BAC clone of the pOka strain[74] using two-step red-mediated mutagenesis as previously described[96,97]. The reporter RFP-tagged VZV variant was generated by insertion of the mRFP cassette into the pP-Oka BAC at the C terminus of both copies of the diploid VZV gene ORF63/70 encoding the immediate-early protein 63 (IE63). To detect the expression and localization of the VZV ORF61 protein, we fused an HA tag with a flexible glycine-serine linker to the C terminus of ORF61 to generate the VZV(pOka)-61–HA. Final clones were confirmed by restriction fragment length polymorphism analyses, PCR and DNA sequencing. Oligonucleotides used for the mutagenesis of ORF61 are given in Supplementary Table 8. The resulting BACs were transfected into MeWo cells, thereby generating the given recombinant VZVs.

VZVs were propagated in MeWo cells. Monolayers of infected cells were monitored microscopically for cytopathic effect and, when appropriate, mRFP expression. Cells that showed high levels of infection were detached and replated on seeded 70% confluent uninfected cells at appropriate ratios (1:2 to 1:4). Aliquots of a defined count of infected cells were cryopreserved in freezing medium (45% FCS, 10% DMSO), stored in liquid nitrogen and thawed for experiments.

VZV titre was determined by plaque assay: confluent monolayers of MeWo cells were infected for 3 h at 37 °C with serial 10-fold dilutions of 1 million freshly thawed VZV MeWo stock. At 3 h post-infection, the culture medium was changed and cells were kept in culture at 37 °C. For reporter fluorescence virus, RFP-fluorescent-infected foci were counted 2 days post-infection. For non-fluorescent virus, cells were fixed with 4% formaldehyde for 30 min at room temperature 4 days post-infection and stained with crystal violet (1% crystal violet, 10% ethanol) for 10 min at room temperature. Titres were defined by foci per plaque forming units per million cells.

### DNA transfection

If not explicitly described, DNA transfection was performed as follows: plasmids were mixed with polyethylenimine (24765; Polysciences) at a DNA:polyethylenimine ratio of 1:3 in Opti-MEM (Gibco Fisher Scientific) for 20 min at room temperature. The DNA:polyethylenimine mix was added to cells and media was exchanged 6 h post-transfection.

### Generation of lentivirus

The pLenti6.3 (expressing VZV ORFs or GFP), pWPI (expressing NPHP4 constructs or tagBFP) or pLVX (expressing shRNA) lentiviral expression plasmids, together with the packaging plasmids pCMV-Gag-Pol and pMD2-VSV-G, were transfected in HEK293T cells as described earlier. Viral supernatants were collected 48 h post-transfection, filtered on 0.45 µM PVDF membrane (Fisher Scientific) and stored at −80 °C. The pSicoR-based lentiviral expression plasmids (spCas9 or gRNAs) together with the packaging plasmids pCMVR8.91 and pMD2-VSV-G were mixed with TransIT-LT1 (Mirus Bio) in Opti-MEM (Gibco Fisher Scientific) for 20 min at room temperature. Supernatants were collected 72 h post-transfection and frozen at −80 °C. Lentiviruses were titred according to standard procedure.

### Cloning and cell line generation

**Generation of SpCas9 and reporter SK-N-BE2 cell lines.** SK-N-BE2 cells were transduced with the SpCas9-ZeoR cassette, allowing the expression of a human codon-optimized nuclear-localized *Streptococcus pyogenes cas9* gene in the absence of a U6 promoter–sgRNA and the zeocin resistance gene (*ZeoR*) using a lentivirus generated with the pSicoR-SpCas9-ZeoR[95] construct and polybrene at 8 µg ml⁻¹. To generate the pWPI-tagBFP-blastR, the tagBFP cassette was amplified by PCR from pSCRPSY-PAC2A-tagBFP and transferred into the lentiviral expression plasmid pWPI-blastR, allowing expression of blasticidin resistance, via BamHI-HF and AscI restriction digest (New England Biolabs). Oligonucleotides used for PCR are given in Supplementary Table 8. SK-N-BE2 wild-type or SK-N-BE2-spCas9(ZeoR) cells were transduced with GFP-blastR or tagBFP-blastR cassettes, respectively, using polybrene at 8 µg ml⁻¹. Cells were selected with 10 µg ml⁻¹ blasticidin for 10 days and maintained with 5 µg ml⁻¹ blasticidin to generate the SK-N-BE2-GFP(blastR) and the SK-N-BE2-spCas9(ZeoR)-tagBFP(blastR) cell lines. SK-N-BE2-spCas9(ZeoR)-tagBFP(blastR) cells were sorted for high BFP expression by flow cytometry (FACSAria III; BD Bioscience).

**Cloning the VZV ORF expression plasmid library and generation of VZV ORF-expressing SK-N-BE2 cell lines.** The generation of the library of VZV ORF expression plasmids has been described previously[14]. Briefly, individual VZV ORFs were amplified by PCR on reversed-transcribed viral RNA extracted from VZV rOka-infected cells, allowing the insertion of an upstream Kozak sequence (GCCGCC) and the removal of the stop codon for subsequent fusion with a C-terminal tag. Fragments were ligated into the pCR8/TOPO entry vector and shuttled via Gateway recombination into the lentiviral expression vector pLenti6.3-TO-V5-DEST-blastR, allowing the fusion of a C-terminal V5 tag and the expression of blasticidin resistance. pLenti6.3-TO-V5-GFP-blastR was used as control. SK-N-BE2 cells (70% confluent) were transduced with individual lentiviruses using

polybrene at 8 µg ml⁻¹. Cells were selected with 8 µg ml⁻¹ blasticidin and expanded for at least 10 days and collected for MS analysis or kept as frozen stock. VZV ORF2, ORF22, ORF27, ORF31, ORF42-45 and ORF62 are not included in this study. Oligonucleotides used for VZV ORF amplification from the viral genome are available in ref. 14.

**Cloning of the host gene gRNA expression plasmid library.** Two to four gRNA sequences per gene were designed using the CRISPOR online tool (http://crispor.tefor.net). Forward and reverse oligonucleotides, allowing the generation of BsmBI overhang, were purchased from IDT. Each complementary oligonucleotide (100 µM) was annealed in annealing buffer (1 mM EDTA, 50 mM NaCl in 10 mM Tris pH 8.0) by denaturation at 95 °C, followed by progressive cooling down to 5 °C. Annealed oligos were ligated into BsmBI.v2-digested (New England Biolabs) pSicoR-U6-gRNA-EFS-PuroR with T4 DNA ligase (Thermo Fisher Scientific). gRNA sequences are listed in Supplementary Table 8.

**Cloning of UBXN7 shRNA and generation of HFF knockdown cells.** Scramble (control) and UBXN7-targeting shRNA sequences were identified in the MISSION TRC library (Sigma-Aldrich). Forward and reverse oligonucleotides, allowing shRNA expression with overhangs for cloning into BamHI and EcoRI sites, were purchased from Eurofins. Each complementary oligonucleotide (100 µM) was annealed in annealing buffer (1 mM EDTA, 50 mM NaCl in 10 mM Tris pH 8.0) by denaturation at 95 °C, followed by progressive cooling down to 5 °C. Annealed oligonucleotides were ligated into BamHI-EcoRI-digested (New England Biolabs) pLVX with T4 DNA ligase (Thermo Fisher Scientific). Final constructs were confirmed by DNA sequencing. Oligonucleotides are available in Supplementary Table 8. HFF cells were transduced with the respective shRNA and selected with 1.5 µg ml⁻¹ puromycin after 2 days. Cells were used for experiments after another 3 days in culture.

**Generation of stable knockout SK-N-BE2 cell lines.** Stable knockout SK-N-BE2 cell lines for selected host genes were generated to replicate the effect on VZV spread observed in the knockout screen and to perform functional assays. SK-N-BE2-spCas9(ZeoR)-BFP(blastR) cells were transduced with the following host gene sgRNA, selected from the same library designed for the knockout screen, using polybrene at 8 µg ml⁻¹: pooled sgRNA targeting MPP8 or ZNF280D, an sgRNA targeting NPHP4 (AAGGCTGGCGCGCTCTCTGT) or an empty vector as NTC. After 2 days, cells were selected with 2 µg ml⁻¹ puromycin and kept in culture for another 12 days to allow efficient knockout and to increase the chances of clearing the remaining expressed protein. Efficient knockout was validated relative to the NTC by genotyping (the targeted regions were amplified by PCR, sequenced and analysed by Synthego ICE analysis[98], WB and MS analysis).

**Cloning of HA-tagged expression cassettes.** ORF61 and GFP cassettes were inserted into pDONR221 by PCR from pLenti6.3 expression plasmids generated as described earlier. The NPHP4, ORF61, ICP0 and GFP cassettes were shuttled from pENTR or pDONR into the lentiviral expression plasmid pWPI-nHA-PuroR-GW by Gateway recombination. pWPI-nHA-PuroR-NPHP4(S862N) encoding for the expression of the NPHP4 variant was generated by two-step PCR mutagenesis from the wild-type construct. In brief, the 5′ and 3′ fragments of NPHP4 were amplified by PCR using the Phusion Hot Start II DNA Polymerase (Thermo Fisher Scientific) and the primers A and B for 5′ fragments and C and D for 3′ fragments. Primers were designed to allow for the incorporation of the c.2585G>A mutation, complementarity within their respective 3′ and 5′ ends, and the insertion of the NdeI restriction site before the start codon and the SpeI site after the stop codon. The complete gene cassette was then amplified by a second PCR using the two fragments as templates and the primers A and D. The resulting product was then shuttled into pWPI-nHA-PuroR using NdeI and

SpeI-HF restriction digests (New England Biolabs), followed by T4 DNA ligation (Thermo Fisher Scientific).

Final constructs were confirmed by DNA sequencing.

## Sample preparation and analysis of MS-based proteomic and transcriptomic experiments

**Host proteome changes induced by VZV infection of SK-N-BE2 cells.** As VZV is a highly cell-associated pathogen that releases few infectious particles into the extracellular media[99,100], SK-N-BE2 cells were infected by co-culture with VZV rOka-infected MeWo inoculum cells using a transwell system to prevent contamination with inoculum cells[31]. A ratio of 2:1 or 5:1 of uninfected:infected MeWo cells, depending on the virus lot, was seeded on 1 side of porous transwells (insert 6-well, PET, 1 μm pores; Sarstedt) in culture medium. After 24 h, the transwell was turned upside down into a six-well plate filled with culture medium, and SK-N-BE2 cells were seeded on the second side of the transwell in culture medium. At 48 h post-infection, SK-N-BE2 cells were scraped from the transwell membrane, washed in ice-cold PBS, lysed in SDS lysis buffer (50 mM TRIS-HCl pH 7.6, 10 mM DTT, 4% SDS), boiled for 5 min at 95 °C, flash frozen and kept at −80 °C. Frozen pellets were thawed, sonicated (5 min, 4 °C, 30 s on and off, high frequency; Bioruptor; Diagenode), boiled for 5 min at 95 °C and alkylated with 55 mM iodoacetamide for 20 min at room temperature. Proteins were precipitated and cleaned from SDS by adding 4 volumes of acetone and incubating for 2 h at −20 °C. The pellet was resolubilized in denaturation buffer (6 M urea, 2 M thiourea, 10 mM HEPES pH 8.0) and frozen at −20 °C. Total proteins (50 μg) were thawed and predigested with 1 μg LysC (Wako Chemicals) at room temperature for 4 h, followed by a 1:5 dilution in ABC buffer (50 mM $NH_4HCO_3$, 100 mM Tris-HCl pH 8) and digested for 15 h at 30 °C with 1 μg trypsin (Sigma-Aldrich). The digest was stopped and the peptides were solubilized by the addition of 0.6% trifluoroacetic acid (TFA) and 2% acetonitrile (ACN). Samples were spin-centrifuged, and the cleared peptide supernatant was transferred into new tubes before peptide purification. Peptides were purified on StageTips with 3 layers of C18 Empore filter discs (3M). MS analysis was performed on an EASY-nLC 1200 system (Thermo Fisher Scientific), coupled with the mass spectrometer (Q Exactive HF-X; Thermo Fisher Scientific) via a nano-electrospray source as previously described[25]. Briefly, peptides were eluted on a 50 cm reverse-phase analytical column (75 μm diameter; ReproSil-Pur C18-AQ 1.9 μm resin; Dr. Maisch) using a gradient of ACN in 0.1% formic acid at a flow rate of 300 nl min⁻¹ (sequential linear gradients of 80% ACN: 5–30% for 95 min, 30–60% for 5 min, 60–95% for 5 min, followed by a stationary step at 95% for 5 min to elute the most hydrophobic peptides and re-equilibration of the column at 5%). To avoid carryover of remaining peptides across conditions, the column was washed with 95% of 80% ACN for 15 min between quadruplicates. The mass spectrometer was operated and MS spectra acquired using the XCalibur software (Thermo Fisher Scientific) with data-dependent acquisition (DDA) mode. Full MS scans (300–1,650 $m/z$, resolution (R) = 60,000) were acquired at an ion target of $3 \times 10^6$. The top 15 most abundant precursor peptides were fragmented by higher-energy collisional dissociation (HCD) with a normalized collision energy of 27% and MS/MS scan (R = 15,000) acquired at an ion target of $1 \times 10^5$ and a maximum injection time of 25 ms. Isolation and fragmentation of the same peptide precursor were eliminated by dynamic exclusion for 20 s. The whole process (infection, sample preparation and MS measurement) was repeated twice, each with four independent experiments.

**Host proteome changes and AP of SK-N-BE2 cells expressing V5-tagged VZV proteins.** SK-N-BE2 cells transduced with individual expression cassettes for VZV ORF or GFP fused to the C-terminal V5 tag were expanded in quadruplicates to reach 2 confluent 15 cm dishes per replicate. Control cell lines (GFP, ORF60 and ORF66) were expanded to reach 16 replicates. Cells were gently washed in ice-cold PBS, scraped,

pooled per replicate and washed twice in ice-cold PBS by centrifugation at 600g at 10 °C for 10 min. Before the last wash, an aliquot of $1 \times 10^6$ cells from each replicate was kept for full proteome analysis. All samples were flashed frozen in liquid nitrogen before storage at −80 °C.

*AP of V5-tagged VZV ORF.* Samples were processed in 3 immunoprecipitation (IP) batches of 19 VZV ORFs. To account for batch effect, the three controls (GFP, ORF60 and ORF66) were included within each batch. Frozen cell pellets were thawed and lysed on ice for 30 min in lysis buffer (0.2% NP-40, 100 mM NaCl, 5% glycine, 1.5 mM $MgCl_2$, 50 mM Tris-HCl pH 7.5) supplemented with 1% in-house benzonase and EDTA-free Complete Protease Inhibitor (Roche). Samples were sonicated (5 min, 4 °C, 30 s on and off, high frequency; Bioruptor; Diagenode) and centrifuged at 15,000g at 4 °C for 30 min. Supernatants were collected in 96-well deep-well plates with randomized positions to minimize plate position effects. Total protein concentrations were measured by Pierce 660 nm Protein Assay (Thermo Fisher Scientific) and normalized to 6 mg in 750 μl lysis buffer supplemented with EDTA-free Complete Protease Inhibitor (Roche) during transfer into 4 24-well deep-well plates. Cleared lysates were mixed with 30 μl of anti-V5 magnetic bead slurry (MBL M215-11), previously equilibrated in lysis buffer, and agitated for 2 h at 4 °C. After incubation, the samples were transferred into a 96-well deep-well plate. To favour intra- and inter-batch reproducibility, IPs were automatized on a Freedom EVO 200 robotic platform (Tecan) equipped with an 8-needle liquid handling station, a plate magnet position and a plate shaker. Immune complexes attached to the magnetic beads were allowed to collect on the magnet for 5 min. The flow-throughs were aspirated, the plate was moved to the shaker position, 480 μl of lysis buffer was added for washing and the plate was agitated at 1,200 rpm for 2 min. The wash was repeated six times in lysis buffer to reduce unspecific binding, followed by eight additional washes in wash buffer (lysis buffer without NP-40) to eliminate remaining detergents. Excess buffer was removed while the plate was positioned on the magnet. Beads were resuspended in 20 μl of 1:10 diluted guanidinium chloride buffer (0.6 M GdmCl, 1 mM tris (2-carboxyethyl) phosphine (TCEP), 4 mM chloroacetamide (CAA) in 0.1 M Tris-HCl pH 8.0) to allow for denaturation, reduction, and alkylation of the enriched proteins, and then transferred into a 96-well microplate and frozen at −20 °C. To prevent further batch effects, protein digest and peptide purification of the three IP batches were processed simultaneously. Sample plates were thawed at room temperature and predigested with 1 μg LysC (Wako Chemicals) for 4 h at 37 °C, followed by a 1:5 dilution in 0.1 M Tris-HCl pH 8.0 and digestion for an additional 15 h at 30 °C with 1 μg trypsin (Sigma-Aldrich). The digestion was stopped and peptides solubilized in 0.6% TFA and 2% ACN. Beads were sedimented by using the plate magnet, peptides were transferred onto a new 96-well microplate and frozen at −20 °C. Peptides were purified on StageTips with 3 layers of C18 Empore filter discs (3M). MS analysis was performed on an EASY-nLC 1200 system (Thermo Fisher Scientific), coupled online with the mass spectrometer (Q Exactive HF-X; Thermo Fisher Scientific) via a nano-electrospray source as previously described[25]. Briefly, peptides were eluted on a 20 cm reverse-phase analytical column (75 μm diameter; ReproSil-Pur C18-AQ 1.9 μm resin; Dr. Maisch) using a gradient of ACN in 0.1% formic acid at a flow rate of 300 nl min⁻¹ (sequential linear gradients of 80% ACN: 5–30% for 85 min, 30–60% for 12 min, 60–80% for 3 min, and 80–95% for 1 min, followed by a stationary step at 95% for 5 min to elute the most hydrophobic peptides and re-equilibration of the column at 5%). As samples were measured in a randomized order, the column was washed with 95% of 80% ACN for 15 min after each run to avoid carryover of peptides between samples. The mass spectrometer was operated and MS spectra acquired using the XCalibur software (Thermo Fisher Scientific) with DDA mode. Full MS scans (300–1,650 $m/z$, R = 60,000) were acquired at an ion target of $3 \times 10^6$. The top 15 most abundant precursor peptides were fragmented by HCD with a normalized collision energy of 27% and MS/MS scan (R = 15,000)

acquired at an ion target of $1 \times 10^5$ and a maximum injection time of 25 ms. Isolation and fragmentation of the same peptide precursor were eliminated by dynamic exclusion for 20 s.

*Full proteome.* Frozen pellets of $1 \times 10^6$ cells were thawed on ice and lysed in guanidinium chloride lysis buffer (6 M GdmCl, 10 mM TCEP, 40 mM CAA in 0.1 M Tris-HCl pH 8.0) for 30 min. Samples were boiled at 99 °C with shaking at 500 rpm for 15 min and then sonicated (5 min, 4 °C, 30 s on and off, high frequency; Bioruptor; Diagenode). Supernatants were collected after centrifugation at 15,000$g$ at 4 °C for 30 min. Total proteins (50 μg) were predigested with 1 μg LysC (Wako Chemicals) for 3 h at 37 °C, followed by a 1:5 dilution in 0.1 M Tris-HCl pH 8.0 and digestion for an additional 15 h at 30 °C with 1 μg trypsin (Sigma-Aldrich). The digest was stopped and the peptides solubilized by the addition of 0.6% TFA and 2% ACN. Samples were spin-centrifuged, and the cleared peptide supernatant was transferred into new tubes before peptide purification. Peptides were purified on StageTips with three layers of C18 Empore filter discs (3M). MS analysis was performed on an EASY-nLC 1200 system (Thermo Fisher Scientific), coupled online with the mass spectrometer (Q Exactive HF-X; Thermo Fisher Scientific) via a nano-electrospray source as previously described[25]. Briefly, peptides were eluted on a 50 cm reverse-phase analytical column (75 μm diameter; ReproSil-Pur C18-AQ 1.9 μm resin; Dr. Maisch) using a gradient of ACN in 0.1% formic acid at a flow rate of 300 nl min$^{-1}$ (sequential linear gradients of 80% ACN: 5–30% for 95 min, 30–60% for 5 min, 60–95% for 5 min, followed by a stationary step at 95% for 5 min to elute the most hydrophobic peptides and re-equilibration of the column at 5%). To avoid carryover of peptides across conditions, the column was washed with 95% of 80% ACN for 15 min between quadruplicates. The mass spectrometer was operated and MS spectra acquired using the XCalibur software (Thermo Fisher Scientific) with DDA mode. Full MS scans (300–1,650 $m/z$, R = 60,000) were acquired at an ion target of $3 \times 10^6$. The top 15 most abundant precursor peptides were fragmented by HCD with a normalized collision energy of 27% and MS/MS scan (R = 15,000) acquired at an ion target of $1 \times 10^5$ and a maximum injection time of 25 ms. Isolation and fragmentation of the same peptide precursor were eliminated by dynamic exclusion for 20 s.

**Proteome changes induced by MPP8 gene depletion in SK-N-BE2 cells.** One million MPP8-knockout or NTC SK-N-BE2-spCas9(ZeoR)-BFP(blastR) cells were collected in triplicate, and the pellets were flash frozen in liquid nitrogen for subsequent MS analysis of the full proteome. Frozen cell pellets were thawed on ice and lysed in guanidinium chloride lysis buffer (6 M GdmCl, 10 mM TCEP, 40 mM CAA in 0.1 M Tris-HCl pH 8.0) for 30 min. Samples were boiled at 99 °C with shaking at 500 rpm for 15 min and then sonicated (5 min, 4 °C, 30 s on and off, high frequency; Bioruptor; Diagenode). Supernatants were collected after centrifugation at 15,000$g$ at 4 °C for 30 min. Total proteins (50 μg) were predigested with 1 μg LysC (Wako Chemicals) for 3 h at 37 °C, followed by a 1:5 dilution in 0.1 M Tris-HCl pH 8.0 and digestion for an additional 15 h at 30 °C with 1 μg trypsin (Sigma-Aldrich). The digest was stopped and the peptides solubilized by the addition of 0.6% TFA and 2% ACN. Samples were spin-centrifuged, and the cleared peptide supernatant was transferred into new tubes before peptide purification. Peptides were purified on StageTips with three layers of C18 Empore filter discs (3M). MS analysis was performed on an EASY-nLC 1200 system (Thermo Fisher Scientific), directly coupled online with the mass spectrometer (Q Exactive HF-X; Thermo Fisher Scientific) via a nano-electrospray source as previously described[25]. Briefly, peptides were eluted on a 50 cm reverse-phase analytical column (75 μm diameter; ReproSil-Pur C18-AQ 1.9 μm resin; Dr. Maisch) using a gradient of ACN in 0.1% formic acid at a flow rate of 300 nl min$^{-1}$ (sequential linear gradients of 80% ACN: 5–30% for 150 min, 30–60% for 5 min, 60–95% for 5 min, followed by a stationary step at 95% for 5 min to elute the most hydrophobic peptides and re-equilibration of the column at 5%). To

avoid carryover of remaining peptides across conditions, the column was washed with 95% of 80% ACN for 15 min between quadruplicates. The mass spectrometer was operated and MS spectra acquired using the XCalibur software (Thermo Fisher Scientific) with DDA mode. Full MS scans (300–1,650 $m/z$, R = 120,000) were acquired at an ion target of $3 \times 10^6$. The top 15 most abundant precursor peptides were fragmented by HCD with a normalized collision energy of 27% and MS/MS scan (R = 15,000) acquired at an ion target of $1 \times 10^5$ and a maximum injection time of 25 ms. Isolation and fragmentation of the same peptide precursor were eliminated by dynamic exclusion for 20 s.

**Transcriptome changes induced by NPHP4 gene depletion in VZV-infected SK-N-BE2 cells.** *Sample preparation and sequencing.* SK-N-BE2 cells, control or depleted for NPHP4, were infected by co-culture with MeWo cells infected with VZV(pOka)-63-RFP/70-RFP using a transwell system to prevent contamination with inoculum cells[31], as described earlier, with a ratio of 3:1 of uninfected:infected MeWo. At 48 h post-infection, SK-N-BE2 cells were scraped from the transwell membrane, washed in ice-cold PBS, lysed in LBP buffer (Macherey-Nagel), flash frozen and kept at −80 °C. RNA was extracted according to the supplier's recommendation (Macherey-Nagel). Bulk sequencing of poly(A)-RNA was done as previously described[101]. Barcoded cDNA of each sample was generated using a Maxima RT polymerase (Thermo Fisher) using oligo(dT) primer containing barcodes, unique molecular identifiers (UMIs) and an adaptor. 5′ ends of the cDNAs were extended by a template switch oligonucleotide, and full-length cDNA was amplified with primers binding to the template switch oligonucleotide site and the adaptor. The NEB UltraII FS kit was used to fragment cDNA. After end repair and A-tailing, a TruSeq adaptor was ligated and 3′ end fragments were amplified using primers with Illumina P5 and P7 overhangs. The library was sequenced on a NextSeq 500 (Illumina) with 61 cycles for the cDNA in read 1 and 19 cycles for the barcodes and UMIs in read 2.

**AP of SK-N-BE2 cells expressing HA-tagged NPHP4 wild type or S862N variant.** SK-N-BE2 cells transduced with NPHP4 wild type, the S862M mutant or GFP fused to the N-terminal HA tag were expanded in quadruplicates to reach 2 confluent 15 cm dishes per replicate. Cells were gently washed in ice-cold PBS, scraped, pooled per replicate and washed twice in ice-cold PBS by centrifugation at 600$g$ at 10 °C for 10 min. All samples were flash frozen in liquid nitrogen before storage at −80 °C. Frozen cell pellets were thawed and lysed on ice for 30 min in 1 ml lysis buffer (0.2% NP-40, 100 mM NaCl, 5% glycine, 1.5 mM MgCl$_2$, 50 mM Tris-HCl pH 7.5) supplemented with 1% in-house benzonase and EDTA-free Complete Protease Inhibitor (Roche). Samples were sonicated (5 min, 4 °C, 30 s on and off, high frequency; Bioruptor; Diagenode) and centrifuged at 15,000$g$ at 4 °C for 30 min. Total protein concentrations were measured by Pierce 660 nm Protein Assay (Thermo Fisher Scientific) and normalized to 2 mg in 1 ml lysis buffer supplemented with EDTA-free Complete Protease Inhibitor (Roche). Cleared lysates were mixed with 40 μl of anti-HA agarose bead slurry (A2095; Sigma-Aldrich), previously equilibrated in lysis buffer, and agitated for 3 h at 4 °C. Immune complexes attached to the beads were washed 5× with 1 ml lysis buffer and 4× in 1 ml wash buffer (lysis buffer without NP-40) to eliminate remaining detergents. Excess buffer was removed and beads were resuspended in 20 μl of 1:10 diluted guanidinium chloride buffer (0.6 M GdmCl, 1 mM TCEP, 4 mM CAA in 0.1 M Tris-HCl pH 8.0) to allow denaturation, reduction, and alkylation of the enriched proteins, and then frozen at −20 °C. Samples were thawed at room temperature and predigested with 0.5 μg LysC (Wako Chemicals) for 3 h at 37 °C, followed by a 1:4 dilution in 0.1 M Tris-HCl pH 8.0 and digestion for an additional 16 h at 30 °C with 0.5 μg trypsin (Sequencing Grade; Promega). The digest was stopped and peptides solubilized in 0.6% TFA and 2% ACN. Beads were sedimented by spin centrifugation for 5 min at 10,000 rpm, peptides were transferred into new tubes, processed by StageTip purification with 3 layers of C18 Empore

filter discs (3M) and resuspended in 2% ACN, 0.1% TFA. MS analysis was performed on an EASY-nLC 1200 system (Thermo Fisher Scientific), coupled online with the mass spectrometer (Q Exactive HF-X; Thermo Fisher Scientific) via a nano-electrospray source as previously described[25]. Briefly, peptides were eluted on a 20 cm reverse-phase analytical column (75 μm diameter; ReproSil-Pur C18-AQ 1.9 μm resin; Dr. Maisch) using a gradient of ACN in 0.1% formic acid at a flow rate of 300 nl min$^{-1}$ (sequential linear gradients of 80% ACN: 5–30% for 37 min, 30–60% for 6 min, 60–80% for 3 min, and 80–95% for 1 min, followed by a stationary step at 95% for 5 min to elute the most hydrophobic peptides and re-equilibration of the column at 5%). The column was washed with 95% of 80% ACN for 15 min between each quadruplicate. The mass spectrometer was operated and MS spectra acquired using the XCalibur software (Thermo Fisher Scientific) with DDA mode. Full MS scans (300–1,650 $m/z$, R = 60,000) were acquired at an ion target of $3 \times 10^6$. The top 15 most abundant precursor peptides were fragmented by HCD with a normalized collision energy of 27% and MS/MS scan (R = 15,000) acquired at an ion target of $1 \times 10^5$ and a maximum injection time of 25 ms. Isolation and fragmentation of the same peptide precursor were eliminated by dynamic exclusion for 20 s.

### Data processing and bioinformatic analysis

**Host full proteome changes induced by VZV infection of SK-N-BE2 cells.** *Data processing.* The raw MS data files were analysed with Max-Quant (v.1.6.0.15) using the standard settings and label-free quantification (LFQ) with 'match between runs' enabled ('LFQ min ratio count' set to 2 and 'stabilization of large LFQ ratios' enabled). The MS2 spectra were searched against forward and reverse (used as decoy) sequences of the reviewed human proteome, including isoforms (UniprotKB, release 2018.02) and VZV proteins (pOka; UniprotKB, release 2017.12), using the built-in Andromeda search engine.

*Statistical analysis.* LFQ intensities of protein groups were imported from MaxQuant (proteinGroup file) into Perseus (v.1.6.5.0) for statistical analysis of infected proteome changes. Data from two repeats, each including independent quadruplicate mock and infected samples, were analysed. Contaminants, reverse and only-by-site identifications were removed, as were protein groups not quantified in at least three of four samples in at least one condition. Missing intensities were imputed by random sampling from a normal distribution $N(I_0 - 1.8\sigma, (0.3\sigma)^2)$, where $I_0$ and $\sigma$ are the mean and s.d. of total measured log$_2$ intensities, respectively. Log-transformed fold changes were calculated as the log$_2$ difference of the median intensity within infected samples over mock samples for each repeat. The significance was assessed by the Student's *t*-test with permutation-based multiple hypothesis testing correction. Host proteins with absolute log$_2$ difference ≥ 0.5 and adjusted $P \leq 0.05$ in both repeats were reported as strong hits (the direction of the change in both sets also had to match). Gene set enrichment analysis was performed via one-sided Fisher's exact tests on gene ontologies from the GO collections, adjusted by Benjamini–Hochberg false discovery rate (FDR). Terms enriched with an enrichment factor ≥4 and an adjusted $P \leq 0.05$ were considered to be significant. The least-enriched terms that were redundant between GO collections were excluded.

**AP of V5-tagged VZV proteins in SK-N-BE2 cells (interactomes).** *Data processing.* The raw MS data files of the AP–MS experiments, full proteome of SK-N-BE2 cells and six fractions from deep coverage of the SK-N-BE2 cell proteome were analysed with MaxQuant (v.1.6.0.15) using the standard settings and fast LFQ enabled ('LFQ min ratio count' set to 2, 'stabilize large LFQ ratios' disabled and normalization skipped). The AP–MS and fractionated SK-N-BE2 deep proteomic samples were assigned to different parameter groups with match between runs enabled. The MS2 spectra were searched against forward and reverse (used as decoy) sequences of the reviewed human proteome, including isoforms (UniprotKB, release 2018.02), VZV proteins (pOka; UniprotKB, release 2017.12), the GFP protein and the V5-tag sequence.

*Statistical analysis.* R (v.3.6), Julia (v.1.5) and Python (v.3.8) using a collection of in-house scripts[102] were used in this analysis.

MaxQuant output files were imported into R using the in-house msimportr R package[103]. A Bayesian linear mixed effects model, implemented in the msglm R package[43] was used to estimate the enrichment of protein groups in the AP–MS experiments with viral baits. In R generalized linear model (GLM) formula language, the model could be specified as

$$\log \text{Intensity} \sim 1 + \text{APMS} + \text{APMS} : \text{bait}_i + \text{MSbatch}_i$$
$$+ \text{BioChemBatch}_i + \text{PCAbatch}_i,$$

where intensity is the LFQ intensity of a given protein group in a given sample, the APMS effect corresponds to the average shift of protein group intensity (enrichment) in AP–MS data in comparison to the full SK-N-BE2 proteome, and the interaction effect APMS:bait$_i$ models the enrichment of a protein group specific to the viral bait of the *i*th MS replicate; the effects MSbatch$_i$ and BioChemBatch$_i$ account for the protein intensity variations specific to the MS measurement and the AP batches of the *i*th sample, respectively. An additional continuous batch effect, PCAbatch$_i$, corresponds to the specific protein contamination pattern recurring across the AP samples with varying intensity. The pattern was associated with the second principal component of the protein × sample log-intensities matrix, and we used the following formula to define the intensity of the contamination in the *i*th sample:

$$\text{PCAbatch}_i = \frac{w_i - \text{median}(w)}{\max(w) - \text{median}(w)}$$

if $w_i > \text{median}(w) + \max(w)$, and 0 otherwise, where $w_i$ is the weight of the second principal component in the *i*th sample.

Due to the inherent sparsity of AP data, the effects of the model associated with experimental conditions had horseshoe priors[104]. The MSGLM model assumes that the measurement error of the MS instrument follows a Laplace distribution, and the parameters of this distribution depend on the signal intensity (heteroscedastic intensities noise model). The MSGLM noise model was calibrated with technical replicates of our MS instrument. The same MS data were also used to calibrate the missing data model that defines the probability of protein identification as a logit-transformed expected abundance. The model was applied to unnormalized LFQ intensities with per-MS-run normalization multipliers, matching the predictions to the expected intensity of a given MS run. This scheme allows the model likelihood to automatically account for the quality of each individual sample. The MSGLM model was applied to each protein group separately. To infer the posterior distribution of model parameters, 4,000 iterations (2,000 warm-up and 2,000 sampling iterations) of the no-U-turn Markov chain Monte Carlo method, implemented using the rstan package[105] (v.2.19), were run in 8 independent chains, with every 4th collected sample.

The modelling of MS data provided the enrichment estimates for each protein in each AP experiment cleared from the batch effects. The specific AP–MS interactions between a viral bait and a protein had to be significantly enriched relative to the background distribution of that protein, calculated from its abundance in all the other baits, with the 25% least abundant and 10% most abundant viral baits removed. The difference between the medians of the bait-specific and background posterior distribution of protein log$_2$ intensity had to be ≥1 and with $P \leq 0.01$. The $P$ value was calculated as the probability that a random sample from the posterior distribution of the protein abundance in the viral bait AP–MS experiment would be smaller than a random sample drawn from the background distribution. No $P$ value adjustment was done as multiple-hypothesis correction is

handled by the choice of Bayesian priors. Proteins with the median estimated effect size of the PCAbatch term above 2.5 and $P \leq 10^{-5}$ were annotated as 'putative contaminants'. Proteins identified by only one peptide needed to be detected in at least three of the four independent experiments, and identified in at least two experiments by MS/MS. Multiple isoforms identified within a single VZV bait interactome were combined under the canonical prey name, keeping the enrichment and $P$ value of the interaction quantified with the highest number of peptides. To exclude possible contamination due to bait overexpression effects on the background proteome, host preys detected as upregulated in the effectome data for the given bait (see below), with a median $\log_2$ difference smaller than four times the AP–MS enrichment, were filtered out.

## Host full proteome changes induced the expression of V5-tagged VZV proteins in SK-N-BE2 cells (effectomes). *Data processing*. The raw MS data files of the effectome experiments were analysed with MaxQuant (v.1.6.14.0) using the standard settings and fast LFQ with match between runs (LFQ min ratio count 2, classic normalization and stabilize large LFQ ratios enabled). The MS2 spectra were searched against forward and reverse (used as decoy) sequences of the reviewed human proteome (UniprotKB, release 2019.12), VZV proteins (pOka; UniprotKB, release 2017.12), the GFP protein and the V5 tag independently using the built-in Andromeda search engine.

*Statistical analysis.* Protein groups and LFQ intensities generated by MaxQuant (proteinGroups file) were imported into R Studio (v.2022.07.2+576). MS runs that did not reach a satisfactory performance were excluded. This led to the removal of the data for 18 VZV ORFs. Runs were grouped per batch corresponding to MS measurement sessions over the course of time. VZV ORF33.5 runs, being the only satisfactory runs in their batch, were associated with another batch with a similar background. Protein groups were filtered for contaminants and reverse identifications. Missing intensities were imputed from a log normal distribution of log-transformed values with a width of 0.3× the s.d. of the measured intensities and downshifted by 1.8 s.d. to simulate the detection limit of the mass spectrometer. VZV ORF-specific effects were calculated as the $\log_2$ difference of the median intensity within the proteome of the given bait and its background. The background was defined as the median intensity within all other ORFs of the batch with the 10% least abundant and 10% most abundant ORFs removed. Significance was assessed using the Wilcoxon rank-sum test, adjusted by the Benjamini–Hochberg approach. Absolute $\log_2$ difference ≥ 0.5 and adjusted $P \leq 0.05$ were used to define a significant effect. An absolute $\log_2$ difference ≥ 1 was used to define 'strong' significant effects, which were used as input for the network diffusion analysis. Finally, 'high confident' significant effects are defined as an adjusted $P \leq 0.01$ and an identification by MS/MS in 3 of 4 of the replicates within the group showing higher abundance (that is, within the VZV ORF replicates if upregulated and within the background if downregulated) on proteins identified by at least 3 peptides.

## Systematic subcellular localization analysis of the interactome preys. Host preys identified in the VZV–host interactome datasets were annotated using the GO Cellular Component terms, curated UniProt subcellular locations and both main and additional locations from the Protein Atlas. Each prey's localizations were summarized within the following categories using a keyword search: nucleus ('nucl'), cytosol ('cytosol'), Golgi apparatus ('golgi'), endoplasmic reticulum ('reticulum'), mitochondria ('mitochon'), cytoskeleton ('actin', 'microtubule' and 'cytoskelet') and cell membrane ('plasma' and 'cell membrane').

## Gene set enrichment analysis of the V5-tagged VZV proteins interactome and effectome. We used EnrichmentMap gene sets of human proteins (v.2020.10)[106]. For interactomes analysis, we used GO

(v.2020.10). For effectome analysis, we used GO (v.2021.12) and protein complex annotations from IntAct Complex Portal (v.2019.12)[107] and CORUM (v.2019)[108] and transcriptional interactions from OmniPath (v.2021.06)[109].

To find the non-redundant collection of annotations describing the unique and shared features of multiple experiments in a dataset, we used the in-house Julia package OptEnrichedSetCover[110], which uses an evolutionary multi-objective optimization technique to find a collection of annotation terms that have both significant enrichments in the individual experiments and minimal pairwise overlaps. The resulting set of terms was further filtered by requiring that the annotation term has to be significant with the specified unadjusted Fisher's exact test $P$ value cut-off in at least one of the experiments or comparisons.

For interactomes, GO Cellular Component terms with $P \leq 0.0001$ were retained and only 1 of the redundant terms (same intersect_genes within ORF) was retained, with priority given to that with the lowest $P$ value (Fig. 2b). The analysis of the effectomes was performed on all significant effects (absolute $\log_2$ difference ≥ 0.5 and adjusted $P \leq 0.05$). Terms from GO Biological Processes, protein complexes and transcriptional interactions enriched with $P \leq 0.001$ were kept. The terms enriched within the two ORF66 batch controls were merged. For clarity of the representative figure, only one of the redundant terms (same intersect_genes within ORF) was kept, with priority given to the term enriched in other ORF (Fig. 3b).

## Integration of the interactome and effectome data by network diffusion analysis. To integrate the AP–MS viral–host protein interactions and the virus protein-induced protein changes, we used the HierarchicalHotNet-based method[71], as we have previously done for the severe acute respiratory syndrome coronavirus 2 (ref. 25), with a few modifications as described later. The HierarchicalHotNet. jl v.1.1 Julia Package[111] and in-house script[102] were used for network diffusion and the following analysis of predictions of statistical significance. The analysis was based on the ReactomeFI network of gene functional interactions (v.2019)[112]. Instead of directly using the weighted graph of gene–gene random walk transitions, we used the results of this random walk to weight the edges of the ReactomeFI network:

$$\hat{S} = (1 - r) S \times (W \times w) + r(w),$$

where $r$ is the random walk restart probability, $w_i$ is the probability to restart the random walk from the $i$th gene, $S$ is the weighted adjacency matrix of the gene functional interactions (ReactomeFI) and $W$ is the matrix of random walk transition probabilities. The proteins with significant abundance changes upon bait overexpression ($|\text{median}(\log_2(\text{fold change}))| \geq 1.0$, Benjamini–Hochberg adjusted $P \leq 0.05$) (strong significant effects) were used as the sources of signal diffusion, with node weights set to $w_i = w(n_i) = \sqrt{|\text{median}(\log_2(\text{fold change}))| \times |\log_{10}P|}$, otherwise the node weight was set to zero. The weight of the edge $g_i \to g_j$, between a pair of genes $g$, was set to $w_{ij} = 1 + w(n_j)$ if $|\text{median}(\log_2(\text{fold change}))| \geq 0.25$, unadjusted $P \leq 0.001$, otherwise $w_{ij} = 1$. The random walk restart probability was set to 0.4 as previously described[25]. To define the optimal subnetwork, we applied the same random permutation-based procedure as in ref. 25, which searches for the optimal edge weight threshold $t_*$, which maximizes the weighted difference in the average (avg) inverse path length ($L_{\text{avg}}^{-1}(t)$) between the real data and permuted (perm) data-based analysis:

$$\Delta(t) = \frac{1}{1 + \left(\frac{N_s(t) - 1{,}000}{5{,}000}\right)^2} \left(L_{\text{avg,}real}^{-1}(t) - L_{\text{avg,}perm}^{-1}(t)\right),$$

where $N_s(t)$ is the number of nodes in the 5 largest components of the subnetwork with the minimal edge weight threshold $t$.

To assess the significance of the edge presence in the resulting network, we calculated the edge $P$ value as the probability that its weight in the permuted-data-based analysis would be higher than in the one based on real data:

$$P(w_{\mathrm{real}}(g_i, g_j) \le w_{\mathrm{perm}}(g_i, g_j)).$$

This $P$ value was stored as the 'prob_perm_walkweight_greater' edge attribute. The specific subnetworks predicted by the network diffusion were filtered for edges with $P \le 0.05$. Effects that are part of the functional subnetwork are shown as big squares, interactors that are connected to these effects are shown as circles, and intermediate proteins that are functionally connected but not affected at their expression level or identified as an interactor are shown as small squares. The networks were exported to the GraphML format and prepared for publication using yEd software (v.3.20; https://www.yworks.com). The catalogue of the networks for each viral bait is available as Supplementary Data 1.

**Assembling public interactions.** To compare the VZV AP–MS interactome with the known interactomes of VZV and HSV-1, we assembled the known viral interactions from BioGRID, IntAct, VirHostNet (v.2020.12; see Supplementary Table 7 for the taxonomy and molecular interactions identifiers considered) and interactions manually extracted from publications[15,113]. We mapped the herpesvirus proteins to the homologous VZV proteins according to gene and locus names (Supplementary Table 2).

**Full proteome changes induced by the depletion of the MPP8 gene in SK-N-BE2 cells.** *Data processing.* The raw MS data files were analysed with MaxQuant (v.1.6.14.0) using the standard settings and LFQ with match between runs enabled (LFQ min ratio count 2 and stabilize large LFQ ratios enabled). The MS2 spectra were searched against forward and reverse (used as decoy) sequences of the reviewed human proteome (UniProtKB, release 2019_12) by the built-in Andromeda search engine.

*Statistical analysis.* R (v.4.1) and Julia (v.1.6) were used.

The MaxQuant output files were imported into R using the in-house msimportr R package[103]. A Bayesian linear random effects model, implemented in the msglm R package[114], was used to estimate the change in abundance of protein groups in the knockout experiment. In R GLM formula language, the model could be specified as:

$$\log \mathrm{Intensity} \sim 1 + \mathrm{KO},$$

where intensity is the intensity of a given protein group and KO models the average shift of intensity in the knockout samples in comparison to the control samples.

The peptide-based model was applied to unnormalized MS1 peak intensities of each protein group (evidence.txt table of MaxQuant output), using MS-run-specific normalization multipliers to scale the inferred abundance. The KO effect has regularized horseshoe+ priors[115]. As described in 'AP of V5-tagged VZV proteins in SK-N-BE2 cells (interactomes)'–'Statistical analysis', we assumed a measuring error of the instrument and a logit-based probability of missing MS data. To infer the posterior distribution of the model parameters, 4,000 iterations (2,000 warm-up and 2,000 sampling iterations) of the no-U-turn Markov chain Monte Carlo method, implemented in the cmstanr package (v.0.4.0), were run in 8 independent chains, with every 4th sample collected. The $P$ value was calculated as the probability that on average replicates from the posterior distribution of the knockout condition are different from the control condition. There was no correction for multiple hypothesis testing as this was resolved via the choice of model priors.

A protein group was detected as significantly regulated if (1) the difference between the medians of the posterior distributions of the log₂ intensity in the knockout condition and the control condition was ≥0.5 with $P \le 0.001$, and (2) the protein group was identified with a minimum of 2 peptides detected with MS/MS and at least with 1 peptide identified in 2 of the 3 replicates.

**Host transcriptome changes induced by NPHP4 gene depletion in VZV-infected SK-N-BE2 cells.** *Data processing.* Data were processed using the published Drop-seq pipeline (v.1.0) to generate sample- and gene-wise UMI tables[116]. The reference genome (GRCh38) was used for alignment. Transcript and gene definitions were used according to GENCODE v.38. The VZV genome was derived from GenBank: NC_001348.1. Drop-seq tools v.1.12 was used for mapping raw sequencing data to the reference genome.

*Statistical analysis.* Data normalization, differential expression analysis and $P$ value adjustment were performed with the DESeq2 package (v.1.38.1)[117] using GLM linear modelling with the standard settings. Independent filtering was disabled. For each condition, one of the quadruplicates was excluded following PCA. Transcripts with less than ten counts in at least three samples were excluded. The best sensitivity to identify differentially expressed genes (DEGs) was achieved using the following designs:

- VZV versus mock in NTC cells: defined DEGs following VZV infection of NTC cells (1).
- VZV versus mock in KO cells: defined DEGs following VZV infection of NPHP4-depleted cells (2).
  The log₂-transformed fold changes were shrunken using apeglm. For visualization, we used a scatter plot of shrunken log₂-transformed fold changes between comparisons (1) and (2). Shared DEGs following VZV infection between NTC and KO cells were defined by adjusted $P \le 10^{-4}$ and |shrunken log₂(fold change)| ≥ 1 in each comparison.
- Statistical evaluation of DEGs between KO and NTC cells following infection was performed as follows: KO versus NTC in VZV samples: log₂-transformed fold changes were left unshrunken. DEGs were defined by |log₂(fold change)| ≥ 0.5 and (i) unadjusted $P \le 0.02$ or (ii) unadjusted $P \le 0.1$ and defined as DEGs in (1) or in (2).
- DEGs following NPHP4 gene depletion in mock cells were defined as follows: KO versus NTC in mock samples: log₂-transformed fold changes were left unshrunken. DEGs were defined by |log₂(fold change)| ≥ 0.25 and adjusted $P \le 0.1$.

**AP of SK-N-BE2 cells expressing HA-tagged NPHP4 wild type or S862N mutant.** *Data processing.* The raw MS data files were analysed with MaxQuant (v.1.6.14.0) using the standard settings, with match between runs and LFQ enabled (LFQ min ratio count set to 2, stabilize large LFQ ratios disabled and normalization skipped). The MS2 spectra were searched against forward and reverse (used as decoy) sequences of the reviewed human proteome, including isoforms (UniProtKB, release 2019) and GFP using the built-in Andromeda search engine.

*Statistical analysis.* Protein groups and LFQ intensities generated by MaxQuant (proteinGroups file) were imported into Perseus (2.0.7.0). Protein groups were filtered for contaminants, reverse identification, if only identified by a modified peptide, and if identified in less than two replicates in at least one bait. A background was defined by protein groups identified in all samples (in four of four replicates of all baits). log₂-transformed LFQ intensities were normalized by the median of the background of the given sample. Missing intensities were imputed from a log normal distribution of log-transformed

normalized intensities with a width of 0.3× the s.d. of the measured intensities and downshifted by 1.8 s.d. to simulate the detection limit of the mass spectrometer. Enrichments were calculated as the $\log_2$ difference of the median intensity of each NPHP4 variant with GFP to define interactors, or between the two NPHP4 variants to define specific interactors. Significance was assessed by the Welch $t$-test, adjusted by permutation-based FDR.

### FACS analysis of VZV infection

SK-N-BE2 cells were infected by co-culture with VZV rOka as described earlier in three independent experiments. At 48 h post-infection, cells were detached from the transwell by trypsin-EDTA (Sigma-Aldrich), washed in PBS (Sigma-Aldrich) and fixed with 3.7% formaldehyde (Sigma-Aldrich) for 15 min at room temperature. After a wash in FACS buffer (1% FCS, 2 mM EDTA in PBS), cells were labelled with the VZV DFA Kit reagent (Light Diagnostics; Merck) containing two FITC-conjugated antibodies against VZV immediate-early ORF62 protein and late glycoprotein E for 15 min at room temperature. Labelled cells were analysed on an Attune NxT Acoustic Focusing Cytometer (Thermo Fisher Scientific) and data were processed with FlowJo (v.10). Forward and side scatters were used to exclude cell debris and gate single cells and FITC intensity was analysed.

### WB analysis

**Expression of the V5-tagged VZV proteins.** A total of $5 \times 10^5$ SK-N-BE2 cells expressing individual V5-tagged VZV ORF or GFP were generated as described earlier. Frozen cell pellets were thawed and lysed on ice for 30 min in lysis buffer (0.2% NP-40, 100 mM NaCl, 5% glycine, 1.5 mM MgCl₂, 50 mM Tris-HCl pH 7.5) supplemented with 1% in-house benzonase. Samples were sonicated (5 min, 4 °C, 30 s on and off, high frequency; Bioruptor; Diagenode) and centrifuged at 15,000$g$ at 4 °C for 30 min. Supernatants were collected and total proteins were precipitated overnight at −20 °C in 80% acetone before being resuspended in SDS sample buffer (62.5 mM Tris-HCl pH 6.8, 2% SDS, 10% glycerol, 50 mM DTT, 0.01% bromophenol blue). After boiling for 5 min at 95 °C, samples were loaded on NuPAGE Novex 10% Bis-Tris (Invitrogen) and further submitted to WB using 0.45 µm nitrocellulose membranes (Amersham Protran). Imaging was performed by HRP luminescence using the SuperSignal West Femto kit (Thermo Fisher Scientific).

Expression levels were quantified from the V5-tag blot signal from VZV ORF-expressing SK-N-BE2 cell lines (Extended Data Fig. 2a) using ImageLab (v.6.0.1). The volume of the bait-specific bands was normalized by the volume of the lane-corresponding actin band (loading control). Intensities that required longer exposure detection were normalized by the fold change between short and long exposures, obtained from other bands on the same blot that were detected without saturation in both images. Bands that overlapped the actin signal were excluded from this analysis (ORF36, ORF9 and ORF63). The Pearson correlation coefficient (Pearson's $r$) between the expression level and the numbers of targets and effects were calculated from non-transformed values.

### Degradation of IFI16 in co-transfected HEK293T cells.

A total of $1 \times 10^6$ HEK293T cells were transfected with 500 ng of HA-tagged VZV ORF61, HSV-1 ICP0 or GFP and 100 ng of IFI16 for 24 h. Cells were washed in PBS and pellets were frozen at −20 °C. Pellets were lysed in 2% SDS, 1% DNAse lysis buffer, sonicated (10 min, 4 °C, 30 s on and off, high frequency; Bioruptor; Diagenode) and boiled for 5 min at 95 °C. Protein concentration was measured by Pierce 660 nm Protein Assay (Thermo Fisher Scientific) and normalized in SDS sample buffer. Protein (20 µg) was loaded on 10% Tris-glycine SDS–PAGE and further submitted to western blotting using 0.22 µm nitrocellulose membrane (Amersham Protran). Imaging was performed by HRP luminescence using the Western Lightning Plus-ECL kit (Perkin Elmer).

### IP followed by western blot analysis

**Sample preparation.** *VZV ORF32-V5 IP in SK-N-BE2 cells.* A total of $4 \times 10^6$ SK-N-BE2 cells expressing V5-tagged VZV ORF32 and control cells expressing V5-tagged VZV ORF23 were scraped, washed in ice-cold PBS and flash frozen in liquid nitrogen.

*Reverse IP of VZV ORF32 by GFP-GTF2B in HeLa Kyoto cells.* pLenti6.3-TO-V5 expressing VZV ORF32 or ORF23 (10 µg) was transfected as described above in $1 \times 10^6$ HeLa Kyoto GFP-GTF2B. At 24 h post-transfection, cells were scraped, washed in ice-cold PBS and flash frozen in liquid nitrogen.

**IP and WB.** Cell lysis was performed using lysis buffer (0.2% NP-40, 100 mM NaCl, 5% glycine, 1.5 mM MgCl₂, 50 mM Tris-HCl pH 7.5) supplemented with 1% in-house benzonase as described for the AP–MS. Of the total lysate, 5% was retained as a total lysate control for the WB analysis. IP and washes were performed similarly to the AP–MS, using anti-V5 magnetic beads (MBL M215-11) for V5-ORF32 IP and GFP-Trap Magnetic Agarose (Chromotek) for GFP-GTF2B IP. Washed beads were resuspended in the wash buffer. IP samples and total lysate controls were mixed in SDS sample buffer (62.5 mM Tris-HCl pH 6.8, 2% SDS, 10% glycerol, 50 mM DTT, 0.01% bromophenol blue), boiled at 95 °C for 5 min and loaded on homemade 10% Tris-HCl gels. Proteins were blotted on a 0.45 µm nitrocellulose membrane (Amersham Protran). Imaging was performed by HRP luminescence using the Western Lightning Plus-ECL kit (Perkin Elmer).

### IF analysis

**Fixation.** If not specified otherwise, cells were fixed with 4% formaldehyde for 15 min at room temperature, permeabilized (0.1% Triton X-100 in PBS) for 15 min at room temperature and blocked (0.1% FCS, 1% donkey serum, 0.1% Triton X-100 in PBS) for 1 h at room temperature. Staining was performed by incubation with the indicated primary antibodies at 4 °C overnight, followed by incubation with secondary antibodies for 2 h at room temperature in blocking buffer. DAPI was incubated for 1 min in blocking buffer or included in the secondary antibody incubation.

**Cellular localization of VZV ORFs in SK-N-BE2 cells.** A total of 0.8–$2 \times 10^5$ SK-N-BE2 cells expressing individual V5-tagged VZV ORFs were grown on coverslips precoated with Attachment Factor (Gibco Fisher Scientific) for 48 h before fixation. Three samples were generated and stained independently. Confocal imaging was performed using an LSM900 confocal laser scanning microscope (ZEISS) equipped with a ×63/1.40 oil DIC M27 objective (ZEISS), with Airyscan z2. Airyscan images were processed in Zen 3.5 (blue edition, ZEISS), and brightness and contrast adjustments were made using Zen 3.8 (blue edition, ZEISS).

**Cellular localization of VZV ORF32 and GTF2B.** Cellular localization of VZV ORF32 and GTF2B was analysed by IF analysis in HeLa Kyoto cells stably expressing GFP-tagged GTF2B as previously described[118]. Briefly, HeLa Kyoto GFP-GTF2B cells were grown on coverslips and 2 µg of pLenti6.3-TO-V5 expressing VZV ORF32 or ORF23 were transfected as described earlier. After 36 h, cells were fixed with 4% paraformaldehyde (PFA) for 15 min at room temperature and blocked in blocking buffer (1× PBS containing 0.1% FCS and 0.1% Triton X-100) for 1 h at room temperature. Staining was performed with the indicated antibodies for 1 h at room temperature in blocking buffer. Confocal imaging was performed using an LSM780 confocal laser scanning microscope (ZEISS) equipped with a Plan-APO ×63/NA1.46 immersion oil objective (ZEISS).

**IFI16 degradation and ORF61-UBXN7 co-localization study in VZV-infected cells.** HFF cells were grown on coverslips precoated with Attachment Factor (Gibco Fisher Scientific) and left as mock

or infected by co-culture with MeWo cells infected with VZV(pOka)-61-HA. At 2 h post-infection, cells were left untreated or treated with 1 μM MG-132. Cells were fixed 8 h post-infection. Samples from three experiments were generated and stained independently. Confocal imaging was performed using an LSM900 confocal laser scanning microscope (ZEISS) equipped with a Plan-APO ×20/0.8 M27 objective for the analysis of IFI16 degradation or a ×63/1.40 oil DIC M27 objective for the co-localization analysis of VZV ORF61 and UBXN7 (ZEISS), with Airyscan 2. Airyscan images were processed in Zen 3.5 (blue edition, ZEISS). IFI16 intensity was measured from 10 z-stack images of 0.5-μm intervals centred on IFI16 maximum intensity, processed by maximum intensity projection in Zen 3.5 (blue edition, ZEISS). For VZV ORF61-UBXN7 co-localization analysis, single images at defined z were acquired. Brightness and contrast adjustments and intensity line profile analysis were performed using Zen 3.8 (blue edition, ZEISS).

VZV and IFI16 quantification were analysed with FIJI (v.1.54f; ImageJ). Cell nuclei were masked from the DAPI signal and mean intensities of IFI16 (Alexa Fluor 488) and VZV (Alexa Fluor 647) were analysed. Background VZV signal was determined by its maximum mean intensity measured in mock conditions. Non-infected cells, as defined by the VZV background, were excluded from the infected conditions in the summarized IFI16 plot (Fig. 4c, left).

**IFI16 degradation study in ORF61-expressing cells.** HFF cells were transduced with shRNA-expressing lentivirus for scramble or UBXN7-targeting sequence. Five days later, cells were transduced with V5-tagged VZV ORF61 or GFP as described earlier. Cells were grown for 24 h in 8-well cell culture chambers on a glass slide (Sarstedt) and fixed 7–20 days post-transduction. Samples from two experiments were generated and stained independently as described above with rabbit anti-IFI16 and mouse anti-V5 primary antibodies, anti-rabbit Alexa Fluor 647, anti-mouse Alexa Fluor 488 and DAPI. Confocal imaging was performed using an LSM900 confocal laser scanning microscope (ZEISS) equipped with a Plan-APO ×20/0.8 M27 or ×10/0.45 M27 objective (ZEISS), with Airyscan 2. A z-stack of 1-μm intervals centred on IFI16 maximum intensity and covering the whole signal was acquired. Airyscan images were processed and maximum intensity projections were performed in Zen 3.5 (blue edition, ZEISS).

Data analysis was performed using ICY software (v.2.5.2.0). Nuclei were masked and sorted for positive V5 (Alexa Fluor 488 or Alexa Fluor 647) signal. The nucleoli were isolated using the wavelet spot detector plug-in and subtracted from the region of interest mask for the nuclei. Within these resulting subtraction regions of interest, integrated intensity values of IFI16 (Alexa Fluor 647) were calculated for each nucleus. Outliers were detected with the ROUT method and a maximum desired FDR $Q = 0.5\%$ using GraphPad Prism.

### Host gene knockout screen on VZV growth

**Host gene selection.** Data-driven selection of host proteins upregulated in VZV-infected SK-N-BE2 cells (Supplementary Table 1) or enriched in at least 1 VZV ORF interactome with an enrichment factor above 13 (Supplementary Table 3) was applied. FAM208A, MPP8, APOBEC3B, CNOT6 and USP37 were selected based on biological interest. HCF1 was used as a dependency factor control. DHX9 and MYD88 were additionally included as restriction factor controls. In total, 116 genes from our data and 2 control genes were tested in the knockout screen.

**Generation of knockout cells, VZV infection and flow cytometry analysis.** As VZV propagates mostly via cell-to-cell contact, we set up a fluorescence-based co-culture infection assay to measure the replication of a recombinant VZV using flow cytometry analysis. SK-N-BE2-GFP(blastR) and the SK-N-BE2-spCas9(ZeoR)-BFP(blastR) cells were seeded in a 1:1 ratio in a 96-well plate. The next day, each well was transduced with individual sgRNAs from the host gene library or with an empty vector as NTC, using polybrene at 8 μg ml⁻¹. At

48 h post-transduction, transduced cells were selected with 2 μg ml⁻¹ puromycin and kept in culture for another 12 days to allow clearance of remaining expressed proteins in the target cells. Each well was split into four wells the day before infection. Cell density at infection was monitored by the non-lytic Real-Time Glo luminescence assay (Promega) acquired on an Infinite 200 plate reader (Tecan). Two wells were mock infected by exchanging the culture medium. Two wells were infected at a low multiplicity of infection (MOI; expected between 0.002 and 0.005, depending on individual knockout cell density) by exchanging the medium with a suspension of MeWo cells infected with VZV(pOka)-63-RFP/70-RFP. After 48 h (16 days post-selection), cells were fixed in 1% PFA for 15 min, resuspended in FACS buffer (5 mM EDTA pH 8, 25 mM HEPES, 1% FCS in PBS), and analysed on a CytoFLEX flow cytometer operated by CytExpert (Beckman Coulter). The whole gRNA library was screened within five batches, each batch consisting of independent transduction, selection, infection and measurement steps. NTC was included within each batch.

**Data analysis.** Flow cytometry data were analysed by FlowJo (v.10). Forward and side scatters were used to exclude cell debris and gate single cells. Knockout and control cells were gated from BFP+ and GFP+ cells, respectively. A low percentage of BFP cells in mock wells at 15 days post-selection of transduced cells was associated with a cell growth disadvantage and was used as a proxy for knockout toxicity. Thus, individual gRNA knockouts with less than 35% of BFP cells in mock wells were excluded from subsequent analysis. A total of 218 knockouts covering 108 host genes and the NTC remained. The MRI of the combined population of BFP (knockout) and GFP (control) cells (excluding unmarked inoculum MeWo cells) was used to assess the spread of VZV in each well. The MRIs of each BFP+ and GFP+ population were separately extracted to evaluate the spread of the virus within knockout or control cells, respectively. The normalized target-to-control MRI was used to evaluate each knockout's effect on VZV replication. Pearson's correlation coefficient r between the MRI, average or ratio across BFP and GFP populations, and the absolute difference to the median luminescence intensity were calculated to assess the bias from variability of cell density at infection (Supplementary Discussion). Infection batch effects were normalized by subtracting the batch median MRI ratio from each individual MRI ratio, and then dividing the result by the batch median MRI ratio. Normalized MRI ratios were averaged across replicates and represented as robust z-scores to account for the absolute median deviation within the screen. Host factors for which at least two sgRNAs showed the same effect on VZV growth (increase or decrease), and for which at least one had an absolute robust z-scored ratio above two, were identified as strong restriction or dependency genes for VZV, respectively.

### Validation of the effect of NPHP4, MPP8 and ZNF280D knockouts on VZV infection

**Sample generation, VZV infection and flow cytometry analysis.** Stable knockout SK-N-BE2 cells expressing BFP were generated for host genes of interest as described earlier. Knockout and NTC cells were seeded in 24-well plates in technical duplicates. Just before infection, cell density was monitored by imaging (IncuCyte S3 Live-Cell Analysis System) and masking of the phase contrast (as phase percentage; IncuCyte Software v.2019B Rev2; Sartorius). Cells were infected at an MOI of 0.01 by co-culture with MeWo cells infected with VZV(pOka)-63-RFP/70-RFP. At 48 h post-infection, cells were fixed in 1% PFA for 15 min, washed 3× in PBS, resuspended in FACS buffer (5 mM EDTA pH 8, 25 mM HEPES, 1% FCS in PBS) and analysed on a CytoFLEX flow cytometer operated by CytExpert (Beckman Coulter). The assay was performed in three independent biological replicates. Data were analysed using FlowJo (v.10). SK-N-BE2 cells were distinguished from the inoculum MeWo cells by gating the BFP+ population. VZV spread was then assessed by the MRI of the total BFP+ population. To account for

technical variability of cell density at infection, which affects the spread of VZV, an estimated MRI based on the measured cell density at infection was calculated per well. Polynomial regression was performed to assess the correlation between measured cell density at infection and measured MRI in NTC wells. The model was then applied to calculate an 'estimate' MRI per well based on the measured cell density at infection. The effect of the knockout was then reported as the $\log_2$ difference of the measured MRI to the estimate MRI. The $\log_2$ differences were averaged across technical duplicates and plotted as biological replicates. $P$ values were generated using a one-tailed unpaired $t$-test on biological triplicates of individual knockouts against the NTC.

**Viral growth kinetics.** A total of $2.5 \times 10^5$ SK-N-BE2 cells knocked out for *MPP8*, *ZNF280D*, *NPHP4* or NTC were seeded in 24-well plates in technical duplicates. Cells were infected at an MOI of 0.0002 by co-culture with MeWo cells infected with VZV(pOka)-63-RFP/70-RFP ($10^4$ MeWo cells). Medium was exchanged 3 h post-infection and cells were imaged every 6 h for 5 days (IncuCyte S5 Live-Cell Analysis System). Red and phase signals were processed and masked using the IncuCyte Software (v.2021; Sartorius). Red areas were normalized to the phase and to the first acquisition time (3 h post-infection) to account for viral load (MeWo cell count) variability. The assay was performed in three independent biological replicates. Areas under the curve were calculated using GraphPad Prism.

**Validation of gene knockout efficiency.** NTC or knockout cells' gDNA was extracted as recommended (69504; Qiagen). Each targeted region was amplified by PCR using the GoTaq G2 DNA Polymerase (Promega) and flanking primers, before Sanger sequencing (Eurofins Genomics). Data were analysed using an ICE tool by Synthego[98]. The $R^2$ value indicates how well the calculation explains the sequencing trace.

**MS analysis of protein depletion.** For each knockout cell line, $1 \times 10^6$ cells were prepared in triplicate and analysed by MS as described earlier (see 'Proteome changes induced by MPP8 gene depletion in SK-N-BE2 cells'). To allow for the correct identification of the proteins of interest, matching samples were generated by transfection of 0.7 µg expression plasmid for the given protein (pWPI-nHA-PuroR backbone) in $5 \times 10^5$ HEK293T cells. The next day, cells were prepared as described earlier, along with the knockout samples. The raw MS data files were processed with MaxQuant (v.1.6.10.0), using the standard settings with 'iBAQ' quantification and match between runs enabled. The MS2 spectra were searched against forward and reverse (used as decoy) sequences of the reviewed human proteome, including isoforms (UniProtKB, release 2018.02), by the built-in Andromeda search engine. iBAQ (intensity-based absolute quantification) intensities were normalized by the median of all intensities per sample and averaged across knockout replicates.

**WB analysis of protein depletion.** Knockout and NTC SK-N-BE2 cells expressing BFP were flash frozen in liquid nitrogen as cell pellet and stored at −80 °C. Frozen cell pellets were thawed and lysed on ice for 45 min in lysis buffer (RIPA with 1% SDS supplemented with Complete EDTA-free Protease Inhibitor; Roche). Samples were sonicated (10 min, 4 °C, 30 s on and off, high frequency; Bioruptor; Diagenode) and centrifuged at 15,000$g$ at 4 °C for 30 min. Supernatants were collected and total proteins were precipitated overnight at −20 °C in 80% acetone before being resuspended in SDS sample buffer (62.5 mM Tris-HCl pH 6.8, 2% SDS, 10% glycerol, 50 mM DTT, 0.01% bromophenol blue). Total protein concentrations were measured by Pierce 660 nm Protein Assay (Thermo Fisher Scientific) and normalized to 25 µg in 10 µl SDS sample buffer. After boiling for 5 min at 95 °C, samples were loaded on NuPAGE Novex 4–12% Bis-Tris (Invitrogen) and further submitted to western blotting using 0.45 µm nitrocellulose membrane (Amersham Protran). Imaging was performed by HRP luminescence using the SuperSignal West Femto kit (Thermo Fisher Scientific).

**Viability assay.** Knockout and NTC SK-N-BE2 cell lines were seeded in a 96-well plate. At the given time point, 10 µg per well of resazurin was added in triplicate, the cells were incubated at 37 °C for 90 min, and the fluorescence was measured on the plate reader Infinite 200 (Tecan). Technical triplicate values were averaged. The assay was conducted in biological triplicate.

### Rescue of NPHP4 knockout cell lines and VZV infection
**Sample generation.** NPHP4 knockout or NTC SK-N-BE2 cells, expressing BFP, were transduced with a control empty vector or a vector encoding for N-terminal HA-tagged NPHP4 or N-terminal HA-tagged NPHP4-S862N variant (pWPI-nHA-PuroR backbone) using 8 µg ml$^{-1}$ polybrene. Culture medium was exchanged 6 h post-transduction. The day before the assay, cells were counted and seeded for VZV infection or WB analysis. At 72 h post-transduction, 1 well was collected in SDS sample buffer (62.5 mM Tris-HCl pH 6.8, 2% SDS, 10% glycerol, 50 mM DTT, 0.01% bromophenol blue) and frozen at −20 °C for later WB analysis, 1 well was mock infected, and 3 wells were infected at an MOI of 0.02 by co-culture with MeWo cells infected with VZV(pOka)-63-RFP/70-RFP for 48 h.

**Flow cytometry analysis.** At 48 h, mock- and VZV-infected cells were fixed in 1% PFA for 15 min, washed 3× in PBS, resuspended in FACS buffer (5 mM EDTA pH8, 25 mM HEPES, 1% FCS in PBS) and analysed on a Cyto-FLEX flow cytometer operated by CytExpert (Beckman Coulter). The assay was performed in three independent biological replicates. Data were analysed using FlowJo (v.10). SK-N-BE2 cells were isolated from the inoculum MeWo cells by sorting the single BFP+ cells. VZV spread was assessed by the MRI. MRI was averaged across technical replicates and normalized across biological replicates by calculating the fold change to the given NTC empty vector sample. $P$ values were generated by unpaired one-way analysis of variance (ANOVA) and adjusted using the Bonferroni method.

**WB analysis of ectopic HA-tagged NPHP4 expression.** Cells resuspended in SDS sample buffer were thawed at room temperature, sonicated (5 min, 4 °C, 30 s on and off, high frequency; Bioruptor; Diagenode) and boiled for 10 min at 95 °C. Samples were loaded on NuPAGE Novex 4–12% Bis-Tris (Invitrogen), and then submitted to western blotting using 0.22 µm nitrocellulose membrane (Amersham Protran). Imaging was performed by HRP luminescence using the Western Lightning Plus-ECL kit (Perkin Elmer).

### VZV patient cohort and WES
**Patients.** The patient cohort consisted of adults (>18 years) admitted to the Department of Infectious Diseases at Aarhus University Hospital with a final diagnosis of VZV-associated encephalitis, meningoencephalitis or cerebral vasculitis. A total of 13 adult patients were included on the basis of pleocytosis and a positive PCR for VZV in the cerebrospinal fluid. PCR on cerebrospinal fluid for HSV-1, HSV-2, and enterovirus, along with bacterial cultures, were also performed. All results were negative for all included patients. Exclusion criteria were immunosuppressive therapy, known malignant disease, pregnancy and HIV positivity. All patients had experienced chickenpox in childhood and none had received VZV vaccination (Supplementary Table 6-1).

**Blood sampling.** A total of 50 ml of blood was drawn for isolation of PBMCs and DNA for further analysis. DNA was isolated from EDTA-stabilized blood using the EZ1 DNA Blood 350 µl Kit and an EZ1 Advanced XL instrument (Qiagen) according to the manufacturer's instructions[119].

**WES, bioinformatics and integration of VZV–host interactions.** WES was performed using Kapa HTP Library Preparation Kit and the Nimblegen SeqCap EZ MedExome Plus kit. Sequencing was performed on an

Illumina Nextseq 550 system. Single nucleotide polymorphism calling was performed relative to hg19 using BWA. PCR and optical duplicates were identified and marked. The alignment was recalibrated using GATK. Single nucleotide polymorphisms were called using Haplotype-Caller from the GATK package. VCF files were submitted to Ingenuity Variant Analysis. Unconfident (call quality below 30.0, read depth below 5.0 and allele fraction below 25.0) and common (ExAC database, 1000 Genomes, and GnomAD minor allele frequency > 0.01%) variants were excluded. Known variants classified as 'pathogenic' or unknown variants with a SIFT/PolyPhen prediction as 'damaging' were considered as predicted deleterious. Further filtering was applied to keep only variants with a Combined Annotation Dependent Depletion score > 15, a mutation significance cut-off below the Combined Annotation Dependent Depletion score, localized in exons and non-synonymous. A list of genes assembled from the host proteins with differential protein abundance following infection (Supplementary Table 1) and VZV proteins' host targets as identified by AP–MS (Supplementary Table 3) was submitted as a biological context filter. Finally, identified variants were manually checked by inspecting BAM files using IGV.

### Reporting summary

Further information on research design is available in the Nature Portfolio Reporting Summary linked to this article.

## Data availability

The following public datasets were used in the study: GO annotations (http://download.baderlab.org/EM_Genesets/October_01_2020/Human/UniProt/Human_GO_AllPathways_with_GO_iea_October_01_2020_UniProt.gmt, v.2020.10, and http://download.baderlab.org/EM_Genesets/December_01_2021/Human/UniProt/Human_GO_AllPathways_with_GO_iea_December_01_2021_UniProt.gmt, v.2021.12), IntAct Protein Interactions (https://www.ebi.ac.uk/intact/, v.2019.12), IntAct Protein Complexes (https://www.ebi.ac.uk/complexportal/home, v.2019.12), CORUM Protein Complexes (http://mips.helmholtz-muenchen.de/corum/download/allComplexes.xml.zip, v.2019), Reactome Functional Interactions (https://reactome.org/download/tools/ReatomeFIs/FIsInGene_122220_with_annotations.txt.zip, v.2020.12), BioGRID (https://downloads.thebiogrid.org/File/BioGRID/Release-Archive/BIOGRID-3.5.178/BIOGRID-ALL-3.5.178.psi25.zip, v.2019.10), VirHostNet (v.2.0; 2019.01), Human (v.2018.02 and v.2019.12) and VZV pOka (v.2017.12) protein sequences (https://uniprot.org), human genomes GRCh38 (https://www.ncbi.nlm.nih.gov/assembly/GCF_000001405.26/; transcriptome analysis) and GRCh37 (hg19; https://www.ncbi.nlm.nih.gov/assembly/GCF_000001405.13/; WES), and VZV genome from GenBank (https://www.ncbi.nlm.nih.gov/nuccore/NC_001348.1). The MS proteomics data have been deposited to the ProteomeXchange Consortium via the PRIDE[120] partner repository with the following dataset identifiers: full proteome of VZV infected, PXD047273; VZV ORF interactomes, PXD047821; VZV ORF effectomes, PXD047362; validation of knockouts, PXD047575; full proteome of MPP8-knockout cells, PXD047393; and NPHP4 interactomes, PXD061602. The data and analysis results are accessible online via the interactive web interface at https://vari-zonet.innatelab.org. The VZV ORF interactomes have been submitted to the IMEx (http://www.imexconsortium.org) consortium through IntAct[121] and assigned the identifier IM-30341. The transcriptome data (NPHP4-knockout SK-N-BE2 cells mock and VZV infected) have been deposited in the European Nucleotide Archive at EMBL-EBI under accession number PRJEB86994. Source data are provided with this paper.

## Code availability

In-house R and Julia packages and scripts used for the bioinformatics analysis of the data have been deposited to public Zenodo repositories: https://doi.org/10.5281/zenodo.15647702 (ref. 102), https://doi.org/10.5281/zenodo.4536605 (ref. 43), https://doi.org/10.5281/zenodo.4536596 (ref. 110), https://doi.org/10.5281/zenodo.10487870 (ref. 111), https://doi.org/10.5281/zenodo.7746897 (ref. 103) and https://doi.org/10.5281/zenodo.4536604 (ref. 114).

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

## Acknowledgements

We thank the patients involved in this study. We thank R. Baier, E. Özaltin, J. Basquin, A. Kosinska and S. M. Hamad for technical assistance. We thank F. Seelig for his valuable input in the writing of this paper. We also thank J. Cohen (NIH) for VZV rOka, M. Friedrich (Technical University of Munich) for the pLVX plasmid, G. Superti-Furga (CeMM) for ICP0-pDONR201, C. Rice (Rockefeller University) for pSCRPSY-PAC2A-tagBFP, M. Brinkmann (HZI) for HFF-1 cells, R. Klein (MPI of Neurobiology) for SK-N-BE2 and A.-V. Borbolla (MHH) for MeWo cells. This work was supported by PhD scholarships to V.G., M.E.C.-T. and J.H., who were funded by the European Union under the Horizon 2020 Research and Innovation program (H2020) and Marie Skłodowska-Curie Actions–Innovative Training Networks Programme MSCA-ITN GA 675278 EDGE (Training Network providing cutting-EDGE knowlEDGE on Herpes Virology and Immunology). Work in the authors' laboratories was supported by ERC consolidator grants ERC-CoG ProDAP (817798) and ERC-CoG ENDo-HERPES (101087480) to A. Pichlmair and B.B.K., respectively; the Danish National Research Foundation (DNRF 164; CiViA) for T.H.M. and A. Pichlmair; and the 'COVIPA' consortium (KA1-Co-02 Helmholtz Association's Initiative and Networking Fund) for U.P. and A. Pichlmair. T.H.M. was supported by the Independent Research Fund Denmark (0134-00006B). The work was also supported by the German Federal Ministry of Education and Research (Cluster4Future CNATM) and TUM Innovation Network NextGenDrugs funded under the Excellence Strategy of the Federal Government and the States (to R.R.). Work in the laboratory of U.P. was supported by the German Research Foundation (TRR179/TP14) and the NabScreen project funded by the Federal Ministry of Science and Education (BMBF). J.R. was supported by the UK Medical Research Council (MRC core funding of the MRC Human Immunology Unit, grant number MC_UU_00008/8) and the Wellcome Trust (grant number 100954/Z/13/Z). A. Pichlmair was supported by the German Research Foundation (TRR179/TP11, TRR237/A07 and TRR353/B04) and the 'Prevention of Pandemic-infection-associated Pathology Munich – P3M' funded by the State of Bavaria (BayVFP 2024-2027). The funder had no role in study design, data collection and analysis, decision to publish or preparation of the paper.

## Author contributions

Conceptualization: V.G., A.S., B.B.K., J.R., R.J.L., T.H.M., A. Pichlmair. Investigation: V.G., M.E.C.-T., J.H., K.A., R.Ö., D.A.H., S.M., L.O., A. Piras. Formal analysis: V.G., A.S., M.V. Funding acquisition: J.R., R.J.L., T.H.M., R.R., U.P., A. Pichlmair. Supervision: A. Pichlmair. Writing: V.G., A.S., M.V., A. Pichlmair, with input from M.E.C.-T., D.A.H., B.B.K., J.R., R.J.L., T.H.M. Website: V.G., A.S., M.V., A. Pichlmair.

## Funding

## Competing interests

The authors declare no competing interests.

## Additional information

**Extended data** is available for this paper at https://doi.org/10.1038/s41564-025-02068-7.

**Correspondence and requests for materials** should be addressed to Andreas Pichlmair.

[1]Institute of Virology, School of Medicine and Health, Technical University of Munich, Munich, Germany. [2]Department of Infectious Diseases, Aarhus University Hospital (AUH), Aarhus, Denmark. [3]Medical Research Council Human Immunology Unit, Medical Research Council Weatherall Institute of Molecular Medicine, Radcliffe Department of Medicine, University of Oxford, Oxford, UK. [4]Systems Virology, Helmholtz Center Munich, Munich, Germany. [5]Institute of Molecular Oncology and Functional Genomics and Department of Medicine II, School of Medicine, Technical University of Munich, Munich, Germany. [6]German Centre for Infection Research (DZIF), Munich, Germany. [7]Institute of Virology, Department of Veterinary Medicine at the Freie Universität Berlin, Berlin, Germany. [8]Department of Medical Microbiology, University Medical Center Utrecht, Utrecht, the Netherlands. [9]Department of Biomedicine, Aarhus University, Aarhus, Denmark. [10]Center for Immunology of Viral Infections (CiViA), Aarhus University, Aarhus, Denmark. ✉e-mail: andreas.pichlmair@tum.de

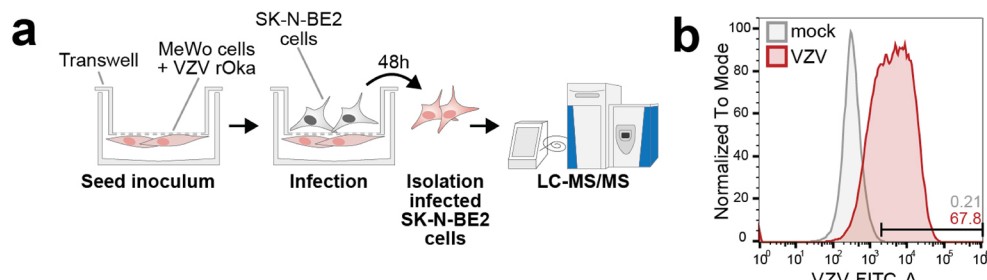

**Extended Data Fig. 1 | VZV infection of SK-N-BE2 cells. (a)** Schematic representation of the infection by transwell co-culture with VZV-infected MeWo cells inoculum. **(b)** Histogram of fluorescence intensity of SK-N-BE2 cells mock-(grey) or VZV-infected (red) with rOka VZV for 48 h as described in (a), and stained for FITC-conjugated antibodies against VZV immediate-early ORF62 protein and late glycoprotein E. Percentage of FITC-positive cells are indicated. (n = 3 independent experiments).

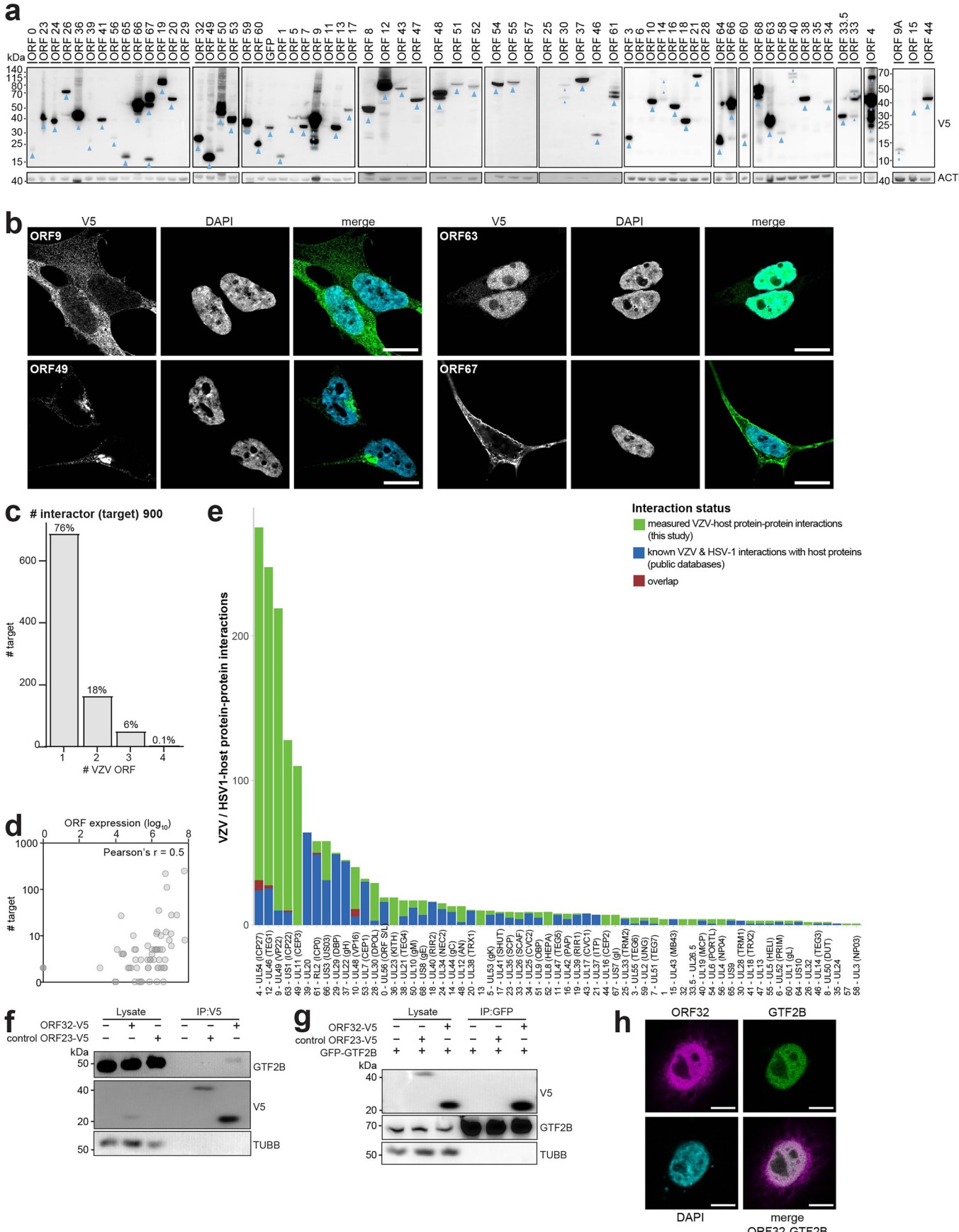

**Extended Data Fig. 2 | See next page for caption.**

**Extended Data Fig. 2 | Evaluation of the V5-tagged VZV proteins' expression and host interactome. (a)** Expression of V5-tagged VZV proteins, in stably transduced SK-N-BE2 cells used in interactome and effectome analysis. Blue arrows indicate the bands used for quantification. n = 4 independent experiments. **(b)** Subcellular localization analysis by immunofluorescence of the indicated transduced V5-tagged VZV ORFs in SK-N-BE2 cells. Cells were fixed and stained with V5+rbAlexa488 (green) and DAPI (cyan), and subjected to confocal microscopy at 63x magnification. Images are representative of n = 3 independent experiments. Scale bar = 10µm. **(c)** Number of host interactor (target) targeted by one or several VZV ORFs, as represented in Fig. 2a. The percentages of total interactors are indicated on top of the bar. **(d)** Correlation analysis between the number of targets (Fig. 2a) and the expression level of individual VZV ORF as quantified from Extended Data Fig. 2a. **(e)** Intersection of the measured VZV-host protein-protein interactions with known VZV and HSV-1 interactions from public databases (Biogrid, Intact, VirHostNet 2.0). (Supplementary Table 2).

VZV proteins are numbered according to their gene name. The gene name of the corresponding HSV-1 homologous protein and the common short protein name (bracket) are indicated. **(f)** Western-blot analysis of the V5 immunoprecipitation of SK-N-BE2 cells expressing VZV ORF32-V5, VZV ORF23-V5 or non-transduced. Total lysates and immunoprecipitation samples (IP) are shown (n = 1 independent experiment). **(g)** Western blot analysis of GFP immunoprecipitation of Hela Kyoto cells expressing GFP-GTF2B, 24 h after transfection of VZV ORF32-V5, VZV ORF23-V5 or non-transfected. Total lysates and immunoprecipitated (IP) samples are shown (n = 1 independent experiment). **(h)** Immunofluorescence analysis of HeLa Kyoto cells stably expressing GFP-GTF2B and transiently transfected with VZV ORF32-V5. Cells were stained with V5+rbAlexa594 (magenta), GFP-DyLight-488 (green) and DAPI (cyan), and subjected to confocal microscopy at 63x magnification. Images are representative of n = 2 independent experiments. Scale bar = 10µm.

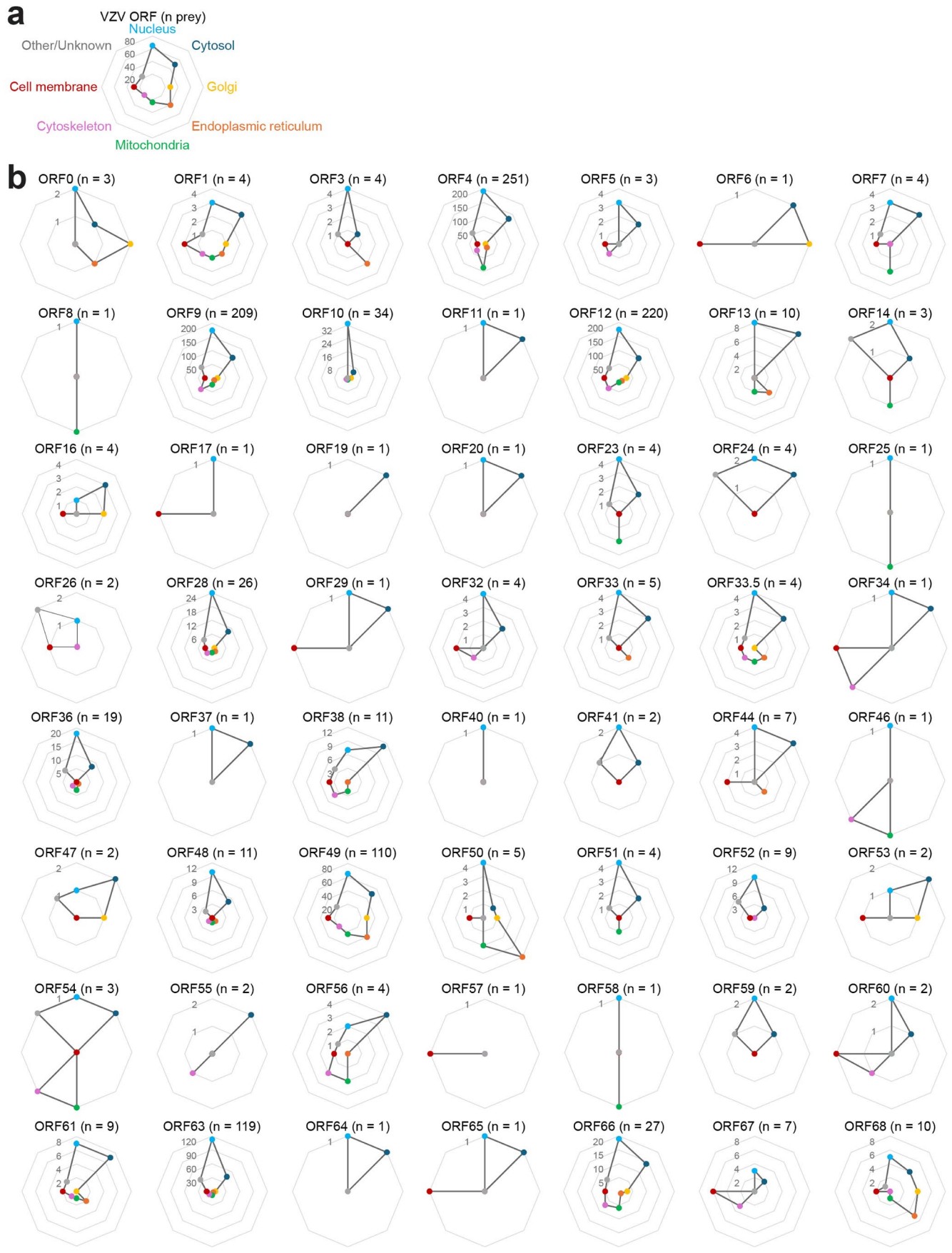

**Extended Data Fig. 3 | Prediction of the subcellular localisation of the VZV proteins.** (**a**) For each VZV ORF interactome (Fig. 2a), the host preys were categorised according to their subcellular localisation annotations (Gene Ontology Cellular Component, Uniprot, Protein Atlas), as indicated

(Supplementary Table 3). (**b**) Systematic analysis of the individual VZV ORF as described in (**a**). n indicates the total number of preys for a given ORF. The numbers in grey indicate the number of preys assigned to the given subcellular category for each VZV ORF.

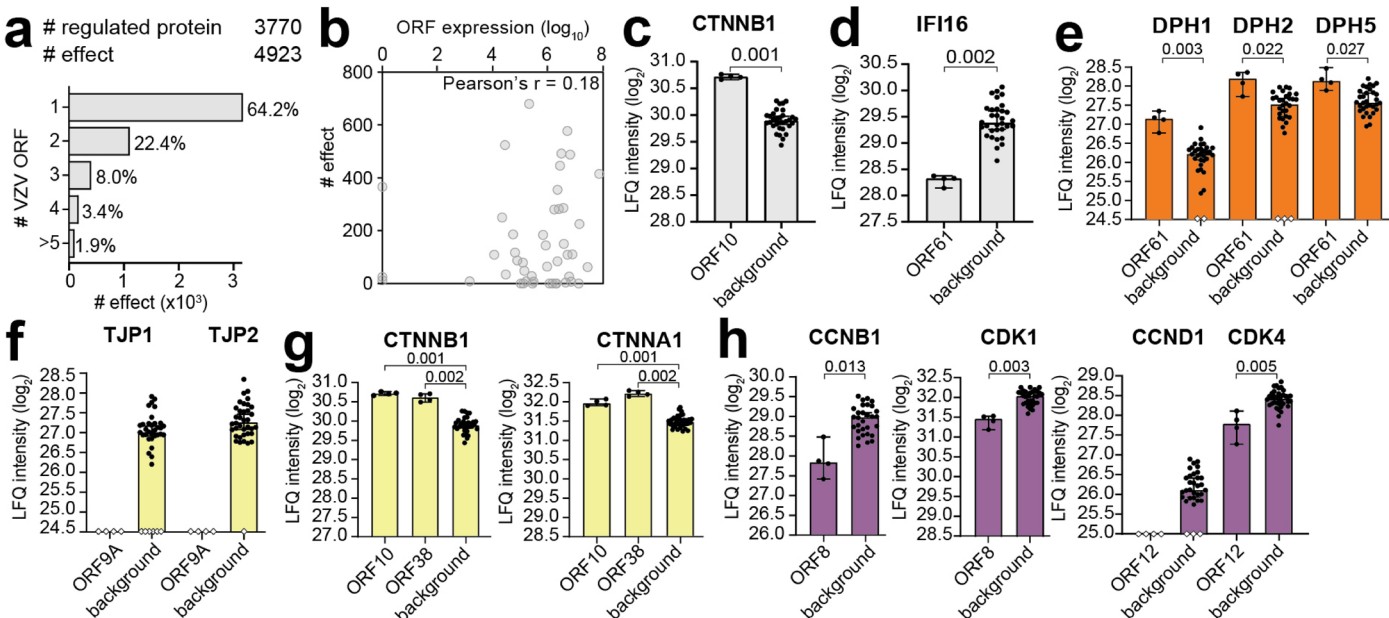

**Extended Data Fig. 4 | Host protein abundance changes identified in the effectome of VZV proteins. (a)** Number of host protein regulated and number of total up- and down-regulation (effect) observed in the effectome of one or several VZV ORFs, as represented in Fig. 3a. The percentages of total effects are indicated. **(b)** Correlation analysis between the number of effects (Fig. 3a) and the expression level of individual VZV ORF as quantified from Extended Data Fig. 2a. **(c, d)** $\log_2$ LFQ intensity of the indicated host protein measured in SK-N-BE2 cells expressing the given VZV ORF. The background is defined by VZV ORF expressing cells from the same measurement batch (See Methods). Median and 95% confidence intervals are indicated (n = 4 independent experiments per VZV ORF). p-value of the effect as compared to the background is indicated (two-sided Wilcoxon rank-sum test; Benjamini-Hochberg adjusted). **(e-h)** Change of protein

abundance of the components of **(e)** the peptidyl-diphthamide metabolic pathway following ORF61 expression, **(f)** the tight junction complex following ORF9A expression, **(g)** the adherens junction complex following ORF10 or ORF38 expressions and **(h)** the cell cycle cyclin-kinase complexes following ORF8 or ORF12 expression. $\log_2$ LFQ intensity of the indicated host protein were measured in SK-N-BE2 cells expressing the given VZV ORF by LC-MS/MS. The background is defined by VZV ORF expressing cells from the same measurement batch (See Methods). Bars are coloured according to the regulated cellular function the given protein belongs to, as represented in Fig. 3b. Median and 95% confidence intervals are indicated (n = 4 independent experiments per VZV ORF). p-value of the effect as compared to the background is indicated (two-sided Wilcoxon rank-sum test; Benjamini-Hochberg adjusted).

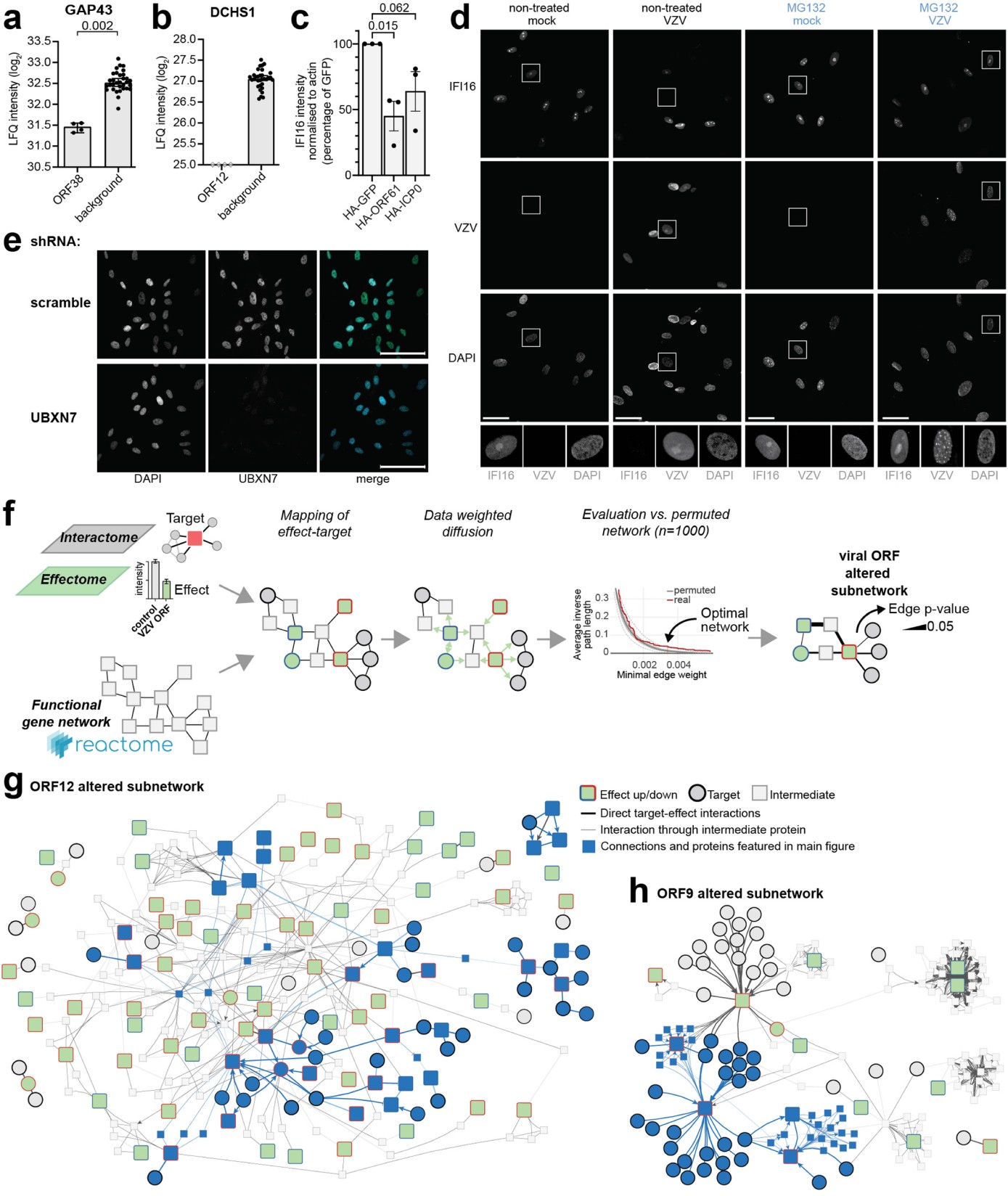

**Extended Data Fig. 5 | See next page for caption.**

**Extended Data Fig. 5 | Hypothesis-driven and systematic data integration.**
**(a, b)** Intensity of the indicated host protein measured by mass spectrometry in
SK-N-BE2 cells, expressing the given VZV ORF or from the background (Methods).
Median and 95% confidence intervals are indicated (n = 4 independent
experiments per ORF; two-sided Wilcoxon rank-sum test; Benjamini-Hochberg
adjusted). **(c)** Western blot analysis of IFI16 in HEK293T cells after co-transfection
with HA-GFP, -ORF61 or -ICP0. Mean +/− SEM are indicated (n = 3 independent
experiments; paired one-way ANOVA test, Bonferroni adjusted). A representative
blot is presented in Fig. 4b. **(d)** Immunofluorescence analysis of HFF cells
mock- or recombinant HA-ORF61 VZV-infected for 8 h, treated or not with the
proteasome inhibitor MG132. HFF cells were stained for IFI16, VZV and with
DAPI, and analysed at 20x magnification. Representative of n = 3 independent
infection experiments. Analysis of nuclear intensities is summarized in Fig.
4c. Scale bar = 50 μm **(e)** Validation of the knockdown of UBXN7 in HFF cells

by immunofluorescence analysis. Scale bar = 100 μm **(f)** Network diffusion
analysis of VZV ORFs. Interactome and effectome data were mapped onto the
cellular gene network ReactomeFI. Effects were used to weight the random
diffusion, resulting in several subnetworks of given minimal edge weight and
average effect-target path length. Comparison between the 'real data' and
randomly permuted networks of given minimal edge weight defines the optimal
subnetwork with the best proximity between targets and effects. The displayed
subnetwork connects targeted, affected and intermediate proteins and provides
p-value which evaluate the significance of the connections as compared to the
random network. **(g, h)** Altered subnetworks of ORF12 (g) and ORF9 (h) resulting
from the network diffusion analysis. Connections featured in Fig. 4g and h
are indicated in blue. Gene names and nature of the connections (for example
protein-protein complex; activation; expression control) can be browsed on
interactive versions in Supplementary Data.

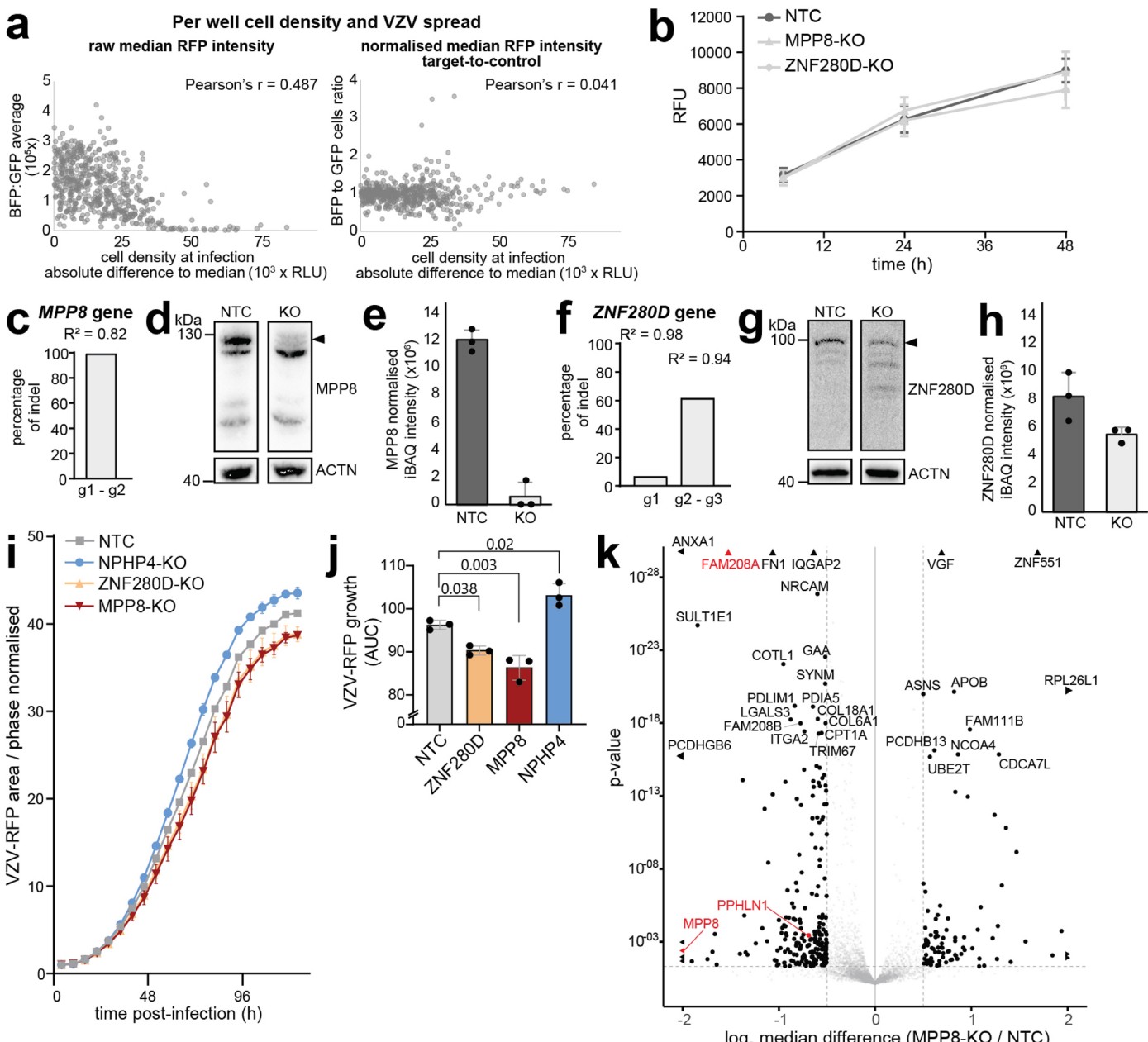

**Extended Data Fig. 6 | CRISPR-Cas9-based knockout of identified host genes and their functional evaluation. (a)** Monitoring of cell density at infection for the host gene knockout screen of VZV replication. Per well correlation analysis between cell density at infection and the raw median RFP intensity of the whole SK-N-BE2 cell population (GFP+ and BFP+) (left), or the target-to-control normalized median RFP intensity (BFP + / GFP+) (right). **(b)** Resazurin viability assay of the NTC and indicated knockout (KO) cell lines. Mean and standard error to the mean are indicated (n = 3 independent experiments) **(c,f)** Percentage of nucleotide insertion/deletion (indel) within the sgRNA-targeted regions of the (c) *MPP8* and (f) *ZNF280D* genes in the MPP8-KO cells as compared to NTC. **(d-h)** Abundance of (d,e) MPP8 and (g,h) ZND280D proteins assessed by western blot (d,g; exact band indicated by an arrow) and mass spectrometry

(e,h; mean normalized intensity +/− SD) analysis in KO and NTC cells. **(i)** Viral growth kinetics within SK-N-BE2 cells NTC or knockout for the indicated gene, and infected with mRFP-VZV(pOka), monitored by live-imaging. Representative of n = 3 independent experiments. **(j)** Area under the curve (AUC) analysis of the VZV-mRFP growth kinetics in SK-N-BE2 cells NTC or knockout for the indicated gene, as presented in (f). Mean and standard error to the mean are indicated (n = 3 independent experiments; paired one-way ANOVA test, Bonferroni adjusted). **(k)** Volcano plot of protein abundance changes induced by MPP8 knockout in SK-N-BE2 cells as compared to NTC. Significant host protein changes (Bayesian linear model-based unadjusted two-sided p-value ≤ 5.10$^{-2}$, |median log2 fold change| ≥ 0.5, n = 4 independent experiments) are represented with black dots. Members of the HUSH complex are indicated in red. (Supplementary Table 5). h = hour.

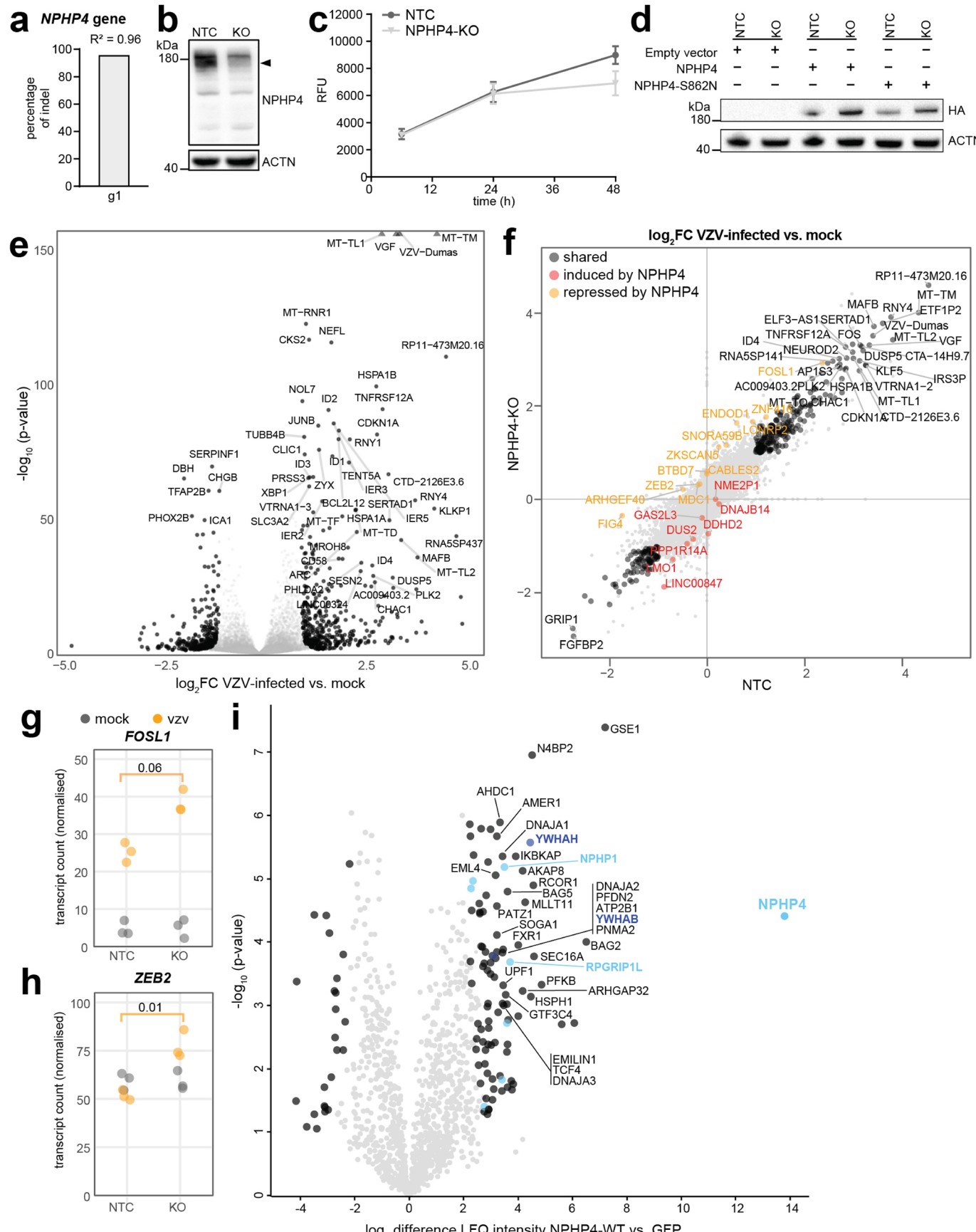

**Extended Data Fig. 7 | See next page for caption.**

**Extended Data Fig. 7 | Functional characterisation of NPHP4 and its S862N variant during VZV infection. (a)** The percentage of nucleotide insertion/deletion (indel) within the sgRNA-targeted region of the *NPHP4* gene in the NPHP4-KO cells as compared to NTC. **(b)** The abundance of NPHP4 protein was assessed by western blot analysis (exact band indicated by an arrow). **(c)** Resazurin viability assay of the NTC and NPHP4 knockout cell line. Mean and standard error to the mean are indicated (n = 3 independent experiments) **(d)** Western blot analysis of the expression level of the HA-NPHP4 and -NPHP4-S862N constructs in NTC and NPHP4-knockout (KO) cells. **(e)** Volcano plot of differentially regulated transcripts in SK-N-BE2 cells infected with VZV, as compared to mock cells. Significant host protein changes (two-sided adjusted p-value ≤ 5.10$^{-2}$, |median log$_2$ fold change| ≥ 1, n = 3 independent experiments) are represented with black dots. (Supplementary Table 5) **(f)** Comparison of differentially regulated transcripts (log$_2$ fold change) following VZV infection in NTC and NPHP4 knockout (KO) SK-N-BE2 cells. (n = 3 independent experiments) (Supplementary Table 5) **(g, h)** Normalized transcript counts of the WNT target genes *FOSL1* and *ZEB2* in SK-N-BE2 cells NTC or NPHP4-KO following VZV-infection as displayed in (f) (two-sided Wald test unadjusted p-value are indicated; n = 3 independent experiments) **(i)** Interactomes of HA-NPHP4 wild-type generated by affinity-purification coupled to mass-spectrometry in neuroblastoma SK-N-BE2 cells. Significant host protein association are marked in black (two-sided Welch t-test, permutation-based FDR ≤ 1.10$^{-2}$, S0 = 4, n = 4 independent experiments) (Supplementary Table 5). Associated cilia basal body proteins (Reactome 267965) are indicated in light blue, including the NPHP1-NPHP4-NPHP8(RPGRIP1L) complex which is labelled. Associated 14-3-3 proteins are indicated in dark blue.

# Reporting Summary

## Statistics

For all statistical analyses, confirm that the following items are present in the figure legend, table legend, main text, or Methods section.

| n/a | Confirmed | |
|---|---|---|
| ☐ | ☒ | The exact sample size (n) for each experimental group/condition, given as a discrete number and unit of measurement |
| ☐ | ☒ | A statement on whether measurements were taken from distinct samples or whether the same sample was measured repeatedly |
| ☐ | ☒ | The statistical test(s) used AND whether they are one- or two-sided *Only common tests should be described solely by name; describe more complex techniques in the Methods section.* |
| ☐ | ☒ | A description of all covariates tested |
| ☐ | ☒ | A description of any assumptions or corrections, such as tests of normality and adjustment for multiple comparisons |
| ☐ | ☒ | A full description of the statistical parameters including central tendency (e.g. means) or other basic estimates (e.g. regression coefficient) AND variation (e.g. standard deviation) or associated estimates of uncertainty (e.g. confidence intervals) |
| ☐ | ☒ | For null hypothesis testing, the test statistic (e.g. F, t, r) with confidence intervals, effect sizes, degrees of freedom and P value noted *Give P values as exact values whenever suitable.* |
| ☐ | ☒ | For Bayesian analysis, information on the choice of priors and Markov chain Monte Carlo settings |
| ☐ | ☒ | For hierarchical and complex designs, identification of the appropriate level for tests and full reporting of outcomes |
| ☐ | ☒ | Estimates of effect sizes (e.g. Cohen's d, Pearson's r), indicating how they were calculated |

*Our web collection on statistics for biologists contains articles on many of the points above.*

## Software and code

Policy information about availability of computer code

| Data collection | The mass spectrometer was operated and MS spectra acquired using the XCalibur software (Thermo Fisher Scientific). Flow cytometry data: CytExpert software operating the CytoFLEX device. Confocal images: Zen 3.5 (blue edition, ZEISS). Live imaging data: IncuCyte software (v2021, Sartorius) Transcriptome data: Illumina Nextseq 500 system; Genome mapping was done with Dropseq (v1.12, http://mccarrolllab.org/dropseq/) WES data: Illumina Nextseq 550 system. GATK and BWA were used for alignment, calling variants and generate the VCF files. |
|---|---|
| Data analysis | MS data analysis: MaxQuant (v.1.6.0.15, 1.6.10.0, 1.6.14.0); Perseus (v.1.6.5.0, 2.0.7.0); R (v.3.6 and 4.1), R Studio (v.2022.07.2+576), Julia (v.1.5 and 1.6) and Python (v.3.8); In-house R and Julia packages and scripts used for the bioinformatics analysis of the data have been deposited to public GitHub repositories: in-house scripts (for msglm and hotnet): https://doi.org/10.5281/zenodo.15647702; msimportr: https://doi.org/10.5281/zenodo.7746897; msglm: https://doi.org/10.5281/zenodo.4536605, https://doi.org/10.5281/zenodo.4536604; gene set enrichment: https://doi.org/10.5281/zenodo.4536596; hierarchical hotnet: https://doi.org/10.5281/zenodo.10487870; rstan package (v.2.19). Transcriptome analysis: DESeq2 (v.1.38.1) Networks visualization: yEd (v.3.2) FACS data analysis: FlowJo (version 10). Western blot quantification: ImageLab (v. 6.0.1) Live-cell images were processed and masked with IncuCyte Software; Sartorius, v2019B Rev2 or v2021). Confocal images were processed with Zen blue edition (ZEISS): versions 3.5 (airyscan) and 3.8 (brightnes and contrast adjustments and intensity line profiles). Masking and quantification of confocal images were done with FIJI (ImageJ; version 1.54f) or ICY. |

WES variants from patients were filtered using Ingenuity Variant Analysis, and verified using Integrative Genomics Viewer (IGV).
Gene knockout sequencing: Synthego Performance Analysis, ICE Analysis. Synthego (used in 2023)

For manuscripts utilizing custom algorithms or software that are central to the research but not yet described in published literature, software must be made available to editors and reviewers. We strongly encourage code deposition in a community repository (e.g. GitHub). See the Nature Portfolio guidelines for submitting code & software for further information.

# Data

Policy information about availability of data

All manuscripts must include a data availability statement. This statement should provide the following information, where applicable:
- Accession codes, unique identifiers, or web links for publicly available datasets
- A description of any restrictions on data availability
- For clinical datasets or third party data, please ensure that the statement adheres to our policy

The following public datasets were used in the study: Gene Ontology annotations (http://download.baderlab.org/EM_Genesets/October_01_2020/Human/UniProt/Human_GO_AllPathways_with_GO_iea_October_01_2020_UniProt.gmt, v2020.10) and (http://download.baderlab.org/EM_Genesets/December_01_2021/Human/UniProt/Human_GO_AllPathways_with_GO_iea_December_01_2021_UniProt.gmt, v2021.12); IntAct Protein Interactions (https://www.ebi.ac.uk/intact/, v2019.12); IntAct Protein Complexes (https://www.ebi.ac.uk/complexportal/home, v2019.12); CORUM Protein Complexes (http://mips.helmholtz-muenchen.de/corum/download/allComplexes.xml.zip, v2019); Reactome Functional Interactions (https://reactome.org/download/tools/ReatomeFIs/FIsInGene_122220_with_annotations.txt.zip, v2020.12); BioGRID (https://downloads.thebiogrid.org/File/BioGRID/Release-Archive/BIOGRID-3.5.178/BIOGRID-ALL-3.5.178.psi25.zip, v2019.10), VirHostNet (v2.0; 2019.01); Human (versions 2018.02 and 2019.12), and VZV pOka (version 2017.12) protein sequences(https://uniprot.org); Human genomes GRCh38 (https://www.ncbi.nlm.nih.gov/assembly/GCF_000001405.26/) (transcriptome analysis) and GRCh37 (hg19, https://www.ncbi.nlm.nih.gov/assembly/GCF_000001405.13/) (WES). VZV genome from GenBank (https://www.ncbi.nlm.nih.gov/nuccore/NC_001348.1)

The mass spectrometry proteomics data have been deposited to the ProteomeXchange Consortium via the PRIDE partner repository with the following dataset identifiers: Full proteome of VZV-infected: PXD047273; VZV ORF interactomes: PXD047821; VZV ORF effectomes: PXD047362; Validation of knockouts: PXD047575; Full proteome of MPP8-knockout cells: PXD047393; NPHP4 interactomes: PXD061602.
The data and analysis results are accessible online via the interactive web interface at https://varizonet.innatelab.org.
The VZV ORF interactomes have been submitted to the IMEx (http://www.imexconsortium.org) consortium through IntAct and assigned the identifier IM-30341.

The transcriptome data (NPHP4-knockout SK-N-BE2 cells mock- and VZV-infected) have been deposited in the European Nucleotide Archive (ENA) at EMBL-EBI under accession number PRJEB86994.

# Research involving human participants, their data, or biological material

Policy information about studies with human participants or human data. See also policy information about sex, gender (identity/presentation), and sexual orientation and race, ethnicity and racism.

| | |
|---|---|
| Reporting on sex and gender | Sex and gender of the participants are reported in Supplementary Table S6-1, for information. However, none were analyzed nor used to claim any conclusion in this study. |
| Reporting on race, ethnicity, or other socially relevant groupings | N/A. Race, ethnicity, or other socially grouping of the participants were neither analyzed nor used to claim any conclusion in this study. |
| Population characteristics | The patient cohort consisted of adults (>18 years) with a final diagnosis of VZV-associated encephalitis or meningoencephalitis or cerebral vasculitis. A total of thirteen adult patients were included on the basis of pleocytosis and a positive PCR for VZV in the cerebrospinal fluid (CSF). PCR on CSF for HSV-1, HSV-2, and enterovirus as well as bacterial cultures were also performed and were negative for all patients included. Exclusion criteria were immunosuppressive therapy, known malignant disease, pregnancy, and HIV positivity. All patients had experienced chickenpox in childhood and none had received VZV vaccination. |
| Recruitment | The patients were recruited after their admission to the Department of Infectious Diseases at Aarhus University Hospital (AUH) with a final diagnosis of VZV-associated encephalitis or meningoencephalitis or cerebral vasculitis. All referred patients were admitted and included in the study if consent to participate was obtained. We can not exclude that the most severely ill patients were not included to to unconsciousness or a rapidly fatal disease course. There is no way to correct for this but we can not exclude to have missed patients with pathological variants and thus the estimated occurence/frequency of variants may be an underestimation. |
| Ethics oversight | The patients were included following oral and written consent in accordance with The Helsinki Declaration and national ethics guidelines and after approval from the Danish National Committee on Health Research Ethics (# 1-10-72-275-15), the Data Protection Agency, and Institutional Review Board. |

Note that full information on the approval of the study protocol must also be provided in the manuscript.

# Field-specific reporting

Please select the one below that is the best fit for your research. If you are not sure, read the appropriate sections before making your selection.

☒ Life sciences　　☐ Behavioural & social sciences　　☐ Ecological, evolutionary & environmental sciences

# Life sciences study design

All studies must disclose on these points even when the disclosure is negative.

| | |
|---|---|
| Sample size | The sample sizes were chosen from past knowledge on the good sample size to ensure adequate power and reproducibility (Blainey et al., 2014 https://doi.org/10.1038/nmeth.3091; Conesa et al., 2016 https://doi.org/10.1186/s13059-016-0881-8). Sample sizes are always indicated in figure legends or related "Methods" section. |
| Data exclusions | MS runs of the effectome datasets which were not reaching satisfying performance were excluded. This led to the removal of the data for 18 VZV ORFs. |
| Replication | For mass spectrometry, in vitro viral replication validation experiments and cell viability assays, a minimum of three biological experiments were performed independently.<br>Validation of the VZV ORF32 and GTF2B interaction by co-IP was performed from two independent biological experiments, each based on independent samples analysed with a given approach (viral protein IP and reversed host protein IP).<br>Immunofluorescence analysis of VZV ORF32 and GTF2B localisation was performed on 20 cells, across two independent biological experiments. Immunofluorescence-based quantification of IFI16 expression in infected cells was performed on at least 120 cells per conditions, across three independent biological experiments. Immunofluorescence-based quantification of IFI16 expression in ORF61-expressing cells was performed on at least 500 cells per conditions, across two independent biological experiments. Immunofluorescence analysis of VZV ORF61 and UBXN7 localisation was performed on 20 cells, across three independent biological experiments.<br>All attempts at replication of the co-IP and immunofluorescence experiments were successful.<br>The host gene knockout screen on viral replication was conducted with two technical replicates per condition per sgRNA, with three to four sgRNA reagents used independently per tested gene.<br>Knockout of genes of interest were reproduced using the same sgRNA as used in the screen, in independent cell batches, and validated in two to three independent samples each analyzed by a given technique (genome sequencing, western-blot, mass spectrometry). |
| Randomization | Samples for AP-MS experiments were randomzied to avoid carry-over bias during the MS analysis. No randomization was used otherwise given the small number of samples and the lack of influence of randomization on the experimental design and experimental approach used. (no animal experiments were performed in this study). |
| Blinding | N/A. Investigators were not blinded to experimental groups (in vitro experiments required prior knowledge for data interpretation) |

# Reporting for specific materials, systems and methods

We require information from authors about some types of materials, experimental systems and methods used in many studies. Here, indicate whether each material, system or method listed is relevant to your study. If you are not sure if a list item applies to your research, read the appropriate section before selecting a response.

### Materials & experimental systems

| n/a | Involved in the study |
|---|---|
| ☐ | ☒ Antibodies |
| ☐ | ☒ Eukaryotic cell lines |
| ☒ | ☐ Palaeontology and archaeology |
| ☒ | ☐ Animals and other organisms |
| ☒ | ☐ Clinical data |
| ☒ | ☐ Dual use research of concern |
| ☒ | ☐ Plants |

### Methods

| n/a | Involved in the study |
|---|---|
| ☒ | ☐ ChIP-seq |
| ☐ | ☒ Flow cytometry |
| ☒ | ☐ MRI-based neuroimaging |

## Antibodies

| | |
|---|---|
| Antibodies used | Primary antibodies used in this study were the following: TFIIB (Cell Signaling Technology, 4169; WB 1:1000), MPP8 (Proteintech, 16796; WB 1:1500), ZNF280D (Invitrogen, PA5-56410; WB 1:1000), NPHP4 (Abclonal, A8934; WB 1:3000), UBXN7 (Sigma-Aldrich, HPA049442; IF 1:500), IFI16 (Cell Signaling Technology, 14970; WB 1:1000; IF 1:500), V5-tag rabbit (Cell Signaling Technology, 13202; WB 1:1000; IF 1:1000), V5-tag mouse (Invitrogen, R960-25; WB 1:1000; IF 1:400), HA-tag (Cell Signaling Technology, 2367; WB 1:2500; IF 1:100), HA-tag-HRP (Sigma-Aldrich, H6533; WB 1:1000), β-actin-HRP (Santa Cruz, sc-47778; WB 1:2500), β-tubulin (Cell Signaling Technology, 2128; WB 1:500). For western-blct, secondary antibodies conjugated to HRP detecting rabbit IgG (1:2500) and mouse IgG (1:5000) were purchased from Dako and Sigma-Aldrich, respectively. For immunofluorescence, DAPI (1:1000) and secondary antibody detecting rabbit or mouse IgG conjugated to Alexa 488, 594 or 647 (1:200-1:500) were purchased from Invitrogen. GFP-DyLight-488 was purchased from Rockland (600-141-215; 1:1000). |
| Validation | The supplier of the TFIIB antibody claims for specificity for Human, Mouse, Rat, Monkey species ("TFIIB (2F6A3H4) Mouse mAb detects endogenous levels of total TFIIB protein.") and provides western blot data for the evaluation of its specificity in human cells.<br>The specificity of the MPP8 antibody was validated for western blot using control and knockdown human cells in this study https://doi.org/10.1038/s41564-018-0256-x, Figure 2c.<br>The supplier of the NPHP4 antibody provides western blot data for the evaluation of its specificity in human cells. |

The supplier of the UBXN7 provides immunofluorescence data for the evaluation of its specificity in human cells. "enhanced" validations for immunofluorescence are available at https://www.proteinatlas.org/ENSG00000163960-UBXN7/summary/antibody (using human cells with variable expression levels and cross-validation with other antibodies).
The supplier of the ZNF280D antibody provides data for the evaluation of its specificity in human cells. The datasheet claims that it was tested for western blot.
The supplier of the IFI16 antibody provides western blot data for the evaluation of its specificity in human cells. The specificity was also validated for western blot using control and knockdown human cells in this study https://doi.org/10.1016/j.chom.2018.01.012, Figure 5a.
The suppliers of the V5-tag (rabbit and mouse) and the HA-tag antibodies provide western blot and immunofluorescence data for the evaluation of their specificity.
The suppliers of the HA-tag-HRP and β-actin-HRP antibodies provide western blot data for the evaluation of their specificity.
The supplier of the β-tubulin antibody provides references showing its specificity in human cells for western blot analysis.
The supplier of the GFP-DyLight-488 antibody provides immunofluorescence data showing its specifity.

# Eukaryotic cell lines

Policy information about cell lines and Sex and Gender in Research

| | |
|---|---|
| Cell line source(s) | SK-N-BE2 cells (CRL-2271) were kindly provided by Rüdiger Klein (MPI of Neurobiology, Munich, Germany). MeWo cells (HTB-65TM) were kindly provided by Abel-Viejo Borbolla (MHH, Hannover, Germany). HEK293T (CRL-11268) were purchased from ATCC. HeLa Kyoto expressing GFP-tagged GTF2B from BAC transgene was from Ina Poser. HFF-1 (SCRC-1041) cells were a kind gift from Prof. Melanie Brinkmann (HZI, Braunschweig, Germany). |
| Authentication | Authentication was performed by STR profiling. |
| Mycoplasma contamination | All cell lines were tested to be mycoplasma free by standard PCR-based assay. |
| Commonly misidentified lines (See ICLAC register) | No commonly misidentified cell lines were used in this study. |

# Plants

| | |
|---|---|
| Seed stocks | N/A |
| Novel plant genotypes | N/A |
| Authentication | N/A |

# Flow Cytometry

## Plots

Confirm that:

☒ The axis labels state the marker and fluorochrome used (e.g. CD4-FITC).

☒ The axis scales are clearly visible. Include numbers along axes only for bottom left plot of group (a 'group' is an analysis of identical markers).

☒ All plots are contour plots with outliers or pseudocolor plots.

☒ A numerical value for number of cells or percentage (with statistics) is provided.

## Methodology

| | |
|---|---|
| Sample preparation | FACS analysis of VZV rOka infection: SK-N-BE2 cells infected with VZV rOka were fixed with 3.7% Formaldehyde (Sigma-Aldrich) for 15 minutes at room temperature. After a wash in FACS buffer (1% FCS, 2mM EDTA in PBS), cells were labelled with the Varicella Zoster (VZV) DFA Kit reagent (Merck, Light Diagnostic), containing two FITC conjugated antibodies against VZV immediate-early ORF62 protein and late glycoprotein E for 15 minutes at room temperature.
FACS analysis of VZV-RFP infection: SK-N-BE2 cells knockout for the gene of interest or controls and infected with the VZV-RFP reporter virus were fixed in 1% PFA for 15 minutes and resuspended in FACS buffer (5mM EDTA pH8, 25mM Hepes, 1% FCS in PBS). |
| Instrument | VZV rOka infected samples were analyzed on an Attune NxT Acoustic Focusing Cytometer (Thermo Fisher Scientific). Reporter VZV-RFP-infected samples were analyzed on a cytoFLEX (Beckman Coulter). |

| Software | Data were acquired using the software provided with the given instrument by the supplier. FlowJo 10 was used for data analysis. |
|---|---|
| Cell population abundance | Full proteome analysis if VZV-infected cells: living cells represented (median value) 61% of the total acquired objects. Single cells represented (median value) 70% of the parent living cell population. FITC threshold was set from mock samples at 2000. FITC negative cells represented (median value) 99.8% of the single cell population in mock samples.<br>KO screen: living cells represented (median value) 74% of the total acquired objects. Single cells represented (median value) 71% of the parent living cell population. Co-cultured GFP and BFP cells represented (median value) 44% and 38%, respectively, of the parent single cell population, in mock wells. |
| Gating strategy | Forward and side scatters were used to 1) gate living cell and exclude cell debris (SSC-A/FSC-A), 2) gate single cells (FSC-H/FSC-A).<br>VZV rOka infected cells were gated from single cells as FITC positive.<br>In the reporter VZV-RFP replication screen, co-cultured knockout and control cells were gated from single cells as BFP and GFP positive cells, respectively, in order to excluded unmarked inoculum cells. The median RFP intensity of each population was used to assess VZV propagation. For follow-up validations performed in knockout cells only, knockout cells were isolated from the inoculum cells, within the single cells population, as BFP positive cells. |

☒ Tick this box to confirm that a figure exemplifying the gating strategy is provided in the Supplementary Information.

