## [Peer Review File · Nature Microbiology]

Multi-proteomic profiling of the Varicella-Zoster Virus-host interface reveals host susceptibilities to severe infection.

Corresponding Author: Dr Andreas Pichlmair

Version 0:

Reviewer comments:

Reviewer #1

(Remarks to the Author)

This comprehensive proteomics study evaluation interactions between Varicella Zoster virus (VZV) and human proteins in neuronal SK-N-BE2 cells, and identifies protein perturbations following VZV infection and VZV ORF expression. The authors use a network diffusion approach to highlight specific interactions and effects, and specifically identify interactions involved in VZV CNS pathologies by integrating patient data. Using this approach they find new VZV restriction factors, including those implicated in severe VZV patient outcomes. My suggestions are focused on how to evaluate proteomic data quality. The authors should also provide more ties back to virus-host protein interactions since otherwise the findings are correlative and less mechanistic.

Major Comments:

1. Overall the proteomics are well done. The supplementary tables are easy to navigate. I have no major concerns other than some suggestions for validation of ORFs below.
2. Data related to Fig 1: It is possible that some of these proteome changes result from proteins secreted from MeWo cells, instead of the virus released from MeWo cells. Is there a way to dissect the two?
3. I think many virologists might be hesitant regarding ORF over-expression experiments. In my opinion, this manuscript has a nice balance of biologically relevant (infection) and reductionist approaches (individual ORFs) since it is impractical to identify protein interactions for each of the 70 ORFs during infection.
 - a. However, I would like to see more quality control of ORF expression. What is their subcellular localization? Does it correspond to the known subcellular localization of these ORFs? This should be done for the ORFs where subcellular localization during infection has been established and/or antibodies exist to establish their localization during infection.
 - b. How many of these proteins colocalized during VZV infection? I see that this is done for a few protein interactions, but could be done on a larger scale to assess the quality of the data.
 - c. For effectome experiments, can any of these proteins be deleted from VZV and still result in productive infections/replication in vitro? By looking at the effectome that is lost by this deletion and comparing it to the ORF effectome, you could link reductionist data with more biologically relevant infection conditions and be more rigorous.
 - d. The discussion should include some caveats of AP-MS using over expression – mislocalization, absence of relevant protein complexes or virus replication factories.
4. Fig 2A is almost impossible to read because there are 900 host proteins. I understand it will have to look like this if you want to display all 900 proteins because of the scale of the dataset, but I would almost prefer a larger version of Fig 2B, in which you highlight enrichment categories and then go into more specific details about proteins driving those relationships with zoom in panels or just highlighting a few proteins or complexes that are relevant for the biology or the functional validation followup.
5. Related – given the scale of the work, could you make a supp information file that has interactions for each ORF on a separate 1 page network. With this approach you could highlight protein complexes more clearly. The supplementary tables are critical as a resource, but the network figures are certainly more pleasing to look at as a non-expert.
6. The authors note that ORF expression level did not correlate with number of interactors identified. However, upon inspection of the cited data, it appears to be observational, instead of a calculated correlation. Figure 2A also suggests that some viral proteins have many more interactions than others (e.g. ORF 4, 49, 63) and seem to have quite high expression compared to other ORFs in the extended data. The low expressing ORFs, seem to have fewer interactions in Figure 2 (e.g. ORF 6, 35). I recommend removing this statement or explicitly calculating the correlation between relative band intensity and number of interactions. Additionally, was there a correlation between the expression level and the number of effectors identified in the effectome? Perhaps this should be discussed as a possible source of bias for both proteomics discovery approaches. Perhaps that suggests there is more room for discovery of low-expressing ORFs, or a higher rate of false positive hits for high-expressing ORFs.
7. There are some interesting differences in ORF impact on specific proteins/complexes. For example, REST is induced by ORF 66 and 9A, but inhibited by ORF38. Is this a temporal regulation based on viral gene expression (early vs late), or does one ORF win? Placing this back in the context of infection would be valuable. Would temporal infection effectome data help resolve this?

This could be very interesting fodder for future directions.

8. IF16 degradation and ORF61 – the authors speculate this E3 ligase maybe responsible for this through protein interaction data with E3 ligase cofactors. Can they provide more concrete data that this E3 ligase is responsible for IF16 degradation and relies on VCP/UBXN7 interactions? Can you make a ORF61 mutant that is catalytically inactive to rescue IF16 expression? Does IF16 get ubiquitylated? Does VCP or UBXN7 knockdown affect IF16 degradation?
9. What qualified a protein as “prominent” to be considered for the CRISPR screen? Did you take the top 116 hits that were in the interactome and had the strongest effectome changes? Was it prominent following network diffusion in some sort of rank-ordered list?
10. The CRISPR screening approach is interesting in that it takes advantage of a mixed culture of knockout and non-knockout cells. I am not as familiar with the approach and think it would be valuable to discuss the caveats of mixed cultures. For example, could interferon expression from KO cells impact replication in control cells through a bystander effect? Would this result in misinterpretation of the phenotype or false positives/negatives?
11. For the CRISPR screen, the validated host factors studied in detail (Figure 5d-g) should be quantified using more conventional/intuitive readouts of viral replication, like plaque assay.
12. The links to patient mutations in host factors is very interesting but somewhat underdeveloped.
 - a. In the larger CRISPR screen, only 1 gRNA gave a phenotype for NPHP4. Presumably it was not initially followed up because of this and the patient data “rescued” the candidate. That is worth mentioning—it could be valuable for others in the field since screens are not perfect but can be layered with other data to gain confidence in a hit. Also, did the authors test additional gRNAs following the patient data screen? Is the guide from Fig 6 the same as the guide from the screen?
 - b. I greatly appreciate the rescue of the phenotype. The authors should explicitly state that the rescue is partial.
 - c. Aside from differences in virus replication, can the authors identify changes that are more related to patient disease? For example, what is the immune response in KO, rescue, and S862N cells?
 - d. How do the VZV protein interaction(s) relate to the replication phenotype or disease outcome? Which way does the arrow go? Does NPHP4 act on VZV ORFs, or do VZV ORFs act on NPHP4?

Minor comments:

1. Many microscopy images need a scale bar reference.
2. Many network figures could and should be larger – 4e and 4f are examples.
3. References for drug effects on VZV (acyclovir and amenamevir) would be helpful for non-VZV experts.
4. In line 2116 for the description of Figure 3, should the phrase say “expression kinetics”
5. There appear to be formatting issues with exponent notation/superscript values. In line 2122 where a p-value is stated, the number is “10-4” when it likely should be “10⁻⁴”
6. In line 1548, there may be a sentence/grammatical error when describing the use of MaxQuant and Perseus for statistical analysis: “...from MaxQuant the proteinGroup file into Perseus...”

Reviewer #2

(Remarks to the Author)

In this study by Girault and colleagues use a combination of high throughput mass spectrometry, genetic manipulation, immune fluorescence as well as clinical samples to uncover the interactome and effectome of VZV. This study is a real tour de force and the authors were able to reveal key interactions between VZV proteins and host proteins that were subsequently confirmed using tagged over-expression of viral protein and CRISPR/Cas9 genetic KO of host genes. The authors confirmed and extended our understanding of host restriction factors/antiviral proteins as well host proteins that played a key role in virus replication. The authors then validated the importance of some of the proteins identified in their screens by whole genome sequencing of patients who experienced complicated VZV infection (such as encephalitis). Overall, this work is innovative and provides novel and critical insight for the field.

The biggest weakness of this study is the use of one cell line predominantly to investigate the interactions between viral and host proteins. While this neuronal cell line is appropriate since VZV establishes latency in neurons, the virus interacts with several other cell lines, epithelial cells in the respiratory tract as well as T cells which are critical for virus trafficking. While the scope of repeating all these complex experiments in multiple cells lines is outside the scope, the conclusions should be toned to reflect this limitation. For instance, the statement that “did not observe classical antiviral and inflammatory expression signatures, pointing towards a tight control of cellular defense mechanisms by VZV” lines 176-177 is more indicative of the cell line being use since several studies have reported robust antiviral responses in primary cells infected with VZV.

The manuscript is dense, understandable given the amount of data, but the results contain a fair amount of discussion, and the discussion is quite long. It is recommended that the manuscript be streamlined to improve readability.

Reviewer #3

(Remarks to the Author)

Global assessment

The paper by Girault et al reports the results of a massive exploratory multipronged analyses, not yet done for any human herpesvirus, for VZV infections and their effects on the host proteome, and its use to dissect. The multipronged analyses involves and extensive assessment for host proteins whose levels are modulated by VZV infection (both up and down) in a model nerve like cell; the consequences of individual VZV protein expression on the up and down regulation of host proteins upon VZV protein expression, (the effectome) and the host proteins identified by individual protein expression and affinity purification through protein epitope tagging (the interactome). The resulting amount of new data is enormous and will likely be a huge jump to the initiation of many future studies. They then take a sample of implicated host genes 118 and perform CRISPR knockouts using a clever quantitative VZV infection assay to determine candidate restrictive anti VZV factors as well as proviral factors for VZV assessment and identify several novel genes. Then they use the mass proteome information to compare to whole

exon sequencing information of patients with VZV neurological diseases and using classical host gene knockout and restorations, arrive at Human mutations in host that will genetically predispose persons to more severe disease. This huge amount of information serves not only as a database, to start studies, but also as a reference base for possibly more severe VZV disease.

Good things about this paper are that it is indeed a huge amount of information to appreciate, yet the paper is very well written and for the most part clear, giving just specific examples and illustrative pathways to highlight important outcomes of this massive global analyses. It's still a lot to take in, and there is no way that the data can be comprehensively described. But this makes the paper informative to a wide audience, and will no doubt serve as the model for studies on other herpesviruses. Indeed, it was a surprise that this kind of multi-omics approach was done with VZV because VZV is not easy to work with, because it is highly cell associated and very difficult to set up synchronous infections. Maybe a comment on this? I would have thought HSV would be the better virus to start with. Nevertheless, it was done with VZV and the new data will be a big jump for this virus. The authors have also put the data on a web site that is freely accessible, so that others can look at the specifics of a chosen VZV or host protein and how it may influence infection.

There are some problems with the article that need to be addressed or explained in the text.

1. Why did the study get done with rOka, which is generated from Cosmids reverse engineered from a Vaccine strain of VZV? It is well established that the vaccine strain of VZV, vOka, is a large conglomerate of virus genotypes that is attenuated for growth in skin and likely, to some extent reactivation, all based on as yet unknown SNPS and mechanisms. rOka would represent only one of these heterogeneous genotypes. This should be recognized and discussed
 2. The VZV ORFeome is missing some important VZV genes, and the article needs to state some kind of explanation as to why these gene are missing. Little is done with ORF22, which is probably too big to do anything with. However an obvious question is why ORF28 and ORF62 are missing from most of the -eome studies. ORF62 as the main viral gene transactivator would have likely give a large interactome and given this is one of the better studied VZV proteins (along with its HSV equivalent ICP4), it would have been useful to validate the interactome. This is an important topic of discussion that should be included.
 3. There are about 20-25% of the tagged proteins that do not seem to be detectable by blotting. Why? A comment is needed. In addition, interactomes for over or underexpressed individual proteins could well be artifactual due to lack of viral partners, etc
 4. This kind of analyses is not only important for what it finds (and validates previous studies,) but the article needs some assessments for what it did not find. The limitations need to be better discussed more. They mention that pairwise proteins may be important- One example is that BAG3 has been shown to interact with ORF29 but this was not found. Many interactions and lack of interactions could be the result of mislocalization of the individual viral protein, which is minimally considered. This is a subject Worthy of discussion. They should at least state that the global assessment could be a limited dataset and many interactions are not identified. How many suspected interactions identified could be not what happens in the infected cell?
 5. Can the authors better explain network diffusion analyses? I had a hard time following their reasoning and rationale
 6. The cell type used here is a neuroblastoma -admit that it is limited and may not be very representative of VZV infections. I will admit that prior to this paper, I was unfamiliar with this line and was not aware of any studied. Please discuss
 7. Ln 304 discuss if any viral genes cause toxicity
 8. The graph of flow studies that quantify infection boosting or reduction need quadrant %ile indications in the examples provided of how VZV growth is enhanced.
- lots of small errors

Decision Letter:

23rd May 2024

Dear Andreas,

Thank you for your patience while your manuscript "Proteomic profiling of the Varicella-Zoster Virus-host interface identifies host susceptibilities to severe pathogenicity." was under peer-review at Nature Microbiology. It has now been seen by 3 referees, whose expertise and comments you will find at the end of this email. Although they find your work of some potential interest, they have raised a number of concerns that will need to be addressed before we can consider publication of the work in Nature Microbiology.

In particular, they ask to tone down the implications of the data which are not supported (Reviewer #2), discuss the limitations on how representative your dataset is given that infection was conducted in one cell line, perform additional quality checks and follow ups for your proteomic interaction data (reviewer #1) and explain whether other hits that could be relevant for pathogenesis were followed up on. In addition, reviewer #1 requests deeper investigations into the effects relevant to the disease phenotype.

Should further experimental data allow you to address these criticisms, we would be happy to look at a revised manuscript.

Please include a data availability statement as a separate section after Methods but before references, under the heading "Data Availability". This section should inform readers about the availability of the data used to support the conclusions of your study. This information includes accession codes to public repositories (data banks for protein, DNA or RNA sequences, microarray, proteomics data etc...), references to source data published alongside the paper, unique identifiers such as URLs to data repository entries, or data set DOIs, and any other statement about data availability. At a minimum, you should include the following statement: "The data that support the findings of this study are available from the corresponding author upon request", mentioning any restrictions on availability. If DOIs are provided, we also strongly encourage including these in the Reference list (authors, title, publisher (repository name), identifier, year). For more guidance on how to write this section please see: <http://www.nature.com/authors/policies/data/data-availability-statements-data-citations.pdf>

* If you have not done so already we suggest that you begin to revise your manuscript so that it conforms to our Resource format instructions at <http://www.nature.com/nmicrobiol/info/final-submission>. Refer also to any guidelines provided in this letter.

When submitting the revised version of your manuscript, please pay close attention to our [href="https://www.nature.com/nature-portfolio/editorial-policies/image-integrity">Digital Image Integrity Guidelines.](https://www.nature.com/nature-portfolio/editorial-policies/image-integrity) and to the following points below:

Link Redacted

Note: This url links to your confidential homepage and associated information about manuscripts you may have submitted or be reviewing for us. If you wish to forward this e-mail to co-authors, please delete this link to your homepage first.

Nature Microbiology is committed to improving transparency in authorship. As part of our efforts in this direction, we are now requesting that all authors identified as 'corresponding author' on published papers create and link their Open Researcher and Contributor Identifier (ORCID) with their account on the Manuscript Tracking System (MTS), prior to acceptance. This applies to primary research papers only. ORCID helps the scientific community achieve unambiguous attribution of all scholarly contributions. You can create and link your ORCID from the home page of the MTS by clicking on 'Modify my Springer Nature account'. For more information please visit www.springernature.com/orcid.

If you wish to submit a suitably revised manuscript we would hope to receive it within 6 months. If you cannot send it within this time, please let us know. We will be happy to consider your revision, even if a similar study has been accepted for publication at Nature Microbiology or published elsewhere (up to a maximum of 6 months).

Yours sincerely,

Reviewer Expertise:

Referee #1: Virology, proteomics
Referee #2: VZV
Referee #3: VZV

Reviewer Comments:

Reviewer #1 (Remarks to the Author):

This comprehensive proteomics study evaluation interactions between Varicella Zoster virus (VZV) and human proteins in

neuronal SK-N-BE2 cells, and identifies protein perturbations following VZV infection and VZV ORF expression. The authors use a network diffusion approach to highlight specific interactions and effects, and specifically identify interactions involved in VZV CNS pathologies by integrating patient data. Using this approach they find new VZV restriction factors, including those implicated in severe VZV patient outcomes. My suggestions are focused on how to evaluate proteomic data quality. The authors should also provide more ties back to virus-host protein interactions since otherwise the findings are correlative and less mechanistic.

Major Comments:

1. Overall the proteomics are well done. The supplementary tables are easy to navigate. I have no major concerns other than some suggestions for validation of ORFs below.
2. Data related to Fig 1: It is possible that some of these proteome changes result from proteins secreted from MeWo cells, instead of the virus released from MeWo cells. Is there a way to dissect the two?
3. I think many virologists might be hesitant regarding ORF over-expression experiments. In my opinion, this manuscript has a nice balance of biologically relevant (infection) and reductionist approaches (individual ORFs) since it is impractical to identify protein interactions for each of the 70 ORFs during infection.
 - a. However, I would like to see more quality control of ORF expression. What is their subcellular localization? Does it correspond to the known subcellular localization of these ORFs? This should be done for the ORFs where subcellular localization during infection has been established and/or antibodies exist to establish their localization during infection.
 - b. How many of these proteins colocalized during VZV infection? I see that this is done for a few protein interactions, but could be done on a larger scale to assess the quality of the data.
 - c. For effectome experiments, can any of these proteins be deleted from VZV and still result in productive infections/replication in vitro? By looking at the effectome that is lost by this deletion and comparing it to the ORF effectome, you could link reductionist data with more biologically relevant infection conditions and be more rigorous.
 - d. The discussion should include some caveats of AP-MS using over expression – mislocalization, absence of relevant protein complexes or virus replication factories.
4. Fig 2A is almost impossible to read because there are 900 host proteins. I understand it will have to look like this if you want to display all 900 proteins because of the scale of the dataset, but I would almost prefer a larger version of Fig 2B, in which you highlight enrichment categories and then go into more specific details about proteins driving those relationships with zoom in panels or just highlighting a few proteins or complexes that are relevant for the biology or the functional validation followup.
5. Related – given the scale of the work, could you make a supp information file that has interactions for each ORF on a separate 1 page network. With this approach you could highlight protein complexes more clearly. The supplementary tables are critical as a resource, but the network figures are certainly more pleasing to look at as a non-expert.
6. The authors note that ORF expression level did not correlate with number of interactors identified. However, upon inspection of the cited data, it appears to be observational, instead of a calculated correlation. Figure 2A also suggests that some viral proteins have many more interactions than others (e.g. ORF 4, 49, 63) and seem to have quite high expression compared to other ORFs in the extended data. The low expressing ORFs, seem to have fewer interactions in Figure 2 (e.g. ORF 6, 35). I recommend removing this statement or explicitly calculating the correlation between relative band intensity and number of interactions. Additionally, was there a correlation between the expression level and the number of effectors identified in the effectome? Perhaps this should be discussed as a possible source of bias for both proteomics discovery approaches. Perhaps that suggests there is more room for discovery of low-expressing ORFs, or a higher rate of false positive hits for high-expressing ORFs.
7. There are some interesting differences in ORF impact on specific proteins/complexes. For example, REST is induced by ORF 66 and 9A, but inhibited by ORF38. Is this a temporal regulation based on viral gene expression (early vs late), or does one ORF win? Placing this back in the context of infection would be valuable. Would temporal infection effectome data help resolve this? This could be very interesting fodder for future directions.
8. IFI16 degradation and ORF61 – the authors speculate this E3 ligase maybe responsible for this through protein interaction data with E3 ligase cofactors. Can they provide more concrete data that this E3 ligase is responsible for IFI16 degradation and relies on VCP/UBXN7 interactions? Can you make a ORF61 mutant that is catalytically inactive to rescue IFI16 expression? Does IFI16 get ubiquitinated? Does VCP or UBXN7 knockdown affect IFI16 degradation?
9. What qualified a protein as “prominent” to be considered for the CRISPR screen? Did you take the top 116 hits that were in the interactome and had the strongest effectome changes? Was it prominent following network diffusion in some sort of rank-ordered list?
10. The CRISPR screening approach is interesting in that it takes advantage of a mixed culture of knockout and non-knockout cells. I am not as familiar with the approach and think it would be valuable to discuss the caveats of mixed cultures. For example, could interferon expression from KO cells impact replication in control cells through a bystander effect? Would this result in misinterpretation of the phenotype or false positives/negatives?
11. For the CRISPR screen, the validated host factors studied in detail (Figure 5d-g) should be quantified using more conventional/intuitive readouts of viral replication, like plaque assay.
12. The links to patient mutations in host factors is very interesting but somewhat underdeveloped.
 - a. In the larger CRISPR screen, only 1 gRNA gave a phenotype for NPHP4. Presumably it was not initially followed up because of this and the patient data “rescued” the candidate. That is worth mentioning—it could be valuable for others in the field since screens are not perfect but can be layered with other data to gain confidence in a hit. Also, did the authors test additional gRNAs following the patient data screen? Is the guide from Fig 6 the same as the guide from the screen?
 - b. I greatly appreciate the rescue of the phenotype. The authors should explicitly state that the rescue is partial.
 - c. Aside from differences in virus replication, can the authors identify changes that are more related to patient disease? For example, what is the immune response in KO, rescue, and S862N cells?
 - d. How do the VZV protein interaction(s) relate to the replication phenotype or disease outcome? Which way does the arrow go? Does NPHP4 act on VZV ORFs, or do VZV ORFs act on NPHP4?

Minor comments:

1. Many microscopy images need a scale bar reference.
2. Many network figures could and should be larger – 4e and 4f are examples.
3. References for drug effects on VZV (acyclovir and amenamevir) would be helpful for non-VZV experts.
4. In line 2116 for the description of Figure 3, should the phrase say “expression kinetics”
5. There appear to be formatting issues with exponent notation/superscript values. In line 2122 where a p-value is stated, the number is “10-4” when it likely should be “10⁻⁴”
6. In line 1548, there may be a sentence/grammatical error when describing the use of MaxQuant and Perseus for statistical analysis: “...from MaxQuant the proteinGroup file into Perseus...”

Reviewer #2 (Remarks to the Author):

In this study by Girault and colleagues use a combination of high throughput mass spectrometry, genetic manipulation, immune fluorescence as well as clinical samples to uncover the interactome and effectome of VZV. This study is a real tour de force and the authors were able to reveal key interactions between VZV proteins and host proteins that were subsequently confirmed using tagged over-expression of viral protein and CRISPR/Cas9 genetic KO of host genes. The authors confirmed and extended our understanding of host restriction factors/antiviral proteins as well host proteins that played a key role in virus replication. The authors then validated the importance of some of the proteins identified in their screens by whole genome sequencing of patients who experienced complicated VZV infection (such as encephalitis). Overall, this work is innovative and provides novel and critical insight for the field.

The biggest weakness of this study is the use of one cell line predominantly to investigate the interactions between viral and host proteins. While this neuronal cell line is appropriate since VZV establishes latency in neurons, the virus interacts with several other cell lines, epithelial cells in the respiratory tract as well as T cells which are critical for virus trafficking. While the scope of repeating all these complex experiments in multiple cells lines is outside the scope, the conclusions should be toned to reflect this limitation. For instance, the statement that “did not observe classical antiviral and inflammatory expression signatures, pointing towards a tight control of cellular defense mechanisms by VZV” lines 176-177 is more indicative of the cell line being used since several studies have reported robust antiviral responses in primary cells infected with VZV.

The manuscript is dense, understandable given the amount of data, but the results contain a fair amount of discussion, and the discussion is quite long. It is recommended that the manuscript be streamlined to improve readability.

Reviewer #3 (Remarks to the Author):

Global assessment

The paper by Girault et al reports the results of a massive exploratory multipronged analyses, not yet done for any human herpesvirus, for VZV infections and their effects on the host proteome, and its use to dissect. The multipronged analyses involves and extensive assessment for host proteins whose levels are modulated by VZV infection (both up and down) in a model nerve like cell; the consequences of individual VZV protein expression on the up and down regulation of host proteins upon VZV protein expression, (the effectome) and the host proteins identified by individual protein expression and affinity purification through protein epitope tagging (the interactome). The resulting amount of new data is enormous and will likely be a huge jump to the initiation of many future studies. They then take a sample of implicated host genes 118 and perform CRISPR knockouts using a clever quantitative VZV infection assay to determine candidate restrictive anti VZV factors as well as proviral factors for VZV assessment and identify several novel genes. Then they use the mass proteome information to compare to whole exon sequencing information of patients with VZV neurological diseases and using classical host gene knockout and restorations, arrive at Human mutations in host that will genetically predispose persons to more severe disease. This huge amount of information serves not only as a database, to start studies, but also as a reference base for possibly more severe VZV disease.

Good things about this paper are that it is indeed a huge amount of information to appreciate, yet the paper is very well written and for the most part clear, giving just specific examples and illustrative pathways to highlight important outcomes of this massive global analyses. It's still a lot to take in, and there is no way that the data can be comprehensively described. But this makes the paper informative to a wide audience, and will no doubt serve as the model for studies on other herpesviruses. Indeed, it was a surprise that this kind of multi-omics approach was done with VZV because VZV is not easy to work with, because it is highly cell associated and very difficult to set up synchronous infections. Maybe a comment on this? I would have thought HSV would be the better virus to start with. Nevertheless, it was done with VZV and the new data will be a big jump for this virus. The authors have also put the data on a web site that is freely accessible, so that others can look at the specifics of a chosen VZV or host protein and how it may influence infection.

There are some problems with the article that need to be addressed or explained in the text.

1. Why did the study get done with rOka, which is generated from Cosmids reverse engineered from a Vaccine strain of VZV? It is well established that the vaccine strain of VZV, vOka, is a large conglomerate of virus genotypes that is attenuated for growth in skin and likely, to some extent reactivation, all based on as yet unknown SNPs and mechanisms. rOka would represent only one of these heterogeneous genotypes. This should be recognized and discussed
2. The VZV ORFeome is missing some important VZV genes, and the article needs to state some kind of explanation as to why these gene are missing. Little is done with ORF22, which is probably too big to do anything with. However an obvious question is why ORF28 and ORF62 are missing from most of the -eome studies. ORF62 as the main viral gene transactivator would have likely give a large interactome and given this is one of the better studied VZV proteins (along with its HSV equivalent ICP4), it would have been useful to validate the interactome. This is an important topic of discussion that should be included.
3. There are about 20-25% of the tagged proteins that do not seem to be detectable by blotting. Why? A comment is needed. In addition, interactomes for over or underexpressed individual proteins could well be artifactual due to lack of viral partners, etc
4. This kind of analyses is not only important for what it finds (and validates previous studies,) but the article needs some

assessments for what it did not find. The limitations need to be better discussed more. They mention that pairwise proteins may be important- One example is that BAG3 has been shown to interact with ORF29 but this was not found. Many interactions and lack of interactions could be the result of mislocalization of the individual viral protein, which is minimally considered. This is a subject Worthy of discussion. They should at least state that the global assessment could be a limited dataset and many interactions are not identified. How many suspected interactions identified could be not what happens in the infected cell?

5. Can the authors better explain network diffusion analyses? I had a hard time following their reasoning and rationale

6. The cell type used here is a neuroblastoma -admit that it is limited and may not be very representative of VZV infections. I will admit that prior to this paper, I was unfamiliar with this line and was not aware of any studied. Please discuss

7. Ln 304 discuss if any viral genes cause toxicity

8. The graph of flow studies that quantify infection boosting or reduction need quadrant %ile indications in the examples provided of how VZV growth is enhanced.

lots of small errors

Version 1:

Reviewer comments:

Reviewer #1

(Remarks to the Author)

The authors have addressed all of my comments. I understand that many comments may be out of the scope of this already large body of work. I appreciate the lengths the authors went to and feel that the manuscript is significantly improved by these changes. I was able to successfully visit their website and found it to be a useful resource to look for my favorite host proteins. As a very minor point, Figure 6e is a bit confusing. The logic of the cartoon suggests that mutant NHPH4 inhibits the antiviral function of WT NHPH4 instead of losing the antiviral activity.

(Remarks on code availability)

I will start by stating that some of the code is in Julia, which I have not used before. I recommend reaching out to an expert in that language if you would like more expert review of that code.

o Are you satisfied that all data and source code needed to reproduce the results of the paper have been made available? -- I think this is an impossible question to answer with 100% confidence. Generally speaking, I think they have provided much of the code and the authors would be responsive to questions if asked.

o Are you satisfied that the results can be replicated using the code/software and dataset provided in the study?

o Were you able to run the tool successfully? -- I did not run any code as that would require input files I did not have (msimport) or knowledge of Julia (HierarchicalHotNet). Some example input files could be I reviewed the code in Rstudio and VSCode to see if I could understand the structure of the package, documentation, README files, and underlying source code.

o Was the code sufficiently documented to allow another researcher to follow the algorithm? -- generally speaking, yes. Some of the base code could use additional commenting on the purpose of the code, but not to the extent that it should preclude publication. Overall it is nothing that someone with coding background or a novice coder with the help of an AI could not figure out. Please note that I did not put these codes into an AI, but it could be done.

o Can the software be run on a widely available operating system?

Yes, Rstudio, VSCode, or Jupyter

o To your knowledge, do available tools or software exist that perform in a similar way to the reported software?

Yes, many groups have in house scripts to manage MS data outputs from Spectronaut, MaxQuant, etc. HierarchicalHotNet was previously built in Python, as the authors note.

Reviewer #2

(Remarks to the Author)

I commend the authors for their very thoughtful and thorough response to reviewers' comments. The manuscript is substantively improved and will be a great resource for the community.

(Remarks on code availability)

Reviewer #3

(Remarks to the Author)

The paper by Girault et al is a Tour de Force work that will have a very large and significant impact on the VZV and Herpesvirus field. No such similar analyses has been done on any human herpesvirus to date, and the authors are to be congratulated and the broad set of results that are presented in this work. This reviewer was one of those who reviewed the original manuscript, where i presented a number of important concerns. I have gone through this very carefully. Indeed there may be a problem with the journal in that it is extensive, has a huge amount of data and is quite long. Nevertheless there is a wealth of important information that numerous investigators will be referring to for many years as they seek to address one of two VZV proteins. For

the most part, I think the authors have done a strong job in trying to address as many of the comments as possible to the previous version of the manuscript, and even more data is provided to strengthen some of the key interactions and effectomes. I have no problem with the data obtained, and it is now presented in table format, figure format and as web accessible information. While Figure 2 is impressive, it is not so useful, but I can accept the arguments of the authors to show it. I have two comments that were raised previously that need to be addressed, and both concern not what is there, but what might be missing. It is likely not complete, as the authors recognize

- 1) The vast majority of VZV infection studies were done using rOka generated by Cohen and Seidel in 1993. IT MUST be acknowledged in the discussion that rOka is derived from Cosmids derived from the attenuated vaccine virus Oka, has several of the coding SNP mutations associated with the vaccine. Since vaccine Oka is attenuated for viral growth in skin, and reduces reactivation efficiency from neuronal latency, some critical interactions and protein changes may not have been captured in the approaches taken.
- 2) The limitation on cell type specific studies is probably much more important than the authors acknowledge. VZV infects multiple cell types in human infections, including differentiating skin epidermis and T cells. This work could miss a huge number of cell specific interactions and variations.
- 3) That being said, the discussion is still very long. But given the mass of information provided, that is, perhaps, to be forgiven.
- 4) Most proteins were expressed and purified by epitope tagging, in the absence of other proteins. It should be acknowledged that tagging of some proteins may destroy protein function/protein interaction. This may contribute to some missing interactions. I found the statement lines For example, ORF23 does not tolerate a C terminal tag in the context of virus, and it is C tagged in this study. ORF4 function is also changed by C -epitope tagging. This is worthy of a note of caution in the discussion

(Remarks on code availability)

Decision Letter:

Our ref: NMICROBIOL-24030882A

15th May 2025

Dear Andreas,

Thank you for submitting your revised manuscript "Proteomic profiling of the Varicella-Zoster Virus-host interface identifies host susceptibilities to severe pathogenicity." (NMICROBIOL-24030882A). It has now been seen by the original referees and their comments are below. The reviewers find that the paper has improved in revision, and therefore we'll be happy in principle to publish it in Nature Microbiology, pending minor revisions to satisfy the referees' final requests and to comply with our editorial and formatting guidelines.

We would need you to improve the code according to the suggestions from referee #1. While it is unlikely we will send the manuscript back to peer review at this stage, we would ask for further referee feedback if the code as clearly not improved.

Thank you again for your interest in Nature Microbiology Please do not hesitate to contact me if you have any questions.

Sincerely,

Reviewer #1 (Remarks to the Author):

The authors have addressed all of my comments. I understand that many comments may be out of the scope of this already large body of work. I appreciate the lengths the authors went to and feel that the manuscript is significantly improved by these changes. I was able to successfully visit their website and found it to be a useful resource to look for my favorite host proteins. As a very minor point, Figure 6e is a bit confusing. The logic of the cartoon suggests that mutant NHPH4 inhibits the antiviral function of WT NHPH4 instead of losing the antiviral activity.

Reviewer #1 (Remarks on code availability):

I will start by stating that some of the code is in Julia, which I have not used before. I recommend reaching out to an expert in that language if you would like more expert review of that code.

o Are you satisfied that all data and source code needed to reproduce the results of the paper have been made available? -- I think this is an impossible question to answer with 100% confidence. Generally speaking, I think they have provided much of the

code and the authors would be responsive to questions if asked.

o Are you satisfied that the results can be replicated using the code/software and dataset provided in the study?

o Were you able to run the tool successfully? -- I did not run any code as that would require input files I did not have (msimport) or knowledge of Julia (HierarchicalHotNet). Some example input files could be I reviewed the code in Rstudio and VSCode to see if I could understand the structure of the package, documentation, README files, and underlying source code.

o Was the code sufficiently documented to allow another researcher to follow the algorithm? -- generally speaking, yes. Some of the base code could use additional commenting on the purpose of the code, but not to the extent that it should preclude publication. Overall it is nothing that someone with coding background or a novice coder with the help of an AI could not figure out. Please note that I did not put these codes into an AI, but it could be done.

o Can the software be run on a widely available operating system?

Yes, Rstudio, VSCode, or Jupyter

o To your knowledge, do available tools or software exist that perform in a similar way to the reported software?

Yes, many groups have in house scripts to manage MS data outputs from Spectronaut, MaxQuant, etc. HierarchicalHotNet was previously built in Python, as the authors note.

Reviewer #2 (Remarks to the Author):

I commend the authors for their very thoughtful and thorough response to reviewers' comments. The manuscript is substantively improved and will be a great resource for the community.

Reviewer #3 (Remarks to the Author):

The paper by Girault et al is a Tour de Force work that will have a very large and significant impact on the VZV and Herpesvirus field. No such similar analyses has been done on any human herpesvirus to date, and the authors are to be congratulated and the broad set of results that are presented in this work. This reviewer was one of those who reviewed the original manuscript, where I presented a number of important concerns. I have gone through this very carefully. Indeed there may be a problem with the journal in that it is extensive, has a huge amount of data and is quite long. Nevertheless there is a wealth of important information that numerous investigators will be referring to for many years as they seek to address one of two VZV proteins. For the most part, I think the authors have done a strong job in trying to address as many of the comments as possible to the previous version of the manuscript, and even more data is provided to strengthen some of the key interactions and effectomes. I have no problem with the data obtained, and it is now presented in table format, figure format and as web accessible information. While Figure 2 is impressive, it is not so useful, but I can accept the arguments of the authors to show it. I have two comments that were raised previously that need to be addressed, and both concern not what is there, but what might be missing. It is likely not complete, as the authors recognize

1) The vast majority of VZV infection studies were done using rOka generated by Cohen and Seidel in 1993. IT MUST be acknowledged in the discussion that rOka is derived from Cosmids derived from the attenuated vaccine virus Oka, has several of the coding SNP mutations associated with the vaccine. Since vaccine Oka is attenuated for viral growth in skin, and reduces reactivation efficiency from neuronal latency, some critical interactions and protein changes may not have been captured in the approaches taken.

2) The limitation on cell type specific studies is probably much more important than the authors acknowledge. VZV infects multiple cell types in human infections, including differentiating skin epidermis and T cells. This work could miss a huge number of cell specific interactions and variations.

3) That being said, the discussion is still very long. But given the mass of information provided, that is, perhaps, to be forgiven.

4) Most proteins were expressed and purified by epitope tagging, in the absence of other proteins. It should be acknowledged that tagging of some proteins may destroy protein function/protein interaction. This may contribute to some missing interactions. I found the statement lines For example, ORF23 does not tolerate a C terminal tag in the context of virus, and it is C tagged in this study. ORF4 function is also changed by C -epitope tagging. This is worthy of a note of caution in the discussion

Version 2:

Decision Letter:

23rd June 2025

Dear Andreas,

I am pleased to accept your Resource "Multi-proteomic profiling of the Varicella-Zoster Virus-host interface reveals host susceptibilities to severe infection." for publication in Nature Microbiology. Thank you for having chosen to submit your work to us and many congratulations.

Authors may need to take specific actions to achieve [compliance](https://www.springernature.com/gp/open-research/funding/policy-compliance-faqs) with funder and institutional open access mandates. If your research is supported by a funder that requires immediate open access (e.g. according to [Plan S principles](https://www.springernature.com/gp/open-research/plan-s-compliance)) then you should select the gold OA route, and we will direct you to the compliant route where possible. For authors selecting the subscription publication route, the journal's standard licensing terms will need to be accepted, including [self-archiving policies](https://www.nature.com/nature-portfolio/editorial-policies/self-archiving-and-license-to-publish). Those licensing terms will supersede any other terms that the author or any third party may assert apply to any version of the manuscript.

As soon as your article is published, you will receive an automated email with your shareable link. Congrats again to you and your co-authors! I am looking forward to seeing your paper published!

With kind regards,

P.S. Click on the following link if you would like to recommend Nature Microbiology to your librarian
<http://www.nature.com/subscriptions/recommend.html#forms>

** Visit the Springer Nature Editorial and Publishing website at http://editorial-jobs.springernature.com?utm_source=ejP_NMicro_email&utm_medium=ejP_NMicro_email&utm_campaign=ejp_NMicro for more information about our career opportunities. If you have any questions please click [a](http://www.springernature.com/editorial-and-publishing-jobs)

href="mailto:editorial.publishing.jobs@springernature.com">here.**

Open Access This Peer Review File is licensed under a Creative Commons Attribution 4.0 International License, which permits use, sharing, adaptation, distribution and reproduction in any medium or format, as long as you give appropriate credit to the original author(s) and the source, provide a link to the Creative Commons license, and indicate if changes were made. In cases where reviewers are anonymous, credit should be given to 'Anonymous Referee' and the source. The images or other third party material in this Peer Review File are included in the article's Creative Commons license, unless indicated otherwise in a credit line to the material. If material is not included in the article's Creative Commons license and your intended use is not permitted by statutory regulation or exceeds the permitted use, you will need to obtain permission directly from the copyright holder.
